# EXPECTED GRADIENTS OF MAXOUT NETWORKS AND CONSEQUENCES TO PARAMETER INITIALIZATION

## ABSTRACT

We study the gradients of a maxout network with respect to inputs and parameters and obtain bounds for the moments depending on the architecture and the parameter distribution. We observe that the distribution of the input-output Jacobian depends on the input, which complicates a stable parameter initialization. Based on the moments of the gradients, we formulate parameter initialization strategies that avoid vanishing and exploding gradients in wide networks. Experiments with deep fully-connected and convolutional networks show that this strategy improves SGD and Adam training of deep maxout networks. In addition, we obtain refined bounds on the expected number of linear regions, results on the expected curve length distortion, and results on the NTK.

## 1 INTRODUCTION

We study the gradients of maxout networks and derive several implications for training stability, parameter initialization, and expressivity. Concretely, we compute stochastic order bounds and bounds on the moments depending on the parameter distribution and the network architecture. The analysis is based on the input-output Jacobian of maxout networks. We discover that, in contrast to ReLU networks, when initialized with a zero-mean Gaussian distribution, the distribution of the input-output Jacobian of a maxout network depends on the network input, which may lead to unstable gradients and training difficulties. Nonetheless, we can obtain a rigorous parameter initialization recommendation for wide networks. The analysis of gradients also allows us to refine previous bounds on the expected number of linear regions of maxout networks at initialization and derive new results on the length distortion and the NTK.

**Maxout networks**    A rank-$K$ maxout unit, introduced by Goodfellow et al. (2013), computes the maximum of $K$ real-valued parametric affine functions. Concretely, a rank-$K$ maxout unit with $n$ inputs implements a function $\mathbb{R}^n \to \mathbb{R}; \mathbf{x} \mapsto \max_{k \in [K]}\{\langle W_k, \mathbf{x} \rangle + b_k\}$, where $W_k \in \mathbb{R}^n$ and $b_k \in \mathbb{R}$, $k \in [K] := \{1, \ldots, K\}$, are trainable weights and biases. The $K$ arguments of the maximum are called the pre-activation features of the maxout unit. This may be regarded as a multi-argument generalization of a ReLU, which computes the maximum of a real-valued affine function and zero. Goodfellow et al. (2013) demonstrated that maxout networks could perform better than ReLU networks under similar circumstances. Additionally, maxout networks have been shown to be useful for combating catastrophic forgetting in neural networks (Goodfellow et al., 2015). On the other hand, Castaneda et al. (2019) evaluated the performance of maxout networks in a big data setting and observed that increasing the width of ReLU networks is more effective in improving performance than replacing ReLUs with maxout units and that ReLU networks converge faster than maxout networks. We observe that proper initialization strategies for maxout networks have not been studied in the same level of detail as for ReLU networks and that this might resolve some of the problems encountered in previous maxout network applications.

**Parameter initialization**    The vanishing and exploding gradient problem has been known since the work of Hochreiter (1991). It makes choosing an appropriate learning rate harder and slows training Sun (2019). Common approaches to address this difficulty include the choice of specific architectures, e.g. LSTMs (Hochreiter, 1991) or ResNets (He et al., 2016), and normalization methods such as batch normalization (Ioffe & Szegedy, 2015) or explicit control of the gradient magnitude with gradient clipping (Pascanu et al., 2013). We will focus on approaches based on parameter initialization that control the activation length and parameter gradients (LeCun et al., 2012; Glorot & Bengio, 2010;

He et al., 2015; Gurbuzbalaban & Hu, 2021; Zhang et al., 2019; Bachlechner et al., 2021). He et al. (2015) studied forward and backward passes to obtain initialization recommendations for ReLU. A more rigorous analysis of the gradients was performed by Hanin & Rolnick (2018); Hanin (2018), who also considered higher-order moments and derived recommendations on the network architecture. Sun et al. (2018) derived a corresponding strategy for rank $K = 2$ maxout networks. For higher maxout ranks, Tseran & Montúfar (2021) considered balancing the forward pass, assuming Gaussian or uniform distribution on the pre-activation features of each layer. However, this assumption is not fully justified. We will analyze maxout network gradients, including the higher order moments, and give a rigorous justification for the initialization suggested by Tseran & Montúfar (2021).

**Expected number of linear regions**    Neural networks with piecewise linear activation functions subdivide their input space into linear regions, i.e., regions over which the computed function is (affine) linear. The number of linear regions serves as a complexity measure to differentiate network architectures (Pascanu et al., 2014; Montufar et al., 2014; Telgarsky, 2015; 2016). The first results on the expected number of linear regions were obtained by Hanin & Rolnick (2019a;b) for ReLU networks, showing that it can be much smaller than the maximum possible number. Tseran & Montúfar (2021) obtained corresponding results for maxout networks. An important factor controlling the bounds in these works is a constant depending on the gradient of the neuron activations with respect to the network input. By studying the input-output Jacobian of maxout networks, we obtain a refined bound for this constant and, consequently, the expected number of linear regions.

**Expected curve distortion**    Another complexity measure is the distortion of the length of an input curve as it passes through a network. Poole et al. (2016) studied the propagation of Riemannian curvature through wide neural networks using a mean-field approach, and later, a related notion of "trajectory length" was considered by Raghu et al. (2017). It was demonstrated that these measures can grow exponentially with the network depth, which was linked to the ability of deep networks to "disentangle" complex representations. Based on these notions, Murray et al. (2022) studies how to avoid rapid convergence of pairwise input correlations, vanishing and exploding gradients. However, Hanin et al. (2021) proved that for a ReLU network with He initialization the length of the curve does not grow with the depth and even shrinks slightly. We establish similar results for maxout networks.

**NTK**    It is known that the Neural Tangent Kernel (NTK) of a finite network can be approximated by its expectation (Jacot et al., 2018). However, for ReLU networks Hanin & Nica (2020a) showed that if both the depth and width tend to infinity, the NTK does not converge to a constant in probability. By studying the expectation of the gradients, we show that similarly to ReLU, the NTK of maxout networks does not converge to a constant when both width and depth are sent to infinity.

**Contributions**    Our contributions can be summarized as follows.

- For **expected gradients**, we derive stochastic order bounds for the directional derivative of the input-output map of a deep fully-connected maxout network (Theorem 1) as well as bounds for the moments (Corollary 2). Additionally, we derive an equality in distribution for the directional derivatives (Theorem 3), based on which we also discuss the moments (Remark 4) in wide networks. We further derive the moments of the activation length of a fully-connected maxout network (Corollary 5).

- We rigorously derive **parameter initialization** guidelines for wide maxout networks preventing vanishing and exploding gradients and formulate architecture recommendations. We experimentally demonstrate that they make it possible to train standard-width deep fully-connected and convolutional maxout networks using simple procedures (such as SGD with momentum and Adam), yielding higher accuracy than other initializations or ReLU networks on image classification tasks.

- We derive **several implications** refining previous bounds on the expected number of linear regions (Corollary 6), and new results on length distortion (Corollary 7) and the NTK (Corollary 9).

## 2 PRELIMINARIES

**Architecture**    We consider feedforward fully-connected maxout neural networks with $n_0$ inputs, $L$ hidden layers of widths $n_1, \ldots, n_{L-1}$, and a linear output layer, which implement functions of the form $\mathcal{N} = \psi \circ \phi_{L-1} \circ \cdots \circ \phi_1$. The $l$-th hidden layer is a function $\phi_l \colon \mathbb{R}^{n_{l-1}} \to \mathbb{R}^{n_l}$ with components

$i \in [n_l] := \{1, \ldots, n_l\}$ given by the maximum of $K \geq 2$ trainable affine functions $\phi_{l,i} \colon \mathbb{R}^{n_{l-1}} \to \mathbb{R}$; $\mathbf{x}^{(l-1)} \mapsto \max_{k \in [K]} \{W_{i,k}^{(l)} \mathbf{x}^{(l-1)} + \mathbf{b}_{i,k}^{(l)}\}$, where $W_{i,k}^{(l)} \in \mathbb{R}^{n_{l-1}}$, $\mathbf{b}_{i,k} \in \mathbb{R}$. Here $\mathbf{x}^{(l-1)} \in \mathbb{R}^{n_{l-1}}$ denotes the output of the $(l-1)$th layer and $\mathbf{x}^{(0)} := \mathbf{x}$. We will write $\mathbf{x}_{i,k}^{(l)} = W_{i,k}^{(l)} \mathbf{x}^{(l-1)} + \mathbf{b}_{i,k}^{(l)}$ to denote the $k$th pre-activation of the $i$th neuron in the $l$th layer. Finally $\psi \colon \mathbb{R}^{n_{L-1}} \to \mathbb{R}^{n_L}$ is a linear output layer. We will write $\mathbf{\Theta} = \{\mathbf{W}, \mathbf{b}\}$ for the parameters. Unless stated otherwise, we assume that for each layer, the weights and biases are initialized as i.i.d. samples from a Gaussian distribution with mean 0 and variance $c/n_{l-1}$, where $c$ is a positive constant. For the linear output layer, the variance is set as $1/n_{L-1}$. We shall study appropriate choices of $c$. We will use $\|\cdot\|$ to denote the $\ell_2$ vector norm. We recall that a real valued random variable $X$ is said to be smaller than $Y$ in the stochastic order, denoted by $X \leq_{st} Y$, if $\Pr(X > x) \leq \Pr(Y > x)$ for all $x \in \mathbb{R}$. In Appendix A we review basic notions about maxout networks and random variables that we will use in our results.

**Input-output Jacobian and activation length** We are concerned with the gradients of the outputs with respect to the inputs, $\nabla \mathcal{N}_i(\mathbf{x}) = \nabla_{\mathbf{x}} \mathcal{N}_i$, and with respect to the parameters, $\nabla \mathcal{N}_i(\mathbf{\Theta}) = \nabla_{\mathbf{\Theta}} \mathcal{N}_i$. In our notation, the argument indicates the variables with respect to which we are taking the derivatives. To study these gradients, we consider the input-output Jacobian $\mathbf{J}_{\mathcal{N}}(\mathbf{x}) = [\nabla \mathcal{N}_1(\mathbf{x}), \ldots, \nabla \mathcal{N}_{n_L}(\mathbf{x})]^T$. To see the connection to the gradient with respect to the network parameters, consider any loss function $\mathcal{L} \colon \mathbb{R}^{n_L} \to \mathbb{R}$. A short calculation shows that, for a fixed input $\mathbf{x} \in \mathbb{R}^{n_0}$, the derivative of the loss with respect to one of the weights $W_{i,k',j}^{(l)}$ of a maxout unit is $\langle \nabla \mathcal{L}(\mathcal{N}(\mathbf{x})), \mathbf{J}_{\mathcal{N}}(\mathbf{x}_i^{(l)}) \rangle \mathbf{x}_j^{(l-1)}$ if $k' = \operatorname{argmax}_k \{\mathbf{x}_{i,k}^{(l)}\}$ and zero otherwise, i.e.

$$\frac{\partial \mathcal{L}(\mathbf{x})}{\partial W_{i,k',j}^{(l)}} = C(\mathbf{x}, W) \|\mathbf{J}_{\mathcal{N}}\left(\mathbf{x}^{(l)}\right) \mathbf{u}\| \, \mathbf{x}_j^{(l-1)}, \tag{1}$$

where $C(\mathbf{x}, W) := \|\mathbf{J}_{\mathcal{N}}(\mathbf{x}_i^{(l)})\|^{-1} \langle \nabla \mathcal{L}(\mathcal{N}(\mathbf{x})), \mathbf{J}_{\mathcal{N}}(\mathbf{x}_i^{(l)}) \rangle$ and $\mathbf{u} = \mathbf{e}_i \in \mathbb{R}^{n_l}$. A similar decomposition of the derivative was used by Hanin (2018); Hanin & Rolnick (2018) for ReLU networks. By (1) the fluctuation of the gradient norm around its mean is captured by the joint distribution of the squared norm of the directional derivative $\|\mathbf{J}_{\mathcal{N}}(\mathbf{x})\mathbf{u}\|^2$ and the normalized activation length $A^{(l)} = \|\mathbf{x}^{(l)}\|^2 / n_l$. We also observe that $\|\mathbf{J}_{\mathcal{N}}(\mathbf{x})\mathbf{u}\|^2$ is related to the singular values of the input-output Jacobian, which is of interest since a spectrum concentrated around one at initialization can speed up training (Saxe et al., 2014; Pennington et al., 2017; 2018): First, the sum of singular values is $\operatorname{tr}(\mathbf{J}_{\mathcal{N}}(\mathbf{x})^T \mathbf{J}_{\mathcal{N}}(\mathbf{x})) = \sum_{i=1}^{n_L} \langle \mathbf{J}_{\mathcal{N}}(\mathbf{x})^T \mathbf{J}_{\mathcal{N}}(\mathbf{x})\mathbf{u}_i, \mathbf{u}_i \rangle = \sum_{i=1}^{n_L} \|\mathbf{J}_{\mathcal{N}}(\mathbf{x})\mathbf{u}_i\|^2$, where the vectors $\mathbf{u}_i$ form an orthonormal basis. Second, using the Stieltjes transform, one can show that singular values of the Jacobian depend on the even moments of the entries of $\mathbf{J}_{\mathcal{N}}$ (Hanin, 2018, Section 3.1).

## 3 RESULTS

### 3.1 BOUNDS ON THE INPUT-OUTPUT JACOBIAN

**Theorem 1** (Bounds on $\|\mathbf{J}_{\mathcal{N}}(\mathbf{x})\mathbf{u}\|^2$). *Consider a maxout network with the settings of Section 2. Assume that the biases are independent of the weights but otherwise initialized using any approach. Let $\mathbf{u} \in \mathbb{R}^{n_0}$ be a fixed unit vector. Then, almost surely with respect to the parameter initialization, for any input into the network $\mathbf{x} \in \mathbb{R}^{n_0}$, the following stochastic order bounds hold:*

$$\frac{1}{n_0} \chi_{n_L}^2 \prod_{l=1}^{L-1} \frac{c}{n_l} \sum_{i=1}^{n_l} \xi_{l,i}(\chi_1^2, K) \leq_{st} \|\mathbf{J}_{\mathcal{N}}(\mathbf{x})\mathbf{u}\|^2 \leq_{st} \frac{1}{n_0} \chi_{n_L}^2 \prod_{l=1}^{L-1} \frac{c}{n_l} \sum_{i=1}^{n_l} \Xi_{l,i}(\chi_1^2, K),$$

*where $\xi_{l,i}(\chi_1^2, K)$ and $\Xi_{l,i}(\chi_1^2, K)$ are respectively the smallest and largest order statistic in a sample of size $K$ of chi-squared random variables with 1 degree of freedom, independent of each other and of the vectors $\mathbf{u}$ and $\mathbf{x}$.*

The proof is in Appendix B. It is based on appropriate modifications to the ReLU discussion of Hanin & Nica (2020b); Hanin et al. (2021) and proceeds by writing the Jacobian norm as the product of the layer norms and bounding them with $\min_{k \in [K]} \{\langle W_{i,k}^{(l)}, \mathbf{u}^{(l-1)} \rangle^2\}$ and $\max_{k \in [K]} \{\langle W_{i,k}^{(l)}, \mathbf{u}^{(l-1)} \rangle^2\}$. Since the product of a Gaussian vector with a unit vector is always Gaussian, the lower and upper bounds are distributed as the smallest and largest order statistics in a sample of size $K$ of chi-squared

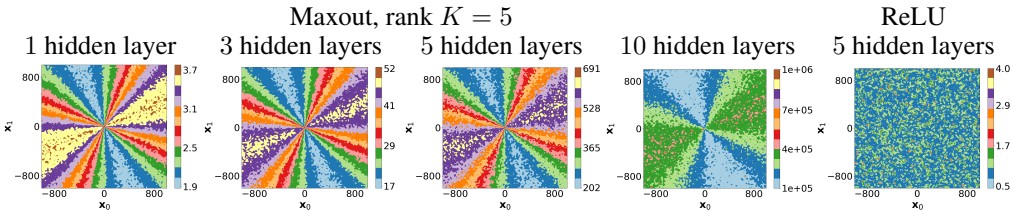

Figure 1: Expectation of the directional derivative of the input-output map $\mathbb{E}[\|\mathbf{J}_{\mathcal{N}}(\mathbf{x})\mathbf{u}\|^2]$ for width-2 fully-connected networks with inputs in $\mathbb{R}^2$. For maxout networks, this expectation depends on the input, while for ReLU networks, it does not. Input points $\mathbf{x}$ were generated as a grid of $100 \times 100$ points in $[-10^3, 10^3]^2$, and $\mathbf{u}$ was a fixed vector sampled from the unit sphere. The expectation was estimated based on 10,000 initializations with weights and biases sampled from $N(0, 1)$.

random variables with 1 degree of freedom. In contrast to ReLU networks, we found that for maxout networks, it is impossible to write equality in distribution involving only independent random variables because of the dependency of the distribution of $\|\mathbf{J}_{\mathcal{N}}(\mathbf{x})\mathbf{u}\|^2$ on the network input $\mathbf{x}$ and the direction vector $\mathbf{u}$ (see Figure 1). We discuss this in more detail in Section 3.2.

**Corollary 2** (Bounds on the moments of $\|\mathbf{J}_{\mathcal{N}}(\mathbf{x})\mathbf{u}\|^2$). *Consider a maxout network with the settings of Section 2. Assume that the biases are independent of the weights but otherwise initialized using any approach. Let $\mathbf{u} \in \mathbb{R}^{n_0}$ be a fixed unit vector and $\mathbf{x} \in \mathbb{R}^{n_0}$ be any input into the network, Then*

*(i)* $\dfrac{n_L}{n_0}(c\mathcal{S})^{L-1} \leq \mathbb{E}[\|\mathbf{J}_{\mathcal{N}}(\mathbf{x})\mathbf{u}\|^2] \leq \dfrac{n_L}{n_0}(c\mathcal{L})^{L-1},$

*(ii)* $\mathrm{Var}\left[\|\mathbf{J}_{\mathcal{N}}(\mathbf{x})\mathbf{u}\|^2\right] \leq \left(\dfrac{n_L}{n_0}\right)^2 c^{2(L-1)}\left(K^{2(L-1)}\exp\left\{4\left(\sum_{l=1}^{L-1}\dfrac{1}{n_l K} + \dfrac{1}{n_L}\right)\right\} - \mathcal{S}^{2(L-1)}\right),$

*(iii)* $\mathbb{E}\left[\|\mathbf{J}_{\mathcal{N}}(\mathbf{x})\mathbf{u}\|^{2t}\right] \leq \left(\dfrac{n_L}{n_0}\right)^t (cK)^{t(L-1)}\exp\left\{t^2\left(\sum_{l=1}^{L-1}\dfrac{1}{n_l K} + \dfrac{1}{n_L}\right)\right\}, \quad t \in \mathbb{N},$

*where the expectation is taken with respect to the distribution of the network weights. The constants $\mathcal{S}$ and $\mathcal{L}$ depend on $K$ and denote the means of the smallest and the largest order statistic in a sample of $K$ chi-squared random variables. For $K = 2, \ldots, 10$, $\mathcal{S} \in [0.02, 0.4]$ and $\mathcal{L} \in [1.6, 4]$. See Table 4 in Appendix C for the exact values.*

Notice that for $t \geq 2$, the $t$th moments of the input-output Jacobian depend on the architecture of the network, but the mean does not (Corollary 2), similarly to their behavior in ReLU networks Hanin (2018). We also observe that the upper bound on the $t$th moments can grow exponentially with the network depth depending on the maxout rank. However, the upper bound on the moments can be tightened provided corresponding bounds for the largest order statistics of the chi-squared distribution.

### 3.2 DISTRIBUTION OF THE INPUT-OUTPUT JACOBIAN

Here we present the equality in distribution for the input-output Jacobian. It contains dependent variables for the individual layers and thus cannot be readily used to obtain bounds on the moments, but it is particularly helpful for studying the behavior of wide maxout networks.

**Theorem 3** (Equality in distribution for $\|\mathbf{J}_{\mathcal{N}}(\mathbf{x})\mathbf{u}\|^2$). *Consider a maxout network with the settings of Section 2. Let $\mathbf{u} \in \mathbb{R}^{n_0}$ be a fixed unit vector and $\mathbf{x} \in \mathbb{R}^{n_0}, \mathbf{x} \neq \mathbf{0}$ be any input into the network. Then, almost surely, with respect to the parameter initialization, $\|\mathbf{J}_{\mathcal{N}}(\mathbf{x})\mathbf{u}\|^2$ equals in distribution*

$$\frac{1}{n_0}\chi^2_{n_L}\prod_{l=1}^{L-1}\frac{c}{n_l}\sum_{i=1}^{n_l}\left(v_i\sqrt{1-\cos^2\gamma_{\mathbf{r}^{(l-1)},\mathbf{u}^{(l-1)}}} + \Xi_{l,i}(N(0,1),K)\cos\gamma_{\mathbf{r}^{(l-1)},\mathbf{u}^{(l-1)}}\right)^2,$$

*where $v_i$ and $\Xi_{l,i}(N(0,1),K)$ are independent, $v_i \sim N(0,1)$, $\Xi_{l,i}(N(0,1),K)$ is the largest order statistic in a sample of $K$ standard Gaussian random variables. Here $\gamma_{\mathbf{r}^{(l)},\mathbf{u}^{(l)}}$ denotes the*

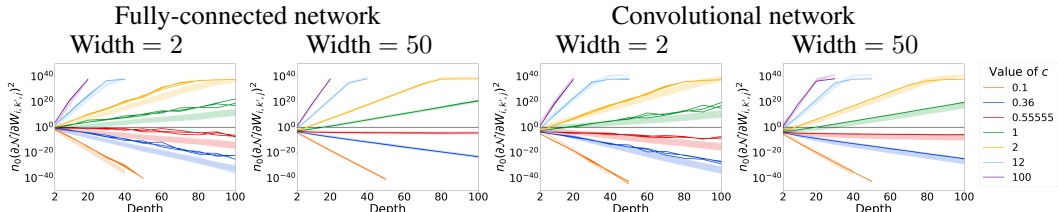

Figure 2: Expected value and interquartile range of the squared gradients $n_0(\partial \mathcal{N}/\partial W_{i,k',j})^2$ as a function of depth. Weights are sampled from $N(0, c/\text{fan-in})$ in fully-connected networks and $N(0, c/(k^2 \cdot \text{fan-in}))$, where $k$ is the kernel size, in CNNs. Biases are zero, and the maxout rank $K$ is 5. The gradient is stable in wide fully-connected and convolutional networks with $c = 0.55555$ (red line), the value suggested in Section 4. The dark and light blue lines represent the bounds from Corollary 2, and equal $1/\mathcal{L} = 0.36$ and $1/\mathcal{S} = 12$. The yellow line corresponds to the ReLU-He initialization. We compute the mean and quartiles from 100 network initializations and a fixed input. The same color lines that are close to each other correspond to 3 different unit-norm network inputs.

*angle between* $\mathfrak{r}^{(l)} := (x_1^{(l)}, \ldots, x_{n_l}^{(l)}, 1)$ *and* $\mathbf{u}^{(l)} := (\mathbf{u}_1^{(l)}, \ldots, \mathbf{u}_{n_l}^{(l)}, 0)$ *in* $\mathbb{R}^{n_l+1}$, *where* $\mathbf{u}^{(l)} = \overline{W}^{(l)}\mathbf{u}^{(l-1)}/\|\overline{W}^{(l)}\mathbf{u}^{(l-1)}\|$ *when* $\overline{W}^{(l)}\mathbf{u}^{(l-1)} \neq 0$ *and* $0$ *otherwise, and* $\mathbf{u}^{(0)} = \mathbf{u}$. *The matrices* $\overline{W}^{(l)}$ *consist of rows* $\overline{W}_i^{(l)} = W_{i,k'}^{(l)} \in \mathbb{R}^{n_{l-1}}$, *where* $k' = \text{argmax}_{k \in [K]}\{W_{i,k}^{(l)}\mathbf{x}^{(l-1)} + b_{i,k}^{(l)}\}$.

This statement is proved in Appendix D. The main strategy is to construct an orthonormal basis $\mathbf{B} = (\mathbf{b}_1, \ldots, \mathbf{b}_{n_l})$, where $\mathbf{b}_1 := \mathfrak{r}^{(l)}/\|\mathfrak{r}^{(l)}\|$, which allows us to express the layer gradient depending on the angle between $\mathfrak{r}^{(l)}$ and $\mathbf{u}^{(l)}$.

**Remark 4** (Wide networks). By Theorem 3, in a maxout network the distribution of $\|\mathbf{J}_{\mathcal{N}}(\mathbf{x})\mathbf{u}\|^2$ depends on the $\cos \gamma_{\mathfrak{r}^{(l-1)}, \mathbf{u}^{(l-1)}}$, which changes as the network gets wider or deeper. Since independent and isotropic random vectors in high-dimensional spaces tend to be almost orthogonal, we expect that the cosine will be close to zero for the earlier layers of wide networks, and individual units will behave similarly to squared standard Gaussians. In wide and deep networks, if the network parameters are sampled from $N(0, c/n_{l-1})$, $c = 1/\mathcal{M}$, and $K \geq 3$, we expect that $|\cos \gamma_{\mathfrak{r}^{(l)}, \mathbf{u}^{(l)}}| \approx 1$ for the later layers of deep networks and individual units will behave more as the squared largest order statistics. Here $\mathcal{M}$ is the second moment of the largest order statistic in a sample of size $K$ of standard Gaussian random variables. Based on this, for deep and wide networks, we can expect that

$$\mathbb{E}[\|\mathbf{J}_{\mathcal{N}}(\mathbf{x})\mathbf{u}\|^2] \approx \frac{n_L}{n_0}(c\mathcal{M})^{L-1} = \frac{n_L}{n_0}. \tag{2}$$

This intuition is discussed in more detail in Appendix D. According to (2), we expect that the expected gradient magnitude will be stable with depth when an appropriate initialization is used. See Figure 2 for a numerical evaluation of the effects of the width and depth on the gradients.

### 3.3 ACTIVATION LENGTH

To have a full picture of the derivatives in (1), we consider the activation length. The full version and proof of Corollary 5 are in Appendix E. The proof is based on Theorem 3, replacing $\mathbf{u}$ with $\mathfrak{r}/\|\mathfrak{r}\|$.

**Corollary 5** (Moments of the normalized activation length). *Consider a maxout network with the settings of Section 2. Let* $\mathbf{x} \in \mathbb{R}^{n_0}$ *be any input into the network. Then, for the moments of the normalized activation length* $A^{(l')}$ *of the* $l'$*th layer we have*

*Mean:*
$$\mathbb{E}\left[A^{(l')}\right] = \|\mathfrak{r}^{(0)}\|^2 \frac{1}{n_0}(c\mathcal{M})^{l'} + \sum_{j=2}^{l'}\left(\frac{1}{n_{j-1}}(c\mathcal{M})^{l'-j+1}\right),$$

*Moments of order* $t \geq 2$:
$$G_1\left((c\mathcal{M})^{tl'}\right) \leq \mathbb{E}\left[\left(A^{(l')}\right)^t\right] \leq G_2\left((cK)^{tl'}\exp\left\{\sum_{l=1}^{l'}\frac{t^2}{n_l K}\right\}\right).$$

Table 1: Recommended values for the constant $c$ for different maxout ranks $K$ based on Section 4.

| $K$ | 2 | 3 | 4 | 5 | 6 | 7 | 8 | 9 | 10 |
|---|---|---|---|---|---|---|---|---|---|
| $c$ | 1 | 0.78391 | 0.64461 | 0.55555 | 0.49462 | 0.45039 | 0.41675 | 0.39023 | 0.36872 |

*The expectation is taken with respect to the distribution of the network weights and biases, and $\mathcal{M}$ is a constant depending on K that can be computed approximately, see Table 4 for the values for $K = 2, \ldots, 10$. See Appendix E for the variance bounds and details on functions $G_1, G_2$.*

We could obtain an exact expression for the mean activation length for a finitely wide maxout network since its distribution only depends on the norm of the input, while this is not the case for the input-output Jacobian (Sections 3.1 and 3.2). We observe that the variance and the $t$th moments, $t \geq 2$, have an exponential dependence on the network architecture, including the maxout rank, whereas the mean does not, similarly to the input-output Jacobian (Corollary 2). Such behavior also occurs for ReLU networks (Hanin & Rolnick, 2018). See Figure 8 in Appendix E for an evaluation of the result.

## 4 IMPLICATIONS TO INITIALIZATION AND NETWORK ARCHITECTURE

We now aim to find initialization approaches and architectures that can avoid exploding and vanishing gradients. We take the annealed exploding and vanishing gradients definition from Hanin (2018) as a starting point for such investigation for maxout networks. Formally, we require

$$\mathbb{E}\left[\left(\frac{\partial \mathcal{L}(\mathbf{x})}{\partial W_{i,k',j}^{(l)}}\right)^2\right] = \Theta(1), \; \operatorname{Var}\left[\left(\frac{\partial \mathcal{L}(\mathbf{x})}{\partial W_{i,k',j}^{(l)}}\right)^2\right] = \Theta(1), \; \sup_{l \geq 1} \mathbb{E}\left[\left(\frac{\partial \mathcal{L}(\mathbf{x})}{\partial W_{i,k',j}^{(l)}}\right)^{2t}\right] < \infty, \forall t \geq 3,$$

where the expectation is with respect to the weights and biases. Based on (1) these conditions can be attained by ensuring that similar conditions hold for $\|\mathbf{J}_{\mathcal{N}}(\mathbf{x})\mathbf{u}\|^2$ and $A^{(l)}$.

**Initialization recommendations** Based on Corollary 2, the mean of $\|\mathbf{J}_{\mathcal{N}}(\mathbf{x})\mathbf{u}\|^2$ can be stabilized for some $c \in [1/\mathcal{L}, 1/\mathcal{S}]$. However, Theorem 3 shows that $\|\mathbf{J}_{\mathcal{N}}(\mathbf{x})\mathbf{u}\|^2$ depends on the input into the network. Hence, we expect that there is no value of $c$ stabilizing input-output Jacobian for every input simultaneously. Nevertheless, based on Remark 4, for wide and deep maxout networks, $\mathbb{E}[\|\mathbf{J}_{\mathcal{N}}(\mathbf{x})\mathbf{u}\|^2] \approx n_L/n_0$ if $c = 1/\mathcal{M}$, and the mean becomes stable. While Remark 4 does not include maxout rank $K = 2$, the same recommendation can be obtained for it using the approach from He et al. (2015), see Sun et al. (2018). Moreover, according to Corollary 5, the mean of the normalized activation length remains stable for different network depths if $c = 1/\mathcal{M}$. Hence, we recommend $c = 1/\mathcal{M}$ as an appropriate value for initialization. See Table 1 for the numerical value of $c$ for $K = 2, \ldots, 10$. We call this type of initialization, when the parameters are sampled from $N(0, c/\text{fan-in})$, $c = 1/\mathcal{M}$, "maxout initialization". We note that this matches the previous recommendation from Tseran & Montúfar (2021), which we now derived rigorously.

**Architecture recommendations** In Corollaries 2 and 5 the upper bound on the moments $t \geq 2$ of $\|\mathbf{J}_{\mathcal{N}}(\mathbf{x})\mathbf{u}\|^2$ and $A^{(l)} = \|\mathbf{x}^{(l)}\|^2/n_l$ can grow exponentially with the depth depending on the values of $(cK)^L$ and $\sum_{l=1}^{L-1} 1/(n_l K)$. Hence, we recommend choosing the widths such that $\sum_{l=1}^{L-1} 1/(n_l K) \leq 1$, which holds, e.g., if $n_l \geq L/K, \forall l = 1, \ldots, L-1$, and choosing a moderate value of the maxout rank $K$. However, the upper bound can still tend to infinity for the high-order moments. From Remark 4, it follows that for $K \geq 3$ to have a stable initialization independent of the network input, a maxout network has to be deep and wide. Experimentally, we observe that for 100-neuron wide networks with $K = 3$, the absolute value of the cosine that determines the initialization stability converges to 1 at around 60 layers, and for $K = 4, 5$, at around 30 layers. See Figure 5 in Appendix D. To sum up, we recommend working with deep and wide maxout networks with widths satisfying $\sum_{l=1}^{L-1} 1/(n_l K) \leq 1$, and choosing the maxout-rank not too small nor too large, e.g., $K = 5$.

## 5 IMPLICATIONS TO EXPRESSIVITY AND NTK

With Theorems 1 and 3 in place, we can now obtain maxout versions of the several types of results that previously have been derived only for ReLU networks.

## 5.1 EXPECTED NUMBER OF LINEAR REGIONS OF MAXOUT NETWORKS

For a piece-wise linear function $f\colon \mathbb{R}^{n_0} \to \mathbb{R}$, a linear region is defined as a maximal connected subset of $\mathbb{R}^{n_0}$ on which $f$ has a constant gradient. Tseran & Montúfar (2021) and Hanin & Rolnick (2019b) established upper bounds on the expected number of linear regions of maxout and ReLU networks, respectively. One of the key factors controlling these bounds is $C_{\mathrm{grad}}$, which is any upper bound on $(\sup_{\mathbf{x}\in\mathbb{R}^{n_0}} \mathbb{E}[\|\nabla\zeta_{z,k}(\mathbf{x})\|^t])^{1/t}$, for any $t \in \mathbb{N}$ and $z = 1, \ldots, N$. Here $\zeta_{z,k}$ is the $k$th pre-activation feature of the $z$th unit in the network, $N$ is the total number of units, and the gradient is with respect to the network input. Using Corollary 2, we obtain a value for $C_{\mathrm{grad}}$ for maxout networks, which remained an open problem in the work of Tseran & Montúfar (2021). The proof of Corollary 6 and the resulting refined bound on the expected number of linear regions are in Appendix F.

**Corollary 6** (Value for $C_{\mathrm{grad}}$). *Consider a maxout network with the settings of Section 2. Assume that the biases are independent of the weights but otherwise initialized using any approach. Consider the pre-activation feature $\zeta_{z,k}$ of a unit $z = 1, \ldots, N$. Then, for any $t \in \mathbb{N}$,*

$$\left( \sup_{\mathbf{x}\in\mathbb{R}^{n_0}} \mathbb{E}\left[ \|\nabla\zeta_{z,k}(x)\|^t \right] \right)^{\frac{1}{t}} \leq n_0^{-\frac{1}{2}} \max\left\{ 1, (cK)^{\frac{L-1}{2}} \right\} \exp\left\{ \frac{t}{2} \left( \sum_{l=1}^{L-1} \frac{1}{n_l K} + 1 \right) \right\}.$$

The value of $C_{\mathrm{grad}}$ given in Corollary 6 grows as $O((cK)^{L-1}\exp\{t\sum_{l=1}^{L-1} 1/(n_l K)\})$. The first factor grows exponentially with the network depth if $cK > 1$. This is the case when the network is initialized as in Section 4. However, since $K$ is usually a small constant and $c \leq 1$, $cK \geq 1$ is a small constant. The second factor grows exponentially with the depth if $\sum_{l=1}^{L-1} 1/(n_l K) \geq 1$. Hence, the exponential growth can be avoided if $n_l \geq (L-1)/K, \forall l = 1, \ldots, L-1$.

## 5.2 EXPECTED CURVE LENGTH DISTORTION

Let $M$ be a smooth 1-dimensional curve in $\mathbb{R}^{n_0}$ of length $\mathrm{len}(M)$ and $\mathcal{N}(M) \subseteq \mathbb{R}^{n_L}$ the image of $M$ under the map $\mathbf{x} \mapsto \mathcal{N}(\mathbf{x})$. We are interested in the length distortion of $M$, defined as $\mathrm{len}(\mathcal{N}(M))/\mathrm{len}(M)$. Using the results from Section 3.1, observing that the input-output Jacobian of maxout networks is well defined almost everywhere, and following Hanin et al. (2021), we obtain the following corollary. The proof is in Appendix G.

**Corollary 7** (Expected curve length distortion). *Consider a maxout network with the settings of Section 2. Assume that the biases are independent of the weights but otherwise initialized using any approach. Let $M$ be a smooth $1$-dimensional curve of unit length in $\mathbb{R}^{n_0}$. Then, the following upper bounds on the moments of $\mathrm{len}(\mathcal{N}(M))$ hold:*

$$\mathbb{E}\left[\mathrm{len}(\mathcal{N}(M))\right] \leq \left(\frac{n_L}{n_0}\right)^{\frac{1}{2}} (c\mathcal{L})^{\frac{L-1}{2}}, \qquad \mathrm{Var}\left[\mathrm{len}(\mathcal{N}(M))\right] \leq \frac{n_L}{n_0}(c\mathcal{L})^{L-1},$$

$$\mathbb{E}\left[\mathrm{len}(\mathcal{N}(M))^t\right] \leq \left(\frac{n_L}{n_0}\right)^{\frac{t}{2}} (cK)^{\frac{t(L-1)}{2}} \exp\left\{ \frac{t^2}{2} \left( \sum_{l=1}^{L-1} \frac{1}{n_l K} + \frac{1}{n_L} \right) \right\},$$

*where $\mathcal{L}$ is a constant depending on $K$, see Table 4 in Appendix C for values for $K = 2, \ldots, 10$.*

**Remark 8** (Expected curve length distortion in wide maxout networks). If the network is initialized according to Section 4, using Remark 4 and repeating the steps of the proof of Corollary 7, we get $\mathbb{E}\left[\mathrm{len}(\mathcal{N}(M))\right] \lesssim (n_L/n_0)^{1/2}$ and $\mathrm{Var}\left[\mathrm{len}(\mathcal{N}(M))\right] \approx n_L/n_0$.

Hence, similarly to ReLU networks, wide maxout networks, if initialized to keep the gradients stable, have low expected curve length distortion at initialization. However, we cannot conclude whether the curve length shrinks. For narrow networks, the upper bound does not exclude the possibility that the expected distortion grows exponentially with the network depth, depending on the initialization.

## 5.3 ON-DIAGONAL NTK

We denote the on-diagonal NTK with $K_{\mathcal{N}}(\mathbf{x}, \mathbf{x}) = \sum_i (\partial\mathcal{N}(\mathbf{x})/\partial\theta_i)^2$. In Appendix H we show:

**Corollary 9** (On-diagonal NTK). *Consider a maxout network with the settings of Section 2. Assume that $n_L = 1$ and that the biases are initialized to zero and are not trained. Assume that $8 \leq c \leq \mathcal{L}$,*

Table 2: Accuracy on the test set for networks trained using SGD with Nesterov momentum. Observe that maxout networks initialized with the maxout or max-pooling initialization perform significantly better than the ones initialized with other initializations and better or comparably to ReLU networks.

| | | MAXOUT | | | RELU |
|---|---|---|---|---|---|
| **VALUE OF c** | Small value | Max-pooling init Section 6 (Ours) | Maxout init Section 4 (Ours) | Naive ReLU He | He init |
| | 0.1 | 0.55 & 0.27 | 0.55555 | 2 | 2 |
| | | | FULLY-CONNECTED | | |
| MNIST | $11.35^{\pm 0.00}$ | — | $\mathbf{97.8^{\pm 0.15}}$ | $53.22^{\pm 24.08}$ | $97.43^{\pm 0.06}$ |
| Iris | $30.00^{\pm 0.00}$ | — | $\mathbf{91.67^{\pm 3.73}}$ | $82.5^{\pm 4.93}$ | $\mathbf{91.67^{\pm 3.73}}$ |
| | | | CONVOLUTIONAL | | |
| MNIST | $11.35^{\pm 0.00}$ | $99.58^{\pm 0.03}$ | $\mathbf{99.59^{\pm 0.04}}$ | $98.02^{\pm 0.21}$ | $99.49^{\pm 0.04}$ |
| CIFAR-10 | $10.00^{\pm 0.00}$ | $\mathbf{91.7^{\pm 0.17}}$ | $91.21^{\pm 0.13}$ | $44.84^{\pm 0.69}$ | $90.12^{\pm 0.25}$ |
| CIFAR-100 | $1.00^{\pm 0.00}$ | $65.33^{\pm 0.27}$ | $\mathbf{65.39^{\pm 0.39}}$ | $12.02^{\pm 0.8}$ | $59.59^{\pm 0.82}$ |
| Fashion MNIST | $10.00^{\pm 0.00}$ | $\mathbf{93.55^{\pm 0.13}}$ | $93.49^{\pm 0.13}$ | $81.56^{\pm 0.15}$ | $93.28^{\pm 0.11}$ |
| SVHN | $19.59^{\pm 0.00}$ | $97.3^{\pm 0.04}$ | $\mathbf{97.78^{\pm 0.02}}$ | $50.97^{\pm 1.71}$ | $96.74^{\pm 0.03}$ |

*where the constants $\mathcal{S}, \mathcal{L}$ are as specified in Table 4. Then,*

$$\|\mathbf{r}^{(0)}\|^2 \frac{(c\mathcal{S})^{L-2}}{n_0} P \leq \mathbb{E}[K_{\mathcal{N}}(\mathbf{x}, \mathbf{x})] \leq \|\mathbf{r}^{(0)}\|^2 \frac{(c\mathcal{L})^{L-2}\mathcal{M}^{L-1}}{n_0} P,$$

$$\mathbb{E}[K_{\mathcal{N}}(\mathbf{x}, \mathbf{x})^2] \leq 2 P P_W (cK)^{2(L-2)} \frac{\|\mathbf{r}^{(0)}\|^4}{n_0^2} \exp\left\{\sum_{j=1}^{L-1} \frac{4}{n_j K} + 4\right\},$$

*where $P = \sum_{l=0}^{L-1} n_l$, $P_W = \sum_{l=0}^{L} n_l n_{l-1}$, and $\mathcal{M}$ is as specified in Table 4.*

By Corollary 9, $\mathbb{E}[K_{\mathcal{N}}(\mathbf{x}, \mathbf{x})^2]/(\mathbb{E}[K_{\mathcal{N}}(\mathbf{x}, \mathbf{x})])^2$ is in $O((P_W/P)C^L \exp\{\sum_{l=1}^{L} 1/(n_l K)\})$, where $C$ depends on $\mathcal{L}, \mathcal{M}$ and $K$. Hence, if widths $n_1, \ldots, n_{L-1}$ and depth $L$ tend to infinity, this upper bound does not converge to a constant, suggesting that the NTK might not converge to a constant in probability. This is in line with previous results for ReLU networks by Hanin & Nica (2020a).

## 6 EXPERIMENTS

We check how the initialization proposed in Section 4 affects the network training. This initialization was first proposed heuristically by Tseran & Montúfar (2021), where it was tested for 10-layer fully-connected networks with an MNIST experiment. We consider both fully-connected and convolutional neural networks and run experiments for MNIST (LeCun & Cortes, 2010), Iris (Fisher, 1936), Fashion MNIST (Xiao et al., 2017), SVHN (Netzer et al., 2011), CIFAR-10 and CIFAR-100 (Krizhevsky et al., 2009). Fully connected networks have 21 layers and CNNs have a VGG-19-like architecture (Simonyan & Zisserman, 2015) with 20 or 16 layers depending on the input size, all with maxout rank 5. Weights are sampled from $N(0, c/\text{fan-in})$ in fully-connected networks and $N(0, c/(k^2 \cdot \text{fan-in}))$ in CNNs of kernel size $k$. The biases are initialized to zero. We report the mean and std of 4 runs.

We use plain deep networks without any kind of modifications or pre-training. We do not use normalization techniques, such as batch normalization (Ioffe & Szegedy, 2015), since this would obscure the effects of the initialization. Because of this, our results are not necessarily state-of-the-art. More details on the experiments are given in Appendix I, and the implementation is made available at https://anonymous.4open.science/r/maxout_expected_gradient-68BD.

**Max-pooling initialization** To account for the maximum in max-pooling layers, a maxout layer appearing after a max-pooling layer is initialized as if its maxout rank was $K \times m^2$, where $m^2$ is the max-pooling window size. For example, we used $K = 5$ and $m^2 = 4$, resulting in $c = 0.26573$ for such maxout layers. All other layers are initialized according to Section 4. We observe that max-pooling initialization often leads to slightly higher accuracy.

**Results for SGD with momentum** Table 2 reports test accuracy for networks trained using SGD with Nesterov momentum. We compare ReLU and maxout networks with different initializations: maxout, max-pooling, small value $c = 0.1$, and He $c = 2$. We observe that maxout and max-pooling initializations allow training deep maxout networks and obtaining better accuracy than ReLU

Table 3: Accuracy on the test set for the networks trained with Adam. Observe that maxout networks initialized with the maxout or max-pooling initialization perform better or comparably to ReLU networks, while maxout networks initialized with ReLU-He converge slower and perform worse.

| VALUE OF $c$ | | MAXOUT | | | RELU |
|---|---|---|---|---|---|
| | | Max-pooling init Section 6 (Ours) 0.55 & 0.27 | Maxout init Section 4 (Ours) 0.55555 | Naive ReLU He 2 | He init 2 |
| | | FULLY-CONNECTED | | | |
| MNIST | 1/10 epochs | — | $\mathbf{97.56^{\pm 0.18}}$ | $97.40^{\pm 0.30}$ | $96.72^{\pm 0.64}$ |
| | 2/10 epochs | — | $\mathbf{98.10^{\pm 0.09}}$ | $97.97^{\pm 0.12}$ | $97.54^{\pm 0.16}$ |
| | All epochs | — | $98.12^{\pm 0.10}$ | $\mathbf{98.13^{\pm 0.09}}$ | $97.37^{\pm 0.08}$ |
| | | CONVOLUTIONAL | | | |
| MNIST | 1/10 epochs | $99.06^{\pm 0.15}$ | $98.59^{\pm 0.58}$ | $98.54^{\pm 0.52}$ | $\mathbf{99.14^{\pm 0.32}}$ |
| | 2/10 epochs | $99.39^{\pm 0.13}$ | $98.51^{\pm 0.25}$ | $99.17^{\pm 0.13}$ | $\mathbf{99.41^{\pm 0.05}}$ |
| | All epochs | $\mathbf{99.53^{\pm 0.04}}$ | $99.47^{\pm 0.07}$ | $99.47^{\pm 0.04}$ | $99.45^{\pm 0.06}$ |
| Fashion MNIST | 1/10 epochs | $92.04^{\pm 0.29}$ | $92.35^{\pm 0.12}$ | $87.95^{\pm 0.33}$ | $\mathbf{92.45^{\pm 0.41}}$ |
| | 2/10 epochs | $92.61^{\pm 0.22}$ | $\mathbf{92.85^{\pm 0.21}}$ | $90.35^{\pm 0.38}$ | $92.71^{\pm 0.25}$ |
| | All epochs | $\mathbf{93.57^{\pm 0.17}}$ | $93.45^{\pm 0.10}$ | $91.63^{\pm 0.36}$ | $92.98^{\pm 0.13}$ |
| CIFAR-10 | 1/10 epochs | $\mathbf{88.25^{\pm 0.49}}$ | $87.31^{\pm 0.51}$ | $74.37^{\pm 0.37}$ | $85.95^{\pm 0.30}$ |
| | 2/10 epochs | $\mathbf{88.79^{\pm 0.72}}$ | $87.96^{\pm 0.75}$ | $81.94^{\pm 0.34}$ | $87.12^{\pm 0.23}$ |
| | All epochs | $\mathbf{91.33^{\pm 0.31}}$ | $91.06^{\pm 0.22}$ | $85.23^{\pm 0.20}$ | $87.70^{\pm 0.10}$ |
| CIFAR-100 | 1/10 epochs | $50.30^{\pm 3.34}$ | $\mathbf{53.43^{\pm 1.08}}$ | $19.22^{\pm 0.51}$ | $50.39^{\pm 0.91}$ |
| | 2/10 epochs | $57.54^{\pm 1.64}$ | $\mathbf{57.65^{\pm 0.75}}$ | $33.21^{\pm 0.51}$ | $51.34^{\pm 0.51}$ |
| | All epochs | $\mathbf{65.33^{\pm 1.26}}$ | $61.96^{\pm 0.58}$ | $37.58^{\pm 0.23}$ | $52.95^{\pm 0.30}$ |

networks, whereas performance is significantly worse or training does not progress for maxout networks with other initializations.

**Results for Adam**  Table 3 reports test accuracy for networks trained using Adam (Kingma & Ba, 2015). We compare ReLU and maxout networks with the following initializations: maxout, max-pooling, and He $c = 2$. We observe that, compared to He initialization, maxout and max-pooling initializations lead to faster convergence and better test accuracy. Compared to ReLU networks, maxout networks have better or comparable accuracy if maxout or max-pooling initialization is used.

## 7 DISCUSSION

We study the gradients of maxout networks with respect to the parameters and the inputs by analyzing a directional derivative of the input-output map. We observe that the distribution of the input-output Jacobian of maxout networks depends on the network input (in contrast to ReLU networks), which can complicate the stable initialization of maxout networks. Based on bounds on the moments, we derive an initialization that provably avoids vanishing and exploding gradients in wide networks. Experimentally, we show that, compared to other initializations, the suggested approach leads to better performance for fully connected and convolutional deep networks of standard width trained with SGD or Adam and better or similar performance compared to ReLU networks. Additionally, we refine previous upper bounds on the expected number of linear regions. We also derive results for the expected curve length distortion, observing that it does not grow exponentially with the depth in wide networks. Furthermore, we obtain bounds on the maxout NTK, suggesting that it might not converge to a constant when both the width and depth are large. These contributions enhance the applicability of maxout networks and add to the theoretical exploration of activation functions beyond ReLU.

**Limitations**  Even though our proposed initialization is in a sense optimal, our results are applicable only when the weights are sampled from $N(0, c/\text{fan-in})$ for some $c$. Further, we derived theoretical results only for fully-connected networks. Our experiments indicate that they also hold for CNNs: Figure 2 demonstrates that gradients behave according to the theory for fully connected and convolutional networks, and Tables 2 and 3 show improvement in CNNs performance under the initialization suggested in Section 4. However, we have yet to conduct the theoretical analysis of CNNs.

**Future work**  In future work, we would like to obtain more general results in settings involving multi-argument functions, such as aggregation functions in graph neural networks, and investigate the effects that initialization strategies stabilizing the initial gradients have at later stages of training.

**Reproducibility Statement** To ensure reproducibility, we make the code public on GitHub at https://anonymous.4open.science/r/maxout_expected_gradient-68BD, and Section 6 and Appendix I provide a detailed description of the experimental settings. Proofs are in Appendices B–H.

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

APPENDIX

The appendix is organized as follows.

## A    BASICS

### A.1    BASICS ON MAXOUT NETWORKS

As mentioned in the introduction, a rank-$K$ maxout unit computes the maximum of $K$ real-valued affine functions. Concretely, a rank-$K$ maxout unit with $n$ inputs implements a function

$$\mathbb{R}^n \to \mathbb{R}; \quad \mathbf{x} \mapsto \max_{k \in [K]}\{\langle W_k, \mathbf{x}\rangle + b_k\},$$

where $W_k \in \mathbb{R}^n$ and $b_k \in \mathbb{R}$, $k \in [K] := \{1, \ldots, K\}$, are trainable weights and biases. The $K$ arguments of the maximum are called the pre-activation features of the maxout unit. A rank-$K$ maxout unit can be regarded as a composition of an affine map with $K$ outputs and a maximum gate. A layer corresponds to parallel computation of several such units. For instance a layer with $n$ inputs and $m$ maxout units computes functions of the form

$$\mathbb{R}^n \to \mathbb{R}^m; \quad \mathbf{x} \mapsto \begin{bmatrix} \max_{k \in [K]}\{\langle W_{1,k}^{(1)}, \mathbf{x}\rangle + \mathbf{b}_{1,k}^{(1)}\} \\ \vdots \\ \max_{k \in [K]}\{\langle W_{m,k}^{(1)}, \mathbf{x}\rangle + \mathbf{b}_{m,k}^{(1)}\} \end{bmatrix},$$

where now $W_{i,k}^{(1)}$ and $\mathbf{b}_{i,k}^{(1)}$ are the weights and biases of the $k$th pre-activation feature of the $i$th maxout unit in the first layer. The situation is illustrated in Figure 3 for the case of a network with two inputs, one layer with two maxout units of rank three, and one output layer with a single output unit.

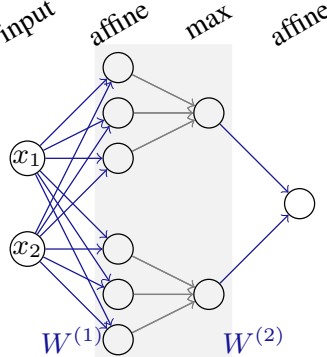

Figure 3: Illustration of a simple maxout network with two input units, one hidden layer consisting of two maxout units of rank 3, and an affine output layer with a single output unit.

## A.2 Basic notions of probability

We ought to remind several probability theory notions that we use to state our results. Firstly, recall that if $v_1, \ldots, v_k$ are independent, univariate standard normal random variables, then the sum of their squares, $\sum_{i=1}^{k} v_i^2$, is distributed according to the chi-squared distribution with $k$ degrees of freedom. We will denote such a random variable with $\chi_k^2$.

Secondly, the largest order statistic is a random variable defined as the maximum of a random sample, and the smallest order statistic is the minimum of a sample. And finally, a real valued random variable $X$ is said to be smaller than $Y$ in the stochastic order, denoted by $X \leq_{st} Y$, if $\Pr(X > x) \leq \Pr(Y > x)$ for all $x \in \mathbb{R}$. We will also denote with $\overset{d}{=}$ equality in distribution (meaning the cdfs are the same). With this, we start with the results for the squared norm of the input-output Jacobian $\|\mathbf{J}_{\mathcal{N}}(\mathbf{x})\mathbf{u}\|^2$.

## A.3 Details on the equation (1)

In (1) we are investigating magnitude of $\frac{\partial \mathcal{L}(\mathbf{x})}{\partial W_{i,k',j}^{(l)}}$. The reason we focus on the Jacobian norm rather than on $C$ is as follows. We have

$$
\begin{aligned}
\frac{\partial \mathcal{L}(\mathbf{x})}{\partial W_{i,k',j}^{(l)}} &= \langle \nabla_{\mathcal{N}} \mathcal{L}(\mathcal{N}(\mathbf{x})), \mathbf{J}_{\mathcal{N}}(W_{i,k',j}^{(l)}) \rangle \\
&= \langle \nabla_{\mathcal{N}} \mathcal{L}(\mathcal{N}(\mathbf{x})), \mathbf{J}_{\mathcal{N}}(\mathbf{x}_i^{(l)}) \rangle \mathbf{x}_j^{(l-1)} \\
&= \langle \nabla_{\mathcal{N}} \mathcal{L}(\mathcal{N}(\mathbf{x})), \mathbf{J}_{\mathcal{N}}(\mathbf{x}^{(l)})\mathbf{u} \rangle \mathbf{x}_j^{(l-1)}, \quad \mathbf{u} = \mathbf{e}_i \\
&= C(\mathbf{x}, W) \|\mathbf{J}_{\mathcal{N}}(\mathbf{x}^{(l)})\mathbf{u}\| \mathbf{x}_j^{(l-1)}
\end{aligned}
$$

Note that $C(\mathbf{x}, W) = \langle \nabla_{\mathcal{N}} \mathcal{L}(\mathcal{N}(\mathbf{x})), \mathbf{v} \rangle$ with $\mathbf{v} = \mathbf{J}_{\mathcal{N}}(\mathbf{x}^{(l)}) \mathbf{u} / \|\mathbf{J}_{\mathcal{N}}(\mathbf{x}^{(l)}) \mathbf{u}\|$, $\|\mathbf{v}\| = 1$. Hence $C(\mathbf{x}, W) \leq \|\nabla_{\mathcal{N}} \mathcal{L}(\mathcal{N}(\mathbf{x}))\| \|\mathbf{v}\| = \|\nabla_{\mathcal{N}} \mathcal{L}(\mathcal{N}(\mathbf{x}))\|$. The latter term does not directly depend on the specific parametrization nor the specific architecture of the network but only on the loss function and the prediction. In view of the description of $\frac{\partial \mathcal{L}(\mathbf{x})}{\partial W_{i,k',j}^{(l)}}$, the variance depends on the square of $\mathbf{x}_j^{(l-1)}$. Similarly, the variance of the gradient $\nabla_{W^{(l)}} \mathcal{L}(\mathbf{x}) = (\frac{\partial \mathcal{L}(\mathbf{x})}{\partial W_{i,k',j}^{(l)}})_j$ depends on $\mathbf{x}^{(l-1)} = (\mathbf{x}_j^{(l-1)})_j$ and thus depends on $\|\mathbf{x}^{(l-1)}\|^2$. This is how activation length appears in (1).

# B Bounds for the input-output Jacobian norm $\|\mathbf{J}_{\mathcal{N}}(\mathbf{x})\mathbf{u}\|^2$

## B.1 Preliminaries

We start by presenting several well-known results we will need for further discussion.

**Product of a Gaussian matrix and a unit vector**

**Lemma 10.** *Suppose $W$ is an $n \times n'$ matrix with i.i.d. Gaussian entries and $\mathbf{u}$ is a random unit vector in $\mathbb{R}^{n'}$ that is independent of $W$ but otherwise has any distribution. Then*

1. *$W\mathbf{u}$ is independent of $\mathbf{u}$ and is equal in distribution to $W\mathbf{v}$ where $\mathbf{v}$ is any fixed unit vector in $\mathbb{R}^{n'}$.*

2. *If the entries of $W$ are sampled i.i.d. from $N(\mu, \sigma^2)$, then for all $i = 1, \ldots, n$, $W_i \mathbf{u} \sim N(\mu, \sigma^2)$ and independent of $\mathbf{u}$.*

3. *If the entries of $W$ are sampled i.i.d. from $N(0, \sigma^2)$, then the squared $\ell_2$ norm $\|W\mathbf{u}\|^2 \overset{d}{=} \sigma^2 \chi_n^2$, where $\chi_n^2$ is a chi-squared random variable with $n$ degrees of freedom that is independent of $\mathbf{u}$.*

*Proof.* Statement 1 was proved in, e.g., Hanin et al. (2021, Lemma C.3) by considering directly the joint distribution of $W\mathbf{u}$ and $\mathbf{u}$.

Statement 2 follows from Statement 1 if we pick $\mathbf{v} = \mathbf{e}_1$.

To prove Statement 3, recall that by definition of the $\ell_2$ norm, $\|W\mathbf{u}\|^2 = \sum_{i=1}^{n}(W_i\mathbf{u})^2$. By Statement 2, for all $i = 1, \ldots, n$, $W_i\mathbf{u}$ are Gaussian random variables independent of $\mathbf{u}$ with mean zero and variance $\sigma^2$. Since any Gaussian random variable sampled from $N(\mu, \sigma^2)$ can be written as $\mu + \sigma v$, where $v \sim N(0, 1)$, we can write $\sum_{i=1}^{n}(W_i\mathbf{u})^2 = \sigma^2 \sum_{i=1}^{n} v_i^2$. By definition of the chi-squared distribution, $\sum_{i=1}^{n} v_i^2$ is a chi-squared random variable with $n$ degrees of freedom denoted with $\chi_n^2$, which leads to the desired result. $\qquad\square$

**Stochastic order**   We recall the definition of a stochastic order. A real-valued random variable $X$ is said to be smaller than $Y$ in the stochastic order, denoted by $X \leq_{st} Y$, if $\Pr(X > x) \leq \Pr(Y > x)$ for all $x \in \mathbb{R}$.

**Remark 11** (Stochastic ordering for functions)**.** Consider two functions $f : \mathbf{X} \to \mathbb{R}$ and $g : \mathbf{X} \to \mathbb{R}$ that satisfy $f(x) \leq g(x)$ for all $x \in \mathbf{X}$. Then, for a random variable $X$, $f(X) \leq_{st} g(X)$. To see this, observe that for any $y \in \mathbb{R}$, $\Pr(f(X) > y) = \Pr(X \in \{x\colon f(x) > y\})$ and $\Pr(g(X) > y) = \Pr(X \in \{x\colon g(x) > y\})$. Since $f(x) \leq g(x)$ for all $x \in \mathbf{X}$, $\{x\colon f(x) > y\} \subseteq \{x\colon g(x) > y\}$. Hence, $\Pr(f(X) > y) \leq \Pr(g(X) > y)$, and $f(X) \leq_{st} g(X)$.

**Remark 12** (Stochastic order and equality in distribution)**.** Consider real-valued random variables $X$, $Y$ and $\hat{Y}$. If $X \leq_{st} Y$ and $Y \stackrel{d}{=} \hat{Y}$, then $X \leq_{st} \hat{Y}$. Since $Y$ and $\hat{Y}$ have the same cdfs by definition of equality in distribution, for any $y \in \mathbb{R}$, $\Pr(X > y) \leq \Pr(Y > y) = \Pr(\hat{Y} > y)$.

### B.2   EXPRESSION FOR $\|\mathbf{J}_{\mathcal{N}}(\mathbf{x})\mathbf{u}\|^2$

Before proceeding to the proof of the main statement, given in Theorem 1, we present Proposition 13. Firstly, in Proposition 13 below, we prove an equality that holds almost surely for an input-output Jacobian under our assumptions. In this particular statement the reasoning closely follows Hanin et al. (2021, Proposition C.2). The modifications are due to the fact that a maxout network Jacobian is a product of matrices consisting of the rows of weights that are selected based on which pre-activation feature attains maximum, while in a ReLU network, the rows in these matrices are either the neuron weights or zeros.

**Proposition 13** (Equality for $\|\mathbf{J}_{\mathcal{N}}(\mathbf{x})\mathbf{u}\|^2$)**.** *Let $\mathcal{N}$ be a fully-connected feed-forward neural network with maxout units of rank $K$ and a linear last layer. Let the network have $L$ layers of widths $n_1, \ldots, n_L$ and $n_0$ inputs. Assume that the weights are continuous random variables (that have a density) and that the biases are independent of the weights but otherwise initialized using any approach. Let $\mathbf{u} \in \mathbb{R}^{n_0}$ be a fixed unit vector. Then, for any input into the network, $\mathbf{x} \in \mathbb{R}^{n_0}$, almost surely with respect to the parameter initialization the Jacobian with respect to the input satisfies*

$$\|\mathbf{J}_{\mathcal{N}}(\mathbf{x})\mathbf{u}\|^2 = \|W^{(L)}\mathbf{u}^{(L-1)}\|^2 \prod_{l=1}^{L-1} \sum_{i=1}^{n_l} \langle \overline{W}_i^{(l)}, \mathbf{u}^{(l-1)} \rangle^2, \tag{3}$$

*where vectors $\mathbf{u}^{(l)}$, $l = 1, \ldots, L-1$ are defined recursively as $\mathbf{u}^{(l)} = \overline{W}^{(l)}\mathbf{u}^{(l-1)}/\|\overline{W}^{(l)}\mathbf{u}^{(l-1)}\|$ when $\overline{W}^{(l)}\mathbf{u}^{(l-1)} \neq 0$ and $0$ otherwise, and $\mathbf{u}^{(0)} = \mathbf{u}$. The matrices $\overline{W}^{(l)}$ consist of rows $\overline{W}_i^{(l)} = W_{i,k'}^{(l)} \in \mathbb{R}^{n_{l-1}}$, $i = 1, \ldots, n_l$, where $k' = \operatorname{argmax}_{k \in [K]}\{W_{i,k}^{(l)}\mathbf{x}^{(l-1)} + b_{i,k}^{(l)}\}$, $\mathbf{x}^{(l)}$ is the output of the $l$th layer, and $\mathbf{x}^{(0)} = \mathbf{x}$.*

*Proof.* The Jacobian $\mathbf{J}_{\mathcal{N}}(\mathbf{x})$ of a network $\mathcal{N}(\mathbf{x})\colon \mathbb{R}^{n_0} \to \mathbb{R}^{n_L}$ can be written as a product of matrices $\overline{W}^{(l)}$, $l = 1, \ldots, L$, depending on the activation region of the input $\mathbf{x}$. The matrix $\overline{W}^{(l)}$ consists of rows $\overline{W}_i^{(l)} = W_{i,k'}^{(l)} \in \mathbb{R}^{n_{l-1}}$, where $k' = \operatorname{argmax}_{k \in [K]}\{W_{i,k}^{(l)}\mathbf{x}^{(l-1)} + b_{i,k}^{(l)}\}$ for $i = 1, \ldots, n_l$, and $\mathbf{x}^{(l-1)}$ is the $l$th layer's input. For the last layer, which is linear, we have $\overline{W}^{(L)} = W^{(L)}$. Thus,

$$\|\mathbf{J}_{\mathcal{N}}(\mathbf{x})\mathbf{u}\|^2 = \|W^{(L)}\overline{W}^{(L-1)} \cdots \overline{W}^{(1)}\mathbf{u}\|^2. \tag{4}$$

Further we denote $\mathbf{u}$ with $\mathbf{u}^{(0)}$ and assume $\|\overline{W}^{(1)}\mathbf{u}^{(0)}\| \neq 0$. To see that this holds almost surely, note that for a fixed unit vector $\mathbf{u}^{(0)}$, the probability of $\overline{W}^{(1)}$ being such that $\|\overline{W}^{(1)}\mathbf{u}^{(0)}\| = 0$ is 0. This is

indeed the case since to satisfy $\|\overline{W}^{(1)}\mathbf{u}^{(0)}\| = 0$, the weights must be a solution to a system of $n_1$ linear equations and this system is regular when $\mathbf{u} \neq 0$, so the solution set has positive co-dimension and hence zero measure. Multiplying and dividing (4) by $\|\overline{W}^{(1)}\mathbf{u}^{(0)}\|^2$,

$$
\|\mathbf{J}_\mathcal{N}(\mathbf{x})\mathbf{u}\|^2 = \left\| W^{(L)}\overline{W}^{(L-1)}\cdots\overline{W}^{(2)}\frac{\overline{W}^{(1)}\mathbf{u}^{(0)}}{\|\overline{W}^{(1)}\mathbf{u}^{(0)}\|} \right\|^2 \|\overline{W}^{(1)}\mathbf{u}^{(0)}\|^2
$$

$$
= \left\| W^{(L)}\overline{W}^{(L-1)}\cdots\overline{W}^{(2)}\mathbf{u}^{(1)} \right\|^2 \|\overline{W}^{(1)}\mathbf{u}^{(0)}\|^2,
$$

where $\mathbf{u}^{(1)} = \overline{W}^{(1)}\mathbf{u}^{(0)}/\|\overline{W}^{(1)}\mathbf{u}^{(0)}\|$. Repeating this procedure layer-by-layer, we get

$$
\|W^{(L)}\mathbf{u}^{(L-1)}\|^2 \|\overline{W}^{(L-1)}\mathbf{u}^{(L-2)}\|^2 \cdots \|\overline{W}^{(1)}\mathbf{u}^{(0)}\|^2. \tag{5}
$$

By definition of the $\ell_2$ norm, for any layer $l$, $\|\overline{W}^{(l)}\mathbf{u}^{(l-1)}\|^2 = \sum_{i=1}^{n_l}\langle\overline{W}_i^{(l)}, \mathbf{u}^{(l-1)}\rangle^2$. Substituting this into (5) we get the desired statement. $\square$

## B.3 STOCHASTIC ORDERING FOR $\|\mathbf{J}_\mathcal{N}(\mathbf{x})\mathbf{u}\|^2$

Now we prove the result for the stochastic ordering of the input-output Jacobian in a finite-width maxout network.

**Theorem 1** (Bounds on $\|\mathbf{J}_\mathcal{N}(\mathbf{x})\mathbf{u}\|^2$). *Consider a maxout network with the settings of Section 2. Assume that the biases are independent of the weights but otherwise initialized using any approach. Let $\mathbf{u} \in \mathbb{R}^{n_0}$ be a fixed unit vector. Then, almost surely with respect to the parameter initialization, for any input into the network $\mathbf{x} \in \mathbb{R}^{n_0}$, the following stochastic order bounds hold:*

$$
\frac{1}{n_0}\chi_{n_L}^2 \prod_{l=1}^{L-1}\frac{c}{n_l}\sum_{i=1}^{n_l}\xi_{l,i}(\chi_1^2, K) \leq_{st} \|\mathbf{J}_\mathcal{N}(\mathbf{x})\mathbf{u}\|^2 \leq_{st} \frac{1}{n_0}\chi_{n_L}^2 \prod_{l=1}^{L-1}\frac{c}{n_l}\sum_{i=1}^{n_l}\Xi_{l,i}(\chi_1^2, K),
$$

*where $\xi_{l,i}(\chi_1^2, K)$ and $\Xi_{l,i}(\chi_1^2, K)$ are respectively the smallest and largest order statistic in a sample of size $K$ of chi-squared random variables with 1 degree of freedom, independent of each other and of the vectors $\mathbf{u}$ and $\mathbf{x}$.*

*Proof.* From Proposition 13, we have the following equality

$$
\|\mathbf{J}_\mathcal{N}(\mathbf{x})\mathbf{u}\|^2 = \|W^{(L)}\mathbf{u}^{(L-1)}\|^2 \prod_{l=1}^{L-1}\sum_{i=1}^{n_l}\langle\overline{W}_i^{(l)}, \mathbf{u}^{(l-1)}\rangle^2, \tag{6}
$$

where vectors $\mathbf{u}^{(l)}$, $l = 0, \ldots, L-1$ are defined recursively as $\mathbf{u}^{(l)} = \overline{W}^{(l)}\mathbf{u}^{(l-1)}/\|\overline{W}^{(l)}\mathbf{u}^{(l-1)}\|$ and $\mathbf{u}^{(0)} = \mathbf{u}$. Matrices $\overline{W}^{(l)}$ consist of rows $\overline{W}_i^{(l)} = W_{i,k'}^{(l)} \in \mathbb{R}^{n_{l-1}}$, $i = 1, \ldots, n_l$, where $k' = \mathrm{argmax}_{k\in[K]}\{W_{i,k}^{(l)}\mathbf{x}^{(l-1)} + b_{i,k}^{(l)}\}$, and $\mathbf{x}^{(l-1)}$ is the $l$th layer's input, $\mathbf{x}^{(0)} = \mathbf{x}$.

We assumed that weights in the last layer are sampled from a Gaussian distribution with mean zero and variance $1/n_{L-1}$. Then, by Lemma 10 item 3, $\|W^{(L)}\mathbf{u}^{(L-1)}\|^2 \stackrel{d}{=} (1/n_{L-1})\chi_{n_L}^2$ and is independent of $\mathbf{u}^{(L-1)}$. In equation (6), using this observation and then multiplying and dividing the summands by $c/n_{l-1}$ and rearranging we obtain

$$
\|\mathbf{J}_\mathcal{N}(\mathbf{x})\mathbf{u}\|^2 \stackrel{d}{=} \frac{1}{n_{L-1}}\chi_{n_L}^2 \prod_{l=1}^{L-1}\frac{c}{n_{l-1}}\sum_{i=1}^{n_l}\left(\frac{n_{l-1}}{c}\langle\overline{W}_i^{(l)}, \mathbf{u}^{(l-1)}\rangle^2\right)
$$

$$
= \frac{1}{n_0}\chi_{n_L}^2 \prod_{l=1}^{L-1}\frac{c}{n_l}\sum_{i=1}^{n_l}\left(\frac{n_{l-1}}{c}\langle\overline{W}_i^{(l)}, \mathbf{u}^{(l-1)}\rangle^2\right).
$$

Now we focus on $\sqrt{n_{l-1}/c}\,\langle\overline{W}_i^{(l)}, \mathbf{u}^{(l-1)}\rangle$. Since we have assumed that the weights are sampled from a Gaussian distribution with zero mean and variance $c/n_{l-1}$, any weight $W_{i,k,j}^{(l)}$, $j = 1, \ldots, n_{l-1}$,

can be written as $\sqrt{c/n_{l-1}}v_{i,k,j}^{(l)}$, where $v_{i,k,j}^{(l)}$ is a standard Gaussian random variable. We also write $W_{i,k}^{(l)} = \sqrt{c/n_{l-1}}V_{i,k}^{(l)}$, where $V_{i,k}^{(l)}$ is an $n_{l-1}$-dimensional standard Gaussian random vector. Observe that for any $k' \in [K]$, $\langle W_{i,k'}^{(l)}, \mathbf{u}^{(l-1)}\rangle^2 \leq \max_{k\in[K]}\{\langle W_{i,k}^{(l)}, \mathbf{u}^{(l-1)}\rangle^2\}$ and $\langle W_{i,k'}^{(l)}, \mathbf{u}^{(l-1)}\rangle^2 \geq \min_{k\in[K]}\{\langle W_{i,k}^{(l)}, \mathbf{u}^{(l-1)}\rangle^2\}$. Therefore,

$$\frac{c}{n_{l-1}}\min_{k\in[K]}\left\{\left\langle V_{i,k}^{(l)}, \mathbf{u}^{(l-1)}\right\rangle^2\right\} \leq \langle \overline{W}_i^{(l)}, \mathbf{u}^{(l-1)}\rangle^2 \leq \frac{c}{n_{l-1}}\max_{k\in[K]}\left\{\left\langle V_{i,k}^{(l)}, \mathbf{u}^{(l-1)}\right\rangle^2\right\}.$$

Notice that vectors $\mathbf{u}^{(l-1)}$ are unit vectors by their definition. By Lemma 10, the inner product of a standard Gaussian vector and a unit vector is a standard Gaussian random variable independent of the given unit vector.

By definition, a squared standard Gaussian random variable is distributed as $\chi_1^2$, a chi-squared random variable with 1 degree of freedom. Hence, $\max_{k\in[K]}\{\langle V_{i,k}^{(l)}, \mathbf{u}^{(l-1)}\rangle^2\}$ is distributed as the largest order statistic in a sample of size $K$ of chi-squared random variables with 1 degree of freedom. We will denote such a random variable with $\Xi_{l,i}(\chi_1^2, K)$. Likewise, $\min_{k\in[K]}\{\langle V_{i,k}^{(l)}, \mathbf{u}^{(l-1)}\rangle^2\}$ is distributed as the smallest order statistic in a sample of size $K$ of chi-squared random variables with 1 degree of freedom, denoted with $\xi_{l,i}(\chi_1^2, K)$.

Combining results for each layer, we obtain the following bounds

$$\|\mathbf{J}_{\mathcal{N}}(\mathbf{x})\mathbf{u}\|^2 \leq \frac{1}{n_0}\chi_{n_L}^2 \prod_{l=1}^{L-1}\frac{c}{n_l}\sum_{i=1}^{n_l}\max_{k\in[K]}\left\{\left\langle V_{i,k}^{(l)}, \mathbf{u}^{(l-1)}\right\rangle^2\right\} \stackrel{d}{=} \frac{1}{n_0}\chi_{n_L}^2 \prod_{l=1}^{L-1}\frac{c}{n_l}\sum_{i=1}^{n_l}\Xi_{l,i}(\chi_1^2, K),$$

$$\|\mathbf{J}_{\mathcal{N}}(\mathbf{x})\mathbf{u}\|^2 \geq \frac{1}{n_0}\chi_{n_L}^2 \prod_{l=1}^{L-1}\frac{c}{n_l}\sum_{i=1}^{n_l}\min_{k\in[K]}\left\{\left\langle V_{i,k}^{(l)}, \mathbf{u}^{(l-1)}\right\rangle^2\right\} \stackrel{d}{=} \frac{1}{n_0}\chi_{n_L}^2 \prod_{l=1}^{L-1}\frac{c}{n_l}\sum_{i=1}^{n_l}\xi_{l,i}(\chi_1^2, K).$$

Then, by Remarks 11 and 12, the following stochastic ordering holds

$$\frac{1}{n_0}\chi_{n_L}^2 \prod_{l=1}^{L-1}\frac{c}{n_l}\sum_{i=1}^{n_l}\xi_{l,i}(\chi_1^2, K) \leq_{st} \|\mathbf{J}_{\mathcal{N}}(\mathbf{x})\mathbf{u}\|^2 \leq_{st} \frac{1}{n_0}\chi_{n_L}^2 \prod_{l=1}^{L-1}\frac{c}{n_l}\sum_{i=1}^{n_l}\Xi_{l,i}(\chi_1^2, K),$$

which concludes the proof. $\qquad\square$

## C  MOMENTS OF THE INPUT-OUTPUT JACOBIAN NORM $\|\mathbf{J}_{\mathcal{N}}(\mathbf{x})\mathbf{u}\|^2$

In the proof on the bounds of the moments we use an approach similar to Hanin et al. (2021) for upper bounding the moments of the chi-squared distribution.

**Corollary 2** (Bounds on the moments of $\|\mathbf{J}_{\mathcal{N}}(\mathbf{x})\mathbf{u}\|^2$). *Consider a maxout network with the settings of Section 2. Assume that the biases are independent of the weights but otherwise initialized using any approach. Let $\mathbf{u} \in \mathbb{R}^{n_0}$ be a fixed unit vector and $\mathbf{x} \in \mathbb{R}^{n_0}$ be any input into the network, Then*

*(i)* $\dfrac{n_L}{n_0}(c\mathcal{S})^{L-1} \leq \mathbb{E}[\|\mathbf{J}_{\mathcal{N}}(\mathbf{x})\mathbf{u}\|^2] \leq \dfrac{n_L}{n_0}(c\mathcal{L})^{L-1}$,

*(ii)* $\text{Var}\left[\|\mathbf{J}_{\mathcal{N}}(\mathbf{x})\mathbf{u}\|^2\right] \leq \left(\dfrac{n_L}{n_0}\right)^2 c^{2(L-1)}\left(K^{2(L-1)}\exp\left\{4\left(\sum_{l=1}^{L-1}\frac{1}{n_l K} + \frac{1}{n_L}\right)\right\} - \mathcal{S}^{2(L-1)}\right)$,

*(iii)* $\mathbb{E}\left[\|\mathbf{J}_{\mathcal{N}}(\mathbf{x})\mathbf{u}\|^{2t}\right] \leq \left(\dfrac{n_L}{n_0}\right)^t (cK)^{t(L-1)}\exp\left\{t^2\left(\sum_{l=1}^{L-1}\frac{1}{n_l K} + \frac{1}{n_L}\right)\right\}, \quad t \in \mathbb{N}$,

*where the expectation is taken with respect to the distribution of the network weights. The constants $\mathcal{S}$ and $\mathcal{L}$ depend on $K$ and denote the means of the smallest and the largest order statistic in a sample of $K$ chi-squared random variables. For $K = 2, \ldots, 10$, $\mathcal{S} \in [0.02, 0.4]$ and $\mathcal{L} \in [1.6, 4]$. See Table 4 in Appendix C for the exact values.*

*Proof.* We first prove results for the mean, then for the moments of order $t > 1$, and finish with the proof for the variance.

**Mean** Using mutual independence of the variables in the bounds in Theorem 1, and that if two non-negative univariate random variables $X$ and $Y$ are such that $X \leq_{st} Y$ then $\mathbb{E}[X^n] \leq \mathbb{E}[Y^n]$ for all $n \geq 1$ (Müller & Stoyan, 2002, Theorem 1.2.12),

$$\frac{1}{n_0}\mathbb{E}\left[\chi_{n_L}^2\right]\prod_{l=1}^{L-1}\frac{c}{n_l}\sum_{i=1}^{n_l}\mathbb{E}\left[\xi_{l,i}\right] \leq \mathbb{E}[\|\mathbf{J}_{\mathcal{N}}(\mathbf{x})\mathbf{u}\|^2] \leq \frac{1}{n_0}\mathbb{E}\left[\chi_{n_L}^2\right]\prod_{l=1}^{L-1}\frac{c}{n_l}\sum_{i=1}^{n_l}\mathbb{E}\left[\Xi_{l,i}\right].$$

where we used $\xi_{l,i}$ and $\Xi_{l,i}$ as shorthands for $\xi_{l,i}(\chi_1^2, K)$ and $\Xi_{l,i}(\chi_1^2, K)$. Using the formulas for the largest and the smallest order statistic pdfs from Remark 14, the largest order statistic mean equals

$$\mathbb{E}\left[\Xi_{l,i}\right] = \frac{K}{\sqrt{2\pi}}\int_0^\infty \left(\mathrm{erf}\left(\sqrt{\frac{x}{2}}\right)\right)^{K-1}x^{1/2}e^{-x/2}dx = \mathcal{L},$$

and the smallest order statistic mean equals

$$\mathbb{E}\left[\xi_{l,i}\right] = \frac{K}{\sqrt{2\pi}}\int_0^\infty \left(1-\mathrm{erf}\left(\sqrt{\frac{x}{2}}\right)\right)^{K-1}x^{1/2}e^{-x/2}dx = \mathcal{S}.$$

Here we denoted the right hand-sides with $\mathcal{L}$ and $\mathcal{S}$, which are constants depending on K, and can be computed exactly for $K = 2$ and $K = 3$, and approximately for higher $K$-s, see Table 4. It is known that $\mathbb{E}\left[\chi_{n_L}^2\right] = n_L$. Combining, we get

$$\frac{n_L}{n_0}(c\mathcal{S})^{L-1} \leq \mathbb{E}[\|\mathbf{J}_{\mathcal{N}}(\mathbf{x})\mathbf{u}\|^2] \leq \frac{n_L}{n_0}(c\mathcal{L})^{L-1}.$$

**Moments of order** $t > 1$ As above, using mutual independence of the variables in the bounds in Theorem 1, and that if two non-negative univariate random variables $X$ and $Y$ are such that $X \leq_{st} Y$ then $\mathbb{E}[X^n] \leq \mathbb{E}[Y^n]$ for all $n \geq 1$ (Müller & Stoyan, 2002, Theorem 1.2.12),

$$\mathbb{E}[\|\mathbf{J}_{\mathcal{N}}(\mathbf{x})\mathbf{u}\|^{2t}] \leq \mathbb{E}\left[\left(\frac{1}{n_0}\chi_{n_L}^2\prod_{l=1}^{L-1}\frac{c}{n_l}\sum_{i=1}^{n_l}\Xi_{l,i}\right)^t\right]$$

$$= \left(\frac{n_L}{n_0}\right)^t\left(\frac{1}{n_L}\right)^t\mathbb{E}\left[(\chi_{n_L}^2)^t\right]\prod_{l=1}^{L-1}\left(\frac{c}{n_l}\right)^t\mathbb{E}\left[\left(\sum_{i=1}^{n_l}\Xi_{l,i}\right)^t\right]. \tag{7}$$

Upper-bounding the maximum of chi-squared variables with a sum,

$$\left(\frac{c}{n_l}\right)^t\mathbb{E}\left[\left(\sum_{i=1}^{n_l}\Xi_{l,i}\right)^t\right] \leq \left(\frac{c}{n_l}\right)^t\mathbb{E}\left[\left(\sum_{i=1}^{n_l}\sum_{k=1}^{K}(\chi_1^2)_{l,i,k}\right)^t\right] = \left(\frac{c}{n_l}\right)^t\mathbb{E}\left[(\chi_{n_lK}^2)^t\right],$$

where we used that a sum of $n_lK$ chi-squared variables with one degree of freedom is a chi-squared variable with $n_lK$ degrees of freedom. Using the formula for noncentral moments of the chi-squared distribution and the inequality $1 + x \leq e^x$,

$$\left(\frac{c}{n_l}\right)^t\mathbb{E}\left[(\chi_{n_lK}^2)^t\right] = \left(\frac{c}{n_l}\right)^t(n_lK)(n_lK+2)\cdots(n_lK+2t-2)$$

$$= c^tK^t\cdot 1\cdot\left(1+\frac{2}{n_lK}\right)\cdots\left(1+\frac{2t-2}{n_lK}\right) \leq c^tK^t\exp\left\{\sum_{i=0}^{t-1}\frac{2i}{n_lK}\right\} \leq c^tK^t\exp\left\{\frac{t^2}{n_lK}\right\},$$

where we used the formula for calculating the sum of consecutive numbers $\sum_{i=1}^{t-1}i = t(t-1)/2$. Similarly,

$$\left(\frac{1}{n_L}\right)^t\mathbb{E}\left[(\chi_{n_L}^2)^t\right] \leq \exp\left\{\frac{t^2}{n_L}\right\}.$$

Combining, we upper bound (7) with

$$\left(\frac{n_L}{n_0}\right)^t(cK)^{t(L-1)}\exp\left\{t^2\left(\sum_{l=1}^{L-1}\frac{1}{n_lK}+\frac{1}{n_L}\right)\right\}.$$

**Variance** Combining the upper bound on the second moment and the lower bound on the mean, we get the following upper bound on the variance

$$\text{Var}\left[\|\mathbf{J}_{\mathcal{N}}(\mathbf{x})\mathbf{u}\|^2\right] \leq \left(\frac{n_L}{n_0}\right)^2 c^{2(L-1)} \left(K^{2(L-1)} \exp\left\{4\left(\sum_{l=1}^{L-1}\frac{1}{n_l K} + \frac{1}{n_L}\right)\right\} - 8^{2(L-1)}\right).$$

which concludes the proof. □

**Remark 14** (Computing the constants). Here we provide the derivations necessary to compute the constants equal to the moments of the largest and the smallest order statistics appearing in the results. Firstly, the cdf of the largest order statistic of independent univariate random variables $y_1, \ldots, y_K$ with cdf $F(x)$ and pdf $f(x)$ is

$$\text{Pr}\left(\max_{k\in[K]}\{y_k\} < x\right) = \text{Pr}\left(\bigcap_{k=1}^{K}(y_k < x)\right) = \prod_{k=1}^{K}\text{Pr}(y_k < x) = (F(x))^K.$$

Hence, the pdf is $K(F(x))^{K-1}f(x)$. For the smallest order statistic, the cdf is

$$\text{Pr}\left(\min_{k\in[K]}\{y_k\} < x\right) = 1 - \prod_{k=1}^{K}\text{Pr}(y_k \geq x) = 1 - (1 - F(x))^K.$$

Thus, the pdf is $K(1 - F(x))^{K-1}f(x)$.

Now we obtain pdfs for the distributions that are used in the results.

**Chi-squared distribution** The cdf of a chi-squared random variable $\chi_k^2$ with $k = 1$ degree of freedom is $F(x) = (\Gamma(k/2))^{-1}\gamma(k/2, x/2) = \text{erf}(\sqrt{x/2})$, and the pdf is $f(x) = (2^{k/2}\Gamma(k/2))^{-1}x^{k/2-1}e^{-x/2} = (2\pi)^{-1/2}x^{-1/2}e^{-x/2}$. Here we used that $\Gamma(1/2) = \sqrt{\pi}$ and $\gamma(1/2, x/2) = \sqrt{\pi}\,\text{erf}(\sqrt{x/2})$. Therefore, the pdf of the largest order statistic in a sample of $K$ chi-squared random variables with 1 degree of freedom $\Xi_{l,i}(\chi_1^2, K)$ is

$$K\left(\text{erf}\left(\sqrt{\frac{x}{2}}\right)\right)^{K-1}\frac{1}{\sqrt{2\pi}}x^{-\frac{1}{2}}e^{-\frac{x}{2}}.$$

The pdf of the smallest order statistic in a sample of $K$ chi-squared random variables with 1 degree of freedom $\xi_{l,i}(\chi_1^2, K)$ is

$$K\left(1 - \text{erf}\left(\sqrt{\frac{x}{2}}\right)\right)^{K-1}\frac{1}{\sqrt{2\pi}}x^{-\frac{1}{2}}e^{-\frac{x}{2}}.$$

**Standard Gaussian distribution** Recall that the cdf of a standard Gaussian random variable is $F(x) = 1/2(1 + \text{erf}(x/\sqrt{2}))$, and the pdf is $f(x) = 1/\sqrt{2\pi}\exp\{-x^2/2\}$. Then, for the pdf of the largest order statistic in a sample of $K$ standard Gaussian random variables $\Xi_{l,i}(N(0,1), K)$ we get

$$\frac{K}{2^{K-1}\sqrt{2\pi}}\left(1 + \text{erf}\left(\frac{x}{\sqrt{2}}\right)\right)^{K-1}e^{-\frac{x^2}{2}}.$$

**Constants** Now we obtain formulas for the constants. For the mean of the smallest order statistic in a sample of $K$ chi-squared random variables with 1 degree of freedom $\xi_{l,i}(\chi_1^2, K)$, we get

$$8 = \frac{K}{\sqrt{2\pi}}\int_0^\infty x^{\frac{1}{2}}\left(1 - \text{erf}\left(\sqrt{\frac{x}{2}}\right)\right)^{K-1}e^{-\frac{x}{2}}dx.$$

The mean of the largest order statistic in a sample of $K$ chi-squared random variables with 1 degree of freedom $\Xi_{l,i}(\chi_1^2, K)$ is

$$\mathcal{L} = \frac{K}{\sqrt{2\pi}}\int_0^\infty x^{\frac{1}{2}}\left(\text{erf}\left(\sqrt{\frac{x}{2}}\right)\right)^{K-1}e^{-\frac{x}{2}}dx.$$

The second moment of the largest order statistic in a sample of $K$ standard Gaussian random variables $\Xi_{l,i}(N(0,1),K)$ equals

$$\mathcal{M} = \frac{K}{2^{K-1}\sqrt{2\pi}} \int_{-\infty}^{\infty} x^2 \left(1 + \operatorname{erf}\left(\frac{x}{\sqrt{2}}\right)\right)^{K-1} e^{-\frac{x^2}{2}} dx.$$

These constants can be evaluated using numerical computation software. The values estimated for $K = 2, \ldots, 10$ using Mathematica (Wolfram Research, Inc, 2022) are in Table 4.

Table 4: Constants $\mathcal{L}$ and $\mathcal{S}$ denote the means of the largest and the smallest order statistics in a sample of size $K$ of chi-squared random variables with 1 degree of freedom. Constant $\mathcal{M}$ denotes the second moment of the largest order statistic in a sample of size $K$ of standard Gaussian random variables. See Remark 14 for the explanation of how these constants are computed.

| MAXOUT RANK | $\mathcal{L}$ | $\mathcal{S}$ | $\mathcal{M}$ |
|---|---|---|---|
| 2 | 1.63662 | 0.36338 | 1 |
| 3 | 2.10266 | 0.1928 | 1.27566 |
| 4 | 2.47021 | 0.1207 | 1.55133 |
| 5 | 2.77375 | 0.08308 | 1.80002 |
| 6 | 3.03236 | 0.06083 | 2.02174 |
| 7 | 3.25771 | 0.04655 | 2.2203 |
| 8 | 3.45743 | 0.0368 | 2.39954 |
| 9 | 3.63681 | 0.02984 | 2.56262 |
| 10 | 3.79962 | 0.0247 | 2.7121 |

## D  EQUALITY IN DISTRIBUTION FOR THE INPUT-OUTPUT JACOBIAN NORM AND WIDE NETWORK RESULTS

Here we prove results from Section 3.2. We will use the following theorem from Anderson (2003). We reference it here without proof, but remark that it is based on the well-known result that uncorrelated jointly Gaussian random variables are independent.

**Theorem 15** (Anderson 2003, Theorem 3.3.1). *Suppose $X_1, \ldots, X_N$ are independent, where $X_\alpha$ is distributed according to $N(\boldsymbol{\mu}_\alpha, \boldsymbol{\Sigma})$. Let $C = (c_{\alpha\beta})$ be an $N \times N$ orthogonal matrix. Then $Y_\alpha = \sum_{\beta=1}^N c_{\alpha\beta} X_\beta$ is distributed according to $N(\boldsymbol{\nu}_\alpha, \boldsymbol{\Sigma})$, where $\boldsymbol{\nu}_\alpha = \sum_{\beta=1}^N c_{\alpha\beta} \boldsymbol{\mu}_\beta$, $\alpha = 1, \ldots, N$, and $Y_1, \ldots, Y_N$ are independent.*

**Remark 16.** We will use Theorem 15 in the following way. Notice that it is possible to consider a vector $\mathbf{v}$ with entries sampled i.i.d. from $N(0, \sigma^2)$ in Theorem 15 and treat entries of $\mathbf{v}$ as a set of 1-dimensional vectors $X_1, \ldots, X_N$. Then we can obtain that products of the columns of the orthogonal matrix $C$ and the vector $\mathbf{v}$, $Y_\beta = \sum_{\alpha=1}^N c_{\alpha\beta} \mathbf{v}_\alpha$, are distributed according to $N(0, \sigma^2)$ and are mutually independent.

**Theorem 3** (Equality in distribution for $\|\mathbf{J}_\mathcal{N}(\mathbf{x})\mathbf{u}\|^2$). *Consider a maxout network with the settings of Section 2. Let $\mathbf{u} \in \mathbb{R}^{n_0}$ be a fixed unit vector and $\mathbf{x} \in \mathbb{R}^{n_0}, \mathbf{x} \neq \mathbf{0}$ be any input into the network. Then, almost surely, with respect to the parameter initialization, $\|\mathbf{J}_\mathcal{N}(\mathbf{x})\mathbf{u}\|^2$ equals in distribution*

$$\frac{1}{n_0} \chi_{n_L}^2 \prod_{l=1}^{L-1} \frac{c}{n_l} \sum_{i=1}^{n_l} \left(v_i \sqrt{1 - \cos^2 \gamma_{\mathbf{r}^{(l-1)}, \mathbf{u}^{(l-1)}}} + \Xi_{l,i}(N(0,1), K) \cos \gamma_{\mathbf{r}^{(l-1)}, \mathbf{u}^{(l-1)}}\right)^2,$$

*where $v_i$ and $\Xi_{l,i}(N(0,1), K)$ are independent, $v_i \sim N(0,1)$, $\Xi_{l,i}(N(0,1), K)$ is the largest order statistic in a sample of $K$ standard Gaussian random variables. Here $\gamma_{\mathbf{r}^{(l)}, \mathbf{u}^{(l)}}$ denotes the angle between $\mathbf{r}^{(l)} := (\mathbf{x}_1^{(l)}, \ldots, \mathbf{x}_{n_l}^{(l)}, 1)$ and $\mathbf{u}^{(l)} := (\mathbf{u}_1^{(l)}, \ldots, \mathbf{u}_{n_l}^{(l)}, 0)$ in $\mathbb{R}^{n_l+1}$, where $\mathbf{u}^{(l)} = \overline{W}^{(l)} \mathbf{u}^{(l-1)} / \|\overline{W}^{(l)} \mathbf{u}^{(l-1)}\|$ when $\overline{W}^{(l)} \mathbf{u}^{(l-1)} \neq 0$ and 0 otherwise, and $\mathbf{u}^{(0)} = \mathbf{u}$. The matrices $\overline{W}^{(l)}$ consist of rows $\overline{W}_i^{(l)} = W_{i,k'}^{(l)} \in \mathbb{R}^{n_{l-1}}$, where $k' = \operatorname{argmax}_{k \in [K]} \{W_{i,k}^{(l)} \mathbf{x}^{(l-1)} + b_{i,k}^{(l)}\}$.*

*Proof.* By Proposition 13, almost surely with respect to the parameter initialization,

$$\|\mathbf{J}_{\mathcal{N}}(\mathbf{x})\mathbf{u}\|^2 = \|W^{(L)}\mathbf{u}^{(L-1)}\|^2 \prod_{l=1}^{L-1} \sum_{i=1}^{n_l} \langle \overline{W}_i^{(l)}, \mathbf{u}^{(l-1)} \rangle^2, \tag{8}$$

where vectors $\mathbf{u}^{(l)}$, $l = 0, \ldots, L-1$ are defined recursively as $\mathbf{u}^{(l)} = \overline{W}^{(l)}\mathbf{u}^{(l-1)}/\|\overline{W}^{(l)}\mathbf{u}^{(l-1)}\|$ and $\mathbf{u}^{(0)} = \mathbf{u}$. Matrices $\overline{W}^{(l)}$ consist of rows $\overline{W}_i^{(l)} = W_{i,k'}^{(l)} \in \mathbb{R}^{n_{l-1}}$, $i = 1, \ldots, n_l$, where $k' = \operatorname{argmax}_{k \in [K]}\{W_{i,k}^{(l)}\mathbf{x}^{(l-1)} + b_{i,k}^{(l)}\}$, and $\mathbf{x}^{(l-1)}$ is the $l$th layer's input, $\mathbf{x}^{(0)} = \mathbf{x}$.

We assumed that weights in the last layer are sampled from a Gaussian distribution with mean zero and variance $1/n_{L-1}$. Then, by Lemma 10, $\|W^{(L)}\mathbf{u}^{(L-1)}\|^2 \overset{d}{=} (1/n_{L-1})\chi_{n_L}^2$ and is independent of $\mathbf{u}^{(L-1)}$. We use this observation in the equation (6), multiply and divide the summands in the expression by $c/n_{l-1}$ and rearrange to obtain that

$$\begin{aligned}\|\mathbf{J}_{\mathcal{N}}(\mathbf{x})\mathbf{u}\|^2 &\overset{d}{=} \frac{1}{n_{L-1}}\chi_{n_L}^2 \prod_{l=1}^{L-1} \frac{c}{n_{l-1}} \sum_{i=1}^{n_l} \left(\frac{n_{l-1}}{c}\langle \overline{W}_i^{(l)}, \mathbf{u}^{(l-1)}\rangle^2\right) \\ &= \frac{1}{n_0}\chi_{n_L}^2 \prod_{l=1}^{L-1} \frac{c}{n_l} \sum_{i=1}^{n_l} \left(\frac{n_{l-1}}{c}\langle \overline{W}_i^{(l)}, \mathbf{u}^{(l-1)}\rangle^2\right).\end{aligned} \tag{9}$$

We define $\mathbf{r}^{(l-1)} := (\mathbf{x}_1^{(l-1)}, \ldots, \mathbf{x}_{n_{l-1}}^{(l-1)}, 1) \in \mathbb{R}^{n_{l-1}+1}$ and $\mathbf{u}^{(l-1)} := (\mathbf{u}_1^{(l-1)}, \ldots, \mathbf{u}_{n_{l-1}}^{(l-1)}, 0) \in \mathbb{R}^{n_{l-1}+1}$, $\|\mathbf{u}\| = 1$. We append the vectors of biases to the weight matrices and denote obtained matrices with $\mathfrak{W}^{(l)} \in \mathbb{R}^{n_l \times (n_{l-1}+1)}$. Then (9) equals

$$\frac{1}{n_0}\chi_{n_L}^2 \prod_{l=1}^{L-1} \frac{c}{n_l} \sum_{i=1}^{n_l} \left(\frac{n_{l-1}}{c}\langle \overline{\mathfrak{W}}_i^{(l)}, \mathbf{u}^{(l-1)}\rangle^2\right).$$

Now we focus on $\sqrt{n_{l-1}/c}\,\langle \overline{\mathfrak{W}}_i^{(l)}, \mathbf{u}^{(l-1)}\rangle$. Since we have assumed that the weights and biases are sampled from the Gaussian distribution with zero mean and variance $c/n_{l-1}$, any weight $W_{i,k,j}^{(l)}$, $j = 1, \ldots, n_{l-1}$ (or bias), can be written as $\sqrt{c/n_{l-1}}v_{i,k,j}^{(l)}$, where $v_{i,k,j}^{(l)}$ is standard Gaussian. Therefore,

$$\frac{n_{l-1}}{c}\langle \overline{\mathfrak{W}}_i^{(l)}, \mathbf{u}^{(l-1)}\rangle^2 = \langle \overline{\mathfrak{V}}_i^{(l)}, \mathbf{u}^{(l-1)}\rangle^2, \tag{10}$$

where $\overline{\mathfrak{V}}_i^{(l)} = \mathfrak{V}_{i,k'}^{(l)} \in \mathbb{R}^{n_{l-1}+1}$, $k' = \operatorname{argmax}_{k \in [K]}\{\langle \mathfrak{V}_{i,k}^{(l)}, \mathbf{r}^{(l-1)}\rangle\}$, $\mathfrak{V}_{i,k}^{(l)}$ are $(n_{l-1}+1)$-dimensional standard Gaussian random vectors.

We construct an orthonormal basis $B = (\mathbf{b}_1, \ldots, \mathbf{b}_{n_{l-1}+1})$ of $\mathbb{R}^{n_{l-1}+1}$, where we set $\mathbf{b}_1 = \mathbf{r}^{(l-1)}/\|\mathbf{r}^{(l-1)}\|$ and choose the other vectors to be unit vectors orthogonal to $\mathbf{b}_1$. The change of basis matrix from the standard basis $I$ to the basis $B$ is given by $B^T$; see, e.g., Anton & Rorres (2013, Theorem 6.6.4). Then, any row $\mathfrak{V}_{i,k}^{(l)}$ an be expressed as

$$\mathfrak{V}_{i,k}^{(l)} = c_{k,1}\mathbf{b}_1 + \cdots + c_{k,n_{l-1}+1}\mathbf{b}_{n_{l-1}+1},$$

where $c_{k,j} = \langle \mathfrak{V}_{i,k}^{(l)}, \mathbf{b}_j \rangle$, $j = 1, \ldots, n_{l-1}+1$.

The coordinate vector of $\mathbf{r}^{(l-1)}$ relative to $B$ is $(\|\mathbf{r}^{(l-1)}\|, 0, \ldots, 0)$. Vector $\mathbf{u}^{(l-1)}$ in $B$ has the coordinate vector $(\langle \mathbf{u}^{(l-1)}, \mathbf{b}_1\rangle, \ldots, \langle \mathbf{u}^{(l-1)}, \mathbf{b}_{n_{l-1}+1}\rangle)$. This coordinate vector has norm 1 since the change of basis between two orthonormal bases does not change the $\ell_2$ norm; see, e.g., Anton & Rorres (2013, Theorem 6.3.2).

For the maximum, using the representation of the vectors in the basis $B$, we get

$$\langle \overline{\mathfrak{V}}_i^{(l)}, \mathbf{r}^{(l-1)}\rangle = \max_{k \in [K]}\left\{\langle \mathfrak{V}_{i,k}^{(l)}, \mathbf{r}^{(l-1)}\rangle\right\} = \max_{k \in [K]}\left\{c_{k,1}\|\mathbf{r}^{(l-1)}\|\right\} = \|\mathbf{r}^{(l-1)}\| \max_{k \in [K]}\{c_{k,1}\}. \tag{11}$$

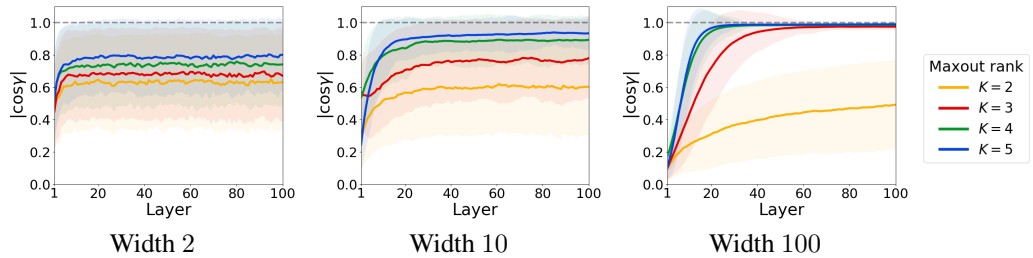

Figure 4: The plots show that $|\cos\gamma_{\mathbf{r}^{(l)},\mathbf{u}^{(l)}}|$ grows with the network depth and eventually converges to 1 for wide networks and maxout rank $K > 2$. The results were averaged over 1000 parameter initializations, and both weights and biases were sampled from $N(0, c/\text{fan-in})$, $c = 1/\mathbb{E}[(\Xi(N(0,1),K))^2]$, as discussed in Section 4. Vectors $\mathbf{x}$ and $\mathbf{u}$ were sampled from $N(0, I)$.

Therefore, in the basis $B$, $\overline{\mathfrak{V}}_i^{(l)}$ has components $(\max_{k\in[K]}\{c_{k,1}\}, c_{k',2}, \ldots, c_{k',n_{l-1}+1})$. By Theorem 15, for all $k = 1, \ldots, K$, $j = 1, \ldots, n_{l-1} + 1$, the coefficients $c_{k,j}$ are mutually independent standard Gaussian random variables that are also independent of vectors $\mathbf{b}_j$, $j = 1, \ldots, n_{l-1}$, by Lemma 10 and of $\mathbf{u}^{(l-1)}$.

$$\langle\overline{\mathfrak{V}}_i^{(l)}, \mathbf{u}^{(l-1)}\rangle = \max_{k\in[K]}\{c_{k,1}\}\langle\mathbf{u}^{(l-1)}, \mathbf{b}_1\rangle + \sum_{j=2}^{n_{l-1}+1} c_{k',j}\langle\mathbf{u}^{(l-1)}, \mathbf{b}_j\rangle$$

$$\stackrel{d}{=} \Xi_{l,i}(N(0,1),K)\langle\mathbf{u}^{(l-1)}, \mathbf{b}_1\rangle + \sum_{j=2}^{n_{l-1}+1} v_j\langle\mathbf{u}^{(l-1)}, \mathbf{b}_j\rangle,$$

(12)

where $\Xi_{l,i}(N(0,1),K)$ is the largest order statistic in a sample of $K$ standard Gaussian random variables, and $v_j \sim N(0,1)$. Since we have simply written equality in distribution for $\max_{k\in[K]}\{c_{k,1}\}$ and $c_{k',j}$, the variables $\Xi_{l,i}(N(0,1),K)$ and $v_j$, $j = 2, \ldots, n_{l-1}$ are also mutually independent, and independent of vectors $\mathbf{b}_j$, $j = 1, \ldots, n_{l-1}$, and of $\mathbf{u}^{(l-1)}$. In the following we will use $\Xi_{l,i}$ as a shorthand for $\Xi_{l,i}(N(0,1),K)$.

A linear combination $\sum_{i=1}^n a_i, v_i$ of Gaussian random variables $v_1, \ldots, v_n$, $v_j \sim N(\mu_j, \sigma_j^2)$, $j = 1, \ldots, n$ with coefficients $a_1, \ldots, a_n$ is distributed according to $N(\sum_{i=1}^n a_i\mu_i, \sum_{i=1}^n a_i^2\sigma_i^2)$. Hence, $\sum_{j=2}^{n_{l-1}+1} v_j\langle\mathbf{u}^{(l-1)}, \mathbf{b}_j\rangle \sim N(0, \sum_{j=2}^{n_{l-1}+1}\langle\mathbf{u}^{(l-1)}, \mathbf{b}_j\rangle^2)$. Since $\sum_{j=2}^{n_{l-1}+1}\langle\mathbf{u}^{(l-1)}, \mathbf{b}_j\rangle^2 = 1 - \langle\mathbf{u}^{(l-1)}, \mathbf{b}_1\rangle^2 = 1 - \cos^2\gamma_{\mathbf{r}^{(l-1)},\mathbf{u}^{(l-1)}}$, we get

$$\sum_{j=2}^{n_{l-1}+1} v_j\langle\mathbf{u}^{(l-1)}, \mathbf{b}_j\rangle + \Xi_{l,i}\langle\mathbf{u}^{(l-1)}, \mathbf{b}_1\rangle$$

(13)

$$\stackrel{d}{=} v_i\sqrt{1 - \cos^2\gamma_{\mathbf{r}^{(l-1)},\mathbf{u}^{(l-1)}}} + \Xi_{l,i}\cos\gamma_{\mathbf{r}^{(l-1)},\mathbf{u}^{(l-1)}},$$

where $v_i \sim N(0,1)$. Notice that $v_i\sqrt{1 - \cos^2\gamma_{\mathbf{r}^{(l-1)},\mathbf{u}^{(l-1)}}}$ and $\Xi_{l,i}\cos\gamma_{\mathbf{u}^{(l-1)},\mathbf{x}^{(l-1)}}$ are stochastically independent because $v_i$ and $\Xi_{l,i}$ are independent and multiplying random variables by constants does not affect stochastic independence. $\square$

**Remark 17.** The result in Theorem 3 also holds when the biases are initialized to zero. The proof is simplified in this case. There is no need to define additional vectors $\mathbf{r}^{(l-1)}$ and $\mathbf{u}^{(l-1)}$, and when constructing the basis, the first vector is defined as $\mathbf{b}_1 := \mathbf{x}^{(l-1)}/\|\mathbf{x}^{(l-1)}\|$. The rest of the proof remains the same.

**Remark 18** (Effects of the width and depth on a maxout network). According to Theorem 3, the behavior of $\|\mathbf{J}_\mathcal{N}(\mathbf{x})\mathbf{u}\|^2$ in a maxout network depends on the $\cos\gamma_{\mathbf{r}^{(l-1)},\mathbf{u}^{(l-1)}}$, which changes as the network gets wider or deeper. Figure 7 demonstrates how the width and depth affect $\|\mathbf{J}_\mathcal{N}(\mathbf{x})\mathbf{u}\|^2$.

**Wide shallow networks** Since independent and isotropic random vectors in high-dimensional spaces tend to be almost orthogonal (Vershynin, 2018, Remark 2.3.5), $\cos\gamma_{\mathbf{x}^{(0)},\mathbf{u}^{(0)}}$ will be close

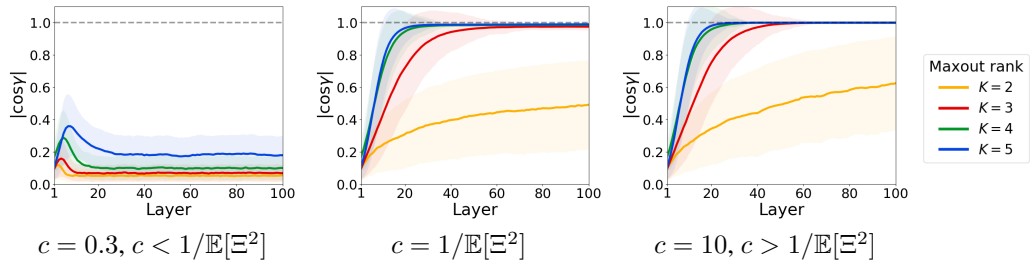

$$c = 0.3, c < 1/\mathbb{E}[\Xi^2] \qquad c = 1/\mathbb{E}[\Xi^2] \qquad c = 10, c > 1/\mathbb{E}[\Xi^2]$$

Figure 5: The plots show that $|\cos \gamma_{\mathbf{r}^{(l)}, \mathbf{u}^{(l)}}|$ does not converge to 1 for $c < 1/\mathbb{E}[\Xi^2]$ and converges for $c \geq 1/\mathbb{E}[\Xi^2]$. The network had 100 neurons at each layer, and both weights and biases were sampled from $N(0, c/\text{fan-in})$. The results were averaged over 1000 parameter initializations. Vectors $\mathbf{x}$ and $\mathbf{u}$ were sampled from $N(0, I)$.

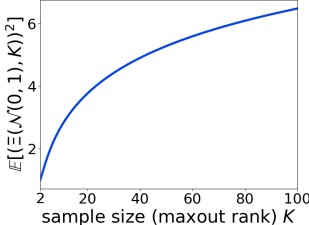

Figure 6: Second moment of $\Xi(N(0,1), K)$ for different sample sizes $K$. It increases with $K$ for any $K$, $2 \leq K \leq 100$, and $\mathbb{E}[(\Xi(N(0,1), K))^2] > 1$ for $K > 2$.

to 0 with high probability for wide networks if the entries of the vectors $\mathbf{x}$ and $\mathbf{u}$ are i.i.d. standard Gaussian (or i.i.d. from an isotropic distribution). Hence, we expect that the cosine will be around zero for the earlier layers of wide networks and individual units will behave more as the squared standard Gaussians.

**Wide deep networks** Consider wide and deep networks, where the layers $l = 0, \ldots, L - 1$ are approximately of the same width $n_{l_1} \approx n_{l_2}, l_1, l_2 = 0, \ldots, L - 1$. Assume that $c = 1/\mathcal{M} = 1/\mathbb{E}[(\Xi(N(0,1), K))^2]$. We will demonstrate that under these conditions. $|\cos \gamma_{\mathbf{r}^{(l)}, \mathbf{u}^{(l)}}| \approx 1$ for the later layers for $2 < K < 100$. Thus, individual units behave as the squared largest order statistics. To see this, we need to estimate $\cos \gamma_{\mathbf{r}^{(l)}, \mathbf{u}^{(l)}}$ from Theorem 3, which is defined as

$$\cos \gamma_{\mathbf{r}^{(l)}, \mathbf{u}^{(l)}} = \rho_{\mathbf{r} \mathbf{u}}^{(l)} = \frac{\langle \mathbf{r}^{(l)}, \mathbf{u}^{(l)} \rangle}{\|\mathbf{r}^{(l)}\| \|\mathbf{u}^{(l)}\|} = \frac{\langle \mathbf{r}^{(l)}, \overline{\mathbf{u}}^{(l)} \rangle}{\|\mathbf{r}^{(l)}\| \|\overline{\mathbf{u}}^{(l)}\|} = \frac{\frac{n_{l-1}}{n_l} \langle \mathbf{r}^{(l)}, \overline{\mathbf{u}}^{(l)} \rangle}{\left( \sqrt{\frac{n_{l-1}}{n_l}} \|\mathbf{r}^{(l)}\| \right) \left( \sqrt{\frac{n_{l-1}}{n_l}} \|\overline{\mathbf{u}}^{(l)}\| \right)},$$

where we denoted $\cos \gamma_{\mathbf{r}^{(l)}, \mathbf{u}^{(l)}}$ with $\rho_{\mathbf{r} \mathbf{u}}^{(l)}$, and with $\overline{\mathbf{u}}^{(l)}, \mathbf{u}^{(l)}$ before the normalization.

Firstly, for $\mathbf{r}^{(l)}$ we get

$$\frac{n_{l-1}}{n_l} \|\mathbf{r}^{(l)}\|^2 = \frac{n_{l-1}}{n_l} \left( \sum_{i=1}^{n_l} \left( \max_{k \in [K]} \left\{ \mathfrak{W}_{i,k}^{(l)} \mathbf{r}^{(l-1)} \right\} \right)^2 + 1 \right)$$

$$= c \|\mathbf{r}^{(l-1)}\|^2 \left( \frac{1}{n_l} \sum_{i=1}^{n_l} \left( \max_{k \in [K]} \left\{ \mathfrak{V}_{i,k}^{(l)} \frac{\mathbf{r}^{(l-1)}}{\|\mathbf{r}^{(l-1)}\|} \right\} \right)^2 + \frac{n_{l-1}}{c \|\mathbf{r}^{(l-1)}\|^2 n_l} \right) \qquad (14)$$

$$\stackrel{d}{=} c \|\mathbf{r}^{(l-1)}\|^2 \left( \frac{1}{n_l} \sum_{i=1}^{n_l} \Xi_{l,i}^2 + \frac{n_{l-1}}{c \|\mathbf{r}^{(l-1)}\|^2 n_l} \right),$$

where in the second line we used that $\mathfrak{W}_{i,k}^{(l)} = \sqrt{c/n_{l-1}} \mathfrak{V}_{i,k}^{(l)}, \mathfrak{V}_{i,k,j}^{(l)} \sim N(0,1), j = 1, \ldots, n_{l-1}$. In the third line, $\Xi_{l,i} \stackrel{d}{=} \Xi(N(0,1), K)$ is the largest order statistic in a sample of $K$ standard

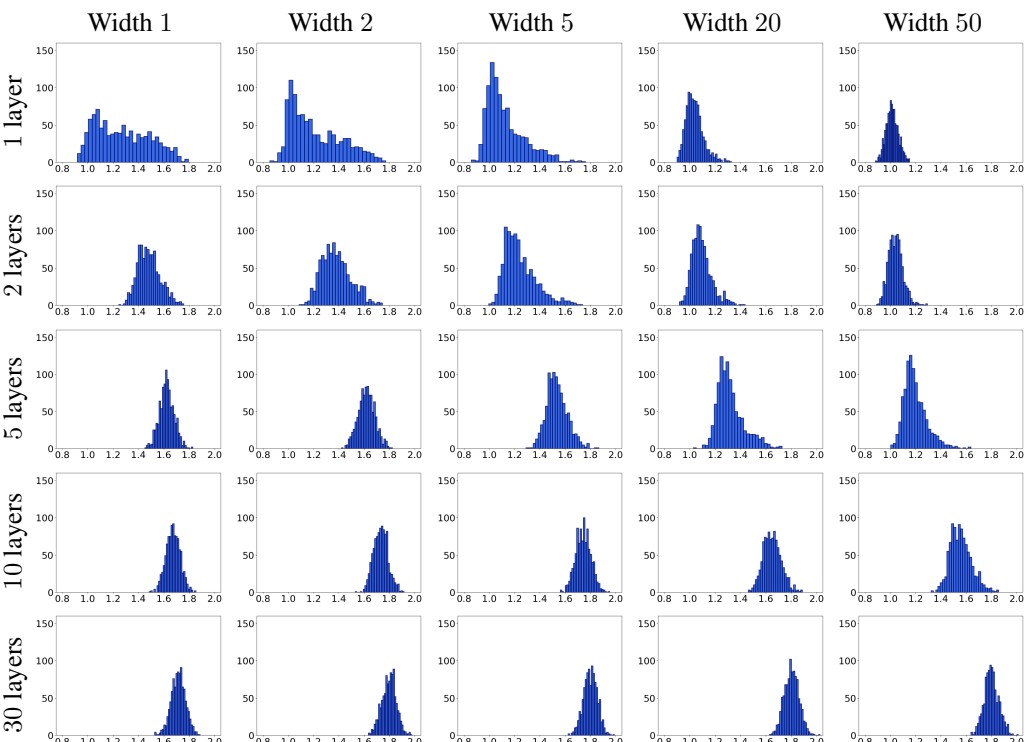

Figure 7: Shown is the expectation value of the square norm of the directional derivative of the input output map of a maxout network for a fixed random direction with respect to the weights, plotted as a function of the input. Weights and biases are sampled from $N(0, 1/\text{fan-in})$ and biases are zero. Inputs are standard Gaussian vectors. Vector **u** is a one-hot vector with 1 at a random position, and it is the same for one setup. We sampled 1000 inputs and 1000 initializations for each input. The left end corresponds to the second moments of the Gaussian distribution and the right end to the second moment of the largest order statistic. Observe that for wide and deep networks the mean is closer to the second moment of the largest order statistic.

Gaussians, since by Lemma 10, $\mathfrak{V}_{i,k}^{(l)}\mathfrak{r}^{(l-1)}/\|\mathfrak{r}^{(l-1)}\|$ are mutually independent standard Gaussian random variables. When the network width is large, $1/n_l \sum_{i=1}^{n_l} \Xi_{l,i}^2$ approximates the second moment of the largest order statistic, and $n_{l-1}/n_l \approx 1$ when the layer widths are approximately the same. Then

$$\frac{n_{l-1}}{n_l}\|\mathfrak{r}^{(l)}\|^2 \approx c\|\mathfrak{r}^{(l-1)}\|^2 \left(\mathbb{E}\left[\Xi^2\right] + \frac{1}{c\|\mathfrak{r}^{(l-1)}\|^2}\right).$$

Now we will show that $1/\|\mathfrak{r}^{(l-1)}\|^2 \approx 0$. Firstly, by the same reasoning as above,

$$\|\mathfrak{r}^{(l-1)}\|^2 = \sum_{i=1}^{n_{l-1}} \left(\max_{k\in[K]}\left\{\mathfrak{W}_{i,k}^{(l)}\mathfrak{r}^{(l-2)}\right\}\right)^2 + 1$$

$$\overset{d}{=} \|\mathfrak{r}^{(0)}\|^2 \frac{c^{l-1}n_{l-1}}{n_0}\prod_{j=1}^{l-1}\frac{1}{n_j}\sum_{i=1}^{n_j}\Xi_{l,i}^2 + \cdots + c^2\frac{n_{l-1}}{n_{l-3}}\prod_{j=l-2}^{l-1}\frac{1}{n_j}\sum_{i=1}^{n_j}\Xi_{l,i}^2 + \frac{cn_{l-1}}{n_{l-2}}\frac{1}{n_{l-1}}\sum_{i=1}^{n_{l-1}}\Xi_{l,i}^2 + 1.$$

Since we assumed that the layer widths are large and approximately the same,

$$\|\mathfrak{r}^{(l-1)}\|^2 \approx \|\mathfrak{r}^{(0)}\|^2\left(c\mathbb{E}[\Xi^2]\right)^{l-1} + \cdots + c\mathbb{E}[\Xi^2] + 1 = \|\mathfrak{r}^{(0)}\|^2\left(c\mathbb{E}[\Xi^2]\right)^{l-1} + \sum_{j=0}^{l-2}\left(c\mathbb{E}[\Xi^2]\right)^j.$$

Using the assumption that $c = 1/\mathbb{E}[\Xi^2]$, we obtain that $\|\mathfrak{r}^{(l-1)}\|^2 \approx \|\mathfrak{r}^{(0)}\|^2 + (l-1)$ and goes to infinity with the network depth. Hence, $1/\|\mathfrak{r}^{(l-1)}\|^2 \approx 0$ and

$$\frac{n_{l-1}}{n_l}\|\mathfrak{r}^{(l)}\|^2 \approx c\|\mathfrak{r}^{(l-1)}\|^2\mathbb{E}\left[\Xi^2\right].$$

Now consider $\overline{\mathbf{u}}^{(l)}$. Using the reasoning from Theorem 3, see equations (12) and (13), $\overline{\mathbf{u}}_i^{(l)} \overset{d}{=} c/n_{l-1}(\Xi_{l,i}\rho_{\mathfrak{r}\mathbf{u}}^{(l-1)} + v_i\sqrt{1-(\rho_{\mathfrak{r}\mathbf{u}}^{(l-1)})^2})$, $i = 1,\ldots,n_l$, $v_i \sim N(0,1)$. Then in a wide network

$$\|\overline{\mathbf{u}}^{(l)}\|^2 \approx \frac{cn_l}{n_{l-1}}\mathbb{E}\left[\left(\Xi\rho_{\mathfrak{r}\mathbf{u}}^{(l-1)} + v\sqrt{1-\left(\rho_{\mathfrak{r}\mathbf{u}}^{(l-1)}\right)^2}\right)^2\right]. \tag{15}$$

Note that the random variable $\Xi$ in equations (14) and (15) is the same based on the derivations in Theorem 3, to see this, compare equations (11) and (12).

Similarly, for the dot product $\langle \mathfrak{r}^{(l)}, \overline{\mathbf{u}}^{(l)}\rangle$ in a wide network we obtain that

$$\langle \mathfrak{r}^{(l)}, \overline{\mathbf{u}}^{(l)}\rangle \approx \|\mathfrak{r}^{(l-1)}\|\frac{cn_l}{n_{l-1}}\mathbb{E}\left[\Xi\left(\Xi\rho_{\mathfrak{r}\mathbf{u}}^{(l-1)} + v\sqrt{1-\left(\rho_{\mathfrak{r}\mathbf{u}}^{(l-1)}\right)^2}\right)\right].$$

Hence, we have the following recursive map for $\rho_{\mathfrak{r}\mathbf{u}}^{(l)}$

$$\rho_{\mathfrak{r}\mathbf{u}}^{(l)} = \frac{\mathbb{E}\left[\Xi\left(\Xi\rho_{\mathfrak{r}\mathbf{u}}^{(l-1)} + v\sqrt{1-\left(\rho_{\mathfrak{r}\mathbf{u}}^{(l-1)}\right)^2}\right)\right]}{\sqrt{\mathbb{E}\left[\Xi^2\right]}\sqrt{\mathbb{E}\left[\left(\Xi\rho_{\mathfrak{r}\mathbf{u}}^{(l-1)} + v\sqrt{1-\left(\rho_{\mathfrak{r}\mathbf{u}}^{(l-1)}\right)^2}\right)^2\right]}}$$

$$= \frac{1}{\sqrt{\mathbb{E}\left[\Xi^2\right]}}\frac{\rho_{\mathfrak{r}\mathbf{u}}^{(l-1)}\mathbb{E}[\Xi^2]}{\sqrt{\left(\rho_{\mathfrak{r}\mathbf{u}}^{(l-1)}\right)^2(\mathbb{E}[\Xi^2]-1)+1}},$$

where we used independence of $v$ and $\Xi$, see Theorem 3, and that $\mathbb{E}[v] = 0$ and $\mathbb{E}[v^2] = 1$. This map has fixed points $\rho^* = \pm 1$, which can be confirmed by direct calculation. To check if these fixed points are stable, we need to consider the values of the derivative $\partial\rho_{\mathfrak{r}\mathbf{u}}^{(l)}/\partial\rho_{\mathfrak{r}\mathbf{u}}^{(l-1)}$ at them. We obtain

$$\frac{\partial\rho_{\mathfrak{r}\mathbf{u}}^{(l)}}{\partial\rho_{\mathfrak{r}\mathbf{u}}^{(l-1)}} = \left(\mathbb{E}[\Xi^2]\right)^{\frac{1}{2}}\left(\left(\rho_{\mathfrak{r}\mathbf{u}}^{(l-1)}\right)^2(\mathbb{E}[\Xi^2]-1)+1\right)^{-\frac{3}{2}}.$$

When $\rho_{\mathbf{ru}}^{(l-1)} = \pm 1$ this partial derivative equals $1/\mathbb{E}[\Xi^2] < 1$ for $K > 2$, since $\mathbb{E}[\Xi^2] > 1$, see Table 4 for $K = 2, \ldots, 10$ and Figure 6 for $K = 2, \ldots, 100$. Hence, the fixed points are stable (Strogatz, 2018, Chapter 10.1). Note that for $K = 2$, $1/\mathbb{E}[\Xi^2] = 1$, and this analysis is inconclusive. Therefore, if the network parameters are sampled from $N(0, c/n_{l-1})$, $c = 1/\mathcal{M} = 1/\mathbb{E}[\Xi(N(0,1), K)^2]$, we expect that $|\cos \gamma_{\mathbf{r}^{(l)}, \mathbf{u}^{(l)}}| \approx 1$ for the later layers of deep networks and individual units will behave more as the squared largest order statistics. Figure 4 demonstrates convergence of $|\cos \gamma_{\mathbf{r}^{(l)}, \mathbf{u}^{(l)}}|$ to 1 with the depth for wide networks, and Figure 5 shows that there is no convergence for $c < 1/\mathbb{E}[\Xi^2]$ and that the cosine still converges for $c > 1/\mathbb{E}[\Xi^2]$.

**Remark 19** (Expectation of $\|\mathbf{J}_{\mathcal{N}}(\mathbf{x})\mathbf{u}\|^2$ in a wide and deep network). According to Remark 18, for deep and wide networks, we can expect that $|\cos \gamma_{\mathbf{r}^{(l-1)}, \mathbf{u}^{(l-1)}}| \approx 1$ if $c = 1/\mathcal{M}$, which allows obtaining an approximate equality for the expectation of $\|\mathbf{J}_{\mathcal{N}}(\mathbf{x})\mathbf{u}\|^2$. Hence, using Theorem 3,

$$\|\mathbf{J}_{\mathcal{N}}(\mathbf{x})\mathbf{u}\|^2 \approx \frac{1}{n_0} \chi_{n_L}^2 \prod_{l=1}^{L-1} \frac{c}{n_l} \sum_{i=1}^{n_l} (\Xi_{l,i}(N(0,1), K))^2 . \tag{16}$$

Then, using mutual independence of the variables in equation (16),

$$\mathbb{E}[\|\mathbf{J}_{\mathcal{N}}(\mathbf{x})\mathbf{u}\|^2] \approx \frac{1}{n_0} \mathbb{E}\left[\chi_{n_L}^2\right] \prod_{l=1}^{L-1} \frac{c}{n_l} \sum_{i=1}^{n_l} \mathbb{E}\left[(\Xi_{l,i}(N(0,1), K))^2\right] .$$

Since $\mathcal{M} = \mathbb{E}[(\Xi_{l,i}(N(0,1), K))^2]$, see Table 4, and $c = 1/\mathcal{M}$, we get

$$\mathbb{E}[\|\mathbf{J}_{\mathcal{N}}(\mathbf{x})\mathbf{u}\|^2] \approx \frac{n_L}{n_0}(c\mathcal{M})^{L-1} = \frac{n_L}{n_0} .$$

**Remark 20** (Lower bound on the moments in a wide and deep network). Using (16) and taking into account the mutual independence of the variables,

$$\mathbb{E}[\|\mathbf{J}_{\mathcal{N}}(\mathbf{x})\mathbf{u}\|^{2t}] \approx \mathbb{E}\left[\left(\frac{1}{n_0} \chi_{n_L}^2 \prod_{l=1}^{L-1} \frac{c}{n_l} \sum_{i=1}^{n_l} (\Xi_{l,i}(N(0,1), K))^2\right)^t\right]$$

$$= \left(\frac{n_L}{n_0}\right)^t \left(\frac{1}{n_L}\right)^t \mathbb{E}\left[(\chi_{n_L}^2)^t\right] \prod_{l=1}^{L-1} \left(\frac{c}{n_l}\right)^t \mathbb{E}\left[\left(\sum_{i=1}^{n_l} (\Xi_{l,i}(N(0,1), K))^2\right)^t\right]$$

$$\geq \left(\frac{n_L}{n_0}\right)^t \left(\frac{1}{n_L}\right)^t \mathbb{E}\left[(\chi_{n_L}^2)^t\right] \prod_{l=1}^{L-1} \left(\frac{c}{n_l}\right)^t \left(\sum_{i=1}^{n_l} \mathbb{E}\left[(\Xi_{l,i}(N(0,1), K))^2\right]\right)^t , \tag{17}$$

where in the last inequality, we used linearity of expectation and Jensen's inequality since taking the $t$th power for $t \geq 1$ is a convex function for non-negative arguments. Using the formula for noncentral moments of the chi-squared distribution and the inequality $\ln x \geq 1 - 1/x, \forall x > 0$, meaning that $x = \exp\{\ln x\} \geq \exp\{1 - 1/x\}$, we get

$$\left(\frac{1}{n_L}\right)^t \mathbb{E}\left[(\chi_{n_L}^2)^t\right] = \left(\frac{1}{n_l}\right)^t (n_L)(n_L + 2)\cdots(n_L + 2t - 2) = \prod_{i=0}^{t-1}\left(1 + \frac{2i}{n_L}\right)$$

$$\geq \exp\left\{\sum_{i=1}^{t-1}\left(\frac{2i}{n_L + 2i}\right)\right\} \geq \exp\left\{\frac{t-1}{2n_L}\right\} , \tag{18}$$

where in the last inequality, we used that $2i/(n_L + 2i) \geq 2/(n_L + 2) \geq 1/(2n_L)$ for all $i, n_L \geq 1$.

Using that $\mathbb{E}[(\Xi_{l,i}(N(0,1), K))^2] = \mathcal{M}$, see Table 4, and combing this with (17) and (18),

$$\mathbb{E}[\|\mathbf{J}_{\mathcal{N}}(\mathbf{x})\mathbf{u}\|^{2t}] \gtrapprox \left(\frac{n_L}{n_0}\right)^t \exp\left\{\frac{t-2}{2n_L}\right\} (c\mathcal{M})^{t(L-1)} . \tag{19}$$

The bound in (19) can be tightened if a tighter lower bound on the moments of the sum of the squared largest order statistics in a sample of $K$ standard Gaussians is known. To derive a lower bound on the moments $t \geq 2$ for the general case in Corollary 2, it is necessary to obtain a non-trivial lower bound on the moments of the sum of the smallest order statistics in a sample of $K$ chi-squared random variables with 1 degree of freedom.

# E  ACTIVATION LENGTH

Here we prove the results from Subsection 3.3. Figure 8 demonstrates a close match between the estimated normalized activation length and the behavior predicted in Corollary 21 and 5.

**Corollary 21** (Distribution of the normalized activation length). *Consider a maxout network with the settings of Section 2. Then, almost surely with respect to the parameter initialization, for any input into the network $\mathbf{x} \in \mathbb{R}^{n_0}$ and $l' = 1, \ldots, L - 1$, the normalized activation length $A^{(l')}$ is equal in distribution to*

$$\|\mathfrak{x}^{(0)}\|^2 \frac{1}{n_0} \prod_{l=1}^{l'} \left( \frac{c}{n_l} \sum_{i=1}^{n_l} \Xi_{l,i}(N(0,1),K)^2 \right) + \sum_{j=2}^{l'} \left[ \frac{1}{n_{j-1}} \prod_{l=j}^{l'} \left( \frac{c}{n_l} \sum_{i=1}^{n_l} \Xi_{l,i}(N(0,1),K)^2 \right) \right],$$

*where $\mathfrak{x}^{(0)} := (\mathbf{x}_1, \ldots, \mathbf{x}_{n_0}, 1) \in \mathbb{R}^{n_0+1}$, $\Xi_{l,i}(N(0,1),K)$ is the largest order statistic in a sample of $K$ standard Gaussian random variables, and $\Xi_{l,i}(N(0,1),K)$ are stochastically independent. Notice that variables $\Xi_{l,i}(N(0,1),K)$ with the same indices are the same random variables.*

*Proof.* Define $\mathfrak{x}^{(l)} = (\mathbf{x}_1, \ldots, \mathbf{x}_{n_l}, 1) \in \mathbb{R}^{n_l+1}$. Append the bias columns to the weight matrices and denote obtained matrices with $\mathfrak{W}^{(l)} \in \mathbb{R}^{n_l \times (n_{l-1}+1)}$. Denote $\mathfrak{W}^{(l)}_{i,k'} \in \mathbb{R}^{n_{l-1}+1}$, $k' = \operatorname{argmax}_{k \in [K]} \{ \langle \mathfrak{W}^{(l)}_{i,k}, \mathfrak{x}^{(l-1)} \rangle \}$, with $\overline{\mathfrak{W}}^{(l)}_i$. Under this notation, $\|\mathfrak{x}^{(l)}\|^2 = (\|\mathbf{x}^{(l)}\|^2 + 1)$, $\|\mathbf{x}^{(l)}\| = \|\overline{\mathfrak{W}}^{(l)} \mathfrak{x}^{(l-1)}\|$. Then $\|\mathbf{x}^{(l')}\|^2$ equals

$$\|\mathbf{x}^{(l')}\|^2 = \|\overline{\mathfrak{W}}^{(l')} \mathfrak{x}^{(l'-1)}\|^2 = \left\| \overline{\mathfrak{W}}^{(l')} \frac{\mathfrak{x}^{(l'-1)}}{\|\mathfrak{x}^{(l'-1)}\|} \right\|^2 \|\mathfrak{x}^{(l'-1)}\|^2$$

$$= \left\| \overline{\mathfrak{W}}^{(l')} \frac{\mathfrak{x}^{(l'-1)}}{\|\mathfrak{x}^{(l'-1)}\|} \right\|^2 \left( \left\| \overline{\mathfrak{W}}^{(l'-1)} \frac{\mathfrak{x}^{(l'-2)}}{\|\mathfrak{x}^{(l'-2)}\|} \right\|^2 \|\mathfrak{x}^{(l'-2)}\|^2 + 1 \right)$$

$$= \cdots = \|\mathfrak{x}^{(0)}\|^2 \prod_{l=1}^{l'} \left\| \overline{\mathfrak{W}}^{(l)} \frac{\mathfrak{x}^{(l-1)}}{\|\mathfrak{x}^{(l-1)}\|} \right\|^2 + \sum_{j=2}^{l'} \left[ \prod_{l=j}^{l'} \left\| \overline{\mathfrak{W}}^{(l)} \frac{\mathfrak{x}^{(l-1)}}{\|\mathfrak{x}^{(l-1)}\|} \right\|^2 \right],$$

where we multiplied and divided $\|\overline{\mathfrak{W}}^{(l)} \mathfrak{x}^{(l-1)}\|^2$ by $\|\mathfrak{x}^{(l-1)}\|^2$ at each step. Using the approach from Theorem 3, more specifically equations (10), (12) and (13), with $\mathbf{u}^{(l)} = \mathfrak{x}^{(l)}/\|\mathfrak{x}^{(l)}\|$, implying that $\cos \gamma_{\mathfrak{x}^{(l)}, \mathbf{u}^{(l)}} = 1$,

$$A^{(l')} = \frac{1}{n_l} \|\mathbf{x}^{(l')}\|^2 \overset{d}{=} \|\mathfrak{x}^{(0)}\|^2 \frac{1}{n_{l'}} \prod_{l=1}^{l'} \left( \frac{c}{n_{l-1}} \sum_{i=1}^{n_l} \Xi_{l,i}^2 \right) + \frac{1}{n_{l'}} \sum_{j=2}^{l'} \left[ \prod_{l=j}^{l'} \left( \frac{c}{n_{l-1}} \sum_{i=1}^{n_l} \Xi_{l,i}^2 \right) \right]$$

$$= \|\mathfrak{x}^{(0)}\|^2 \frac{1}{n_0} \prod_{l=1}^{l'} \left( \frac{c}{n_l} \sum_{i=1}^{n_l} \Xi_{l,i}^2 \right) + \sum_{j=2}^{l'} \left[ \frac{1}{n_{j-1}} \prod_{l=j}^{l'} \left( \frac{c}{n_l} \sum_{i=1}^{n_l} \Xi_{l,i}^2 \right) \right],$$

where $\Xi_{l,i} = \Xi_{l,i}(N(0,1),K)$ is the largest order statistic in a sample of $K$ standard Gaussian random variables, and stochastic independence of variables $\Xi_{l,i}$ follows from Theorem 3. $\qquad\square$

**Corollary 5** (Moments of the activation length). *Consider a maxout network with the settings of Section 2. Let $\mathbf{x} \in \mathbb{R}^{n_0}$ be any input into the network. Then, for the moments of the normalized activation length, the following results hold.*

*Mean:*

$$\mathbb{E}\left[ A^{(l')} \right] = \|\mathfrak{x}^{(0)}\|^2 \frac{1}{n_0} (c\mathcal{M})^{l'} + \sum_{j=2}^{l'} \left( \frac{1}{n_{j-1}} (c\mathcal{M})^{l'-j+1} \right).$$

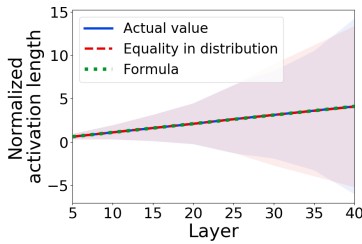

Figure 8: Comparison of the normalized activation length with the equality in distribution result from Corollary 21 and the formula for the mean from Corollary 5. Plotted are means and stds estimated with respect to the distribution of parameters / random variables $\Xi_{l,i}(N(0,1),K)$ averaged over $100,000$ initializations, and a numerically evaluated formula for the mean from Corollary 5. All layers had 10 neurons. The lines for the mean and areas for the std overlap. Note that there is no std for the formula in the plot.

*Variance:*

$$\mathrm{Var}[A^{(l')}] \leq 2 \frac{\|\mathbf{r}^{(0)}\|^4}{n_0^2} c^{2l'} K^{2l'} \exp\left\{ \sum_{l=1}^{l'} \frac{4}{n_l K} \right\}.$$

*Moments of the order $t \geq 2$:*

$$\mathbb{E}\left[ \left( A^{(l')} \right)^t \right] \leq 2^{t-1} \frac{\|\mathbf{r}^{(0)}\|^{2t}}{n_0^t} c^{tl'} K^{tl'} \exp\left\{ \sum_{l=1}^{l'} \frac{t^2}{n_l K} \right\}$$

$$+ (2(l'-1))^{t-1} \sum_{j=2}^{l'} \left( \frac{(cK)^{t(l'-j+1)}}{n_{j-1}^t} \exp\left\{ \sum_{l=j}^{l'} \frac{t^2}{n_l K} \right\} \right),$$

$$\mathbb{E}\left[ \left( A^{(l')} \right)^t \right] \geq \frac{\|\mathbf{r}^{(0)}\|^{2t}}{n_0^t} (c\mathcal{M})^{tl'} + \sum_{j=2}^{l'} \frac{(c\mathcal{M})^{t(l'-j+1)}}{n_{j-1}^t}.$$

*where expectation is taken with respect to the distribution of the network weights and biases, and $\mathcal{M}$ is a constant depending on $K$ that can be computed approximately, see Table 4 for the values for $K = 2, \ldots, 10$.*

*Proof.* **Mean** Taking expectation in Corollary 21 and using independence of $\Xi_{l,i}(N(0,1),K)$,

$$\mathbb{E}\left[ A^{(l')} \right] = \|\mathbf{r}^{(0)}\|^2 \frac{1}{n_0} (c\mathcal{M})^{l'} + \sum_{j=2}^{l'} \left( \frac{1}{n_{j-1}} (c\mathcal{M})^{l'-j+1} \right), \tag{20}$$

where $\mathcal{M}$ is the second moment of $\Xi_{l,i}(N(0,1),K)$, see Table 4 for its values for $K = 2, \ldots, 10$.

**Moments of the order $t \geq 2$** Using Corollary 21, we get

$$\mathbb{E}\left[ \left( A^{(l')} \right)^t \right]$$

$$= \mathbb{E}\left[ \left( \|\mathbf{r}^{(0)}\|^2 \frac{1}{n_0} \prod_{l=1}^{l'} \left( \frac{c}{n_l} \sum_{i=1}^{n_l} \Xi_{l,i}^2 \right) + \sum_{j=2}^{l'} \left[ \frac{1}{n_{j-1}} \prod_{l=j}^{l'} \left( \frac{c}{n_l} \sum_{i=1}^{n_l} \Xi_{l,i}^2 \right) \right] \right)^t \right]. \tag{21}$$

*Upper bound* First, we derive an upper bound on (21). Notice that all arguments in (21) are positive except for a zero measure set of $\Xi_{l,i} \in \mathbb{R}^{\sum_{l=1}^{l'} n_l}$. According to the power mean inequality, for any

$x_1, \dots, x_n \in \mathbb{R}, x_1, \dots, x_n > 0$ and any $t \in \mathbb{R}, t > 1, (x_1 + \dots + x_n)^t \le n^{t-1}(x_1^t + \dots + x_n^t)$. Using the power mean inequality first on the whole expression and then on the second summand,

$$\mathbb{E}\left[\left(A^{(l')}\right)^t\right] \le 2^{t-1}\left(\mathbb{E}\left[\left(\|\mathfrak{r}^{(0)}\|^2 \frac{1}{n_0}\prod_{l=1}^{l'}\left(\frac{c}{n_l}\sum_{i=1}^{n_l}\Xi_{l,i}^2\right)\right)^t\right]\right.$$
$$\left. + (l'-1)^{t-1}\sum_{j=2}^{l'}\mathbb{E}\left[\left(\frac{1}{n_{j-1}}\prod_{l=j}^{l'}\left(\frac{c}{n_l}\sum_{i=1}^{n_l}\Xi_{l,i}^2\right)\right)^t\right]\right). \tag{22}$$

Using independence of $\Xi_{l,i}$, (22) equals

$$2^{t-1}\frac{\|\mathfrak{r}^{(0)}\|^{2t}}{n_0^t}\prod_{l=1}^{l'}\left(\frac{c^t}{n_l^t}\mathbb{E}\left[\left(\sum_{i=1}^{n_l}\Xi_{l,i}^2\right)^t\right]\right)$$
$$+ (2(l'-1))^{t-1}\sum_{j=2}^{l'}\left(\frac{1}{n_{j-1}^t}\prod_{l=j}^{l'}\left(\frac{c^t}{n_l^t}\mathbb{E}\left[\left(\sum_{i=1}^{n_l}\Xi_{l,i}^2\right)^t\right]\right)\right).$$

Upper-bounding the largest order statistic with the sum of squared standard Gaussian random variables, we get that $\sum_{i=1}^{n_l}\Xi_{l,i}^2 \le \chi_{n_l K}^2$. Hence,

$$\mathbb{E}\left[\left(A^{(l')}\right)^t\right] \le 2^{t-1}\frac{\|\mathfrak{r}^{(0)}\|^{2t}}{n_0^t}\prod_{l=1}^{l'}\left(\frac{c^t}{n_l^t}\mathbb{E}\left[\left(\chi_{n_l K}^2\right)^t\right]\right)$$
$$+ (2(l'-1))^{t-1}\sum_{j=2}^{l'}\left(\frac{1}{n_{j-1}^t}\prod_{l=j}^{l'}\left(\frac{c^t}{n_l^t}\mathbb{E}\left[\left(\chi_{n_l K}^2\right)^t\right]\right)\right). \tag{23}$$

Using the formula for noncentral moments of the chi-squared distribution and $1 + x \le e^x, \forall x \in \mathbb{R}$,

$$\frac{c^t}{n_l^t}\mathbb{E}\left[\left(\chi_{n_l K}^2\right)^t\right] = \frac{c^t}{n_l^t}(n_l K)(n_l K + 2)\cdots(n_l K + 2t - 2)$$
$$= c^t K^t \cdot 1 \cdot \left(1 + \frac{2}{n_l K}\right)\cdots\left(1 + \frac{2t-2}{n_l K}\right) \le c^t K^t \exp\left\{\sum_{i=0}^{t-1}\frac{2i}{n_l K}\right\} \le c^t K^t \exp\left\{\frac{t^2}{n_l K}\right\},$$

where we used the formula for calculating the sum of consecutive numbers $\sum_{i=1}^{t-1} i = t(t-1)/2$. Using this result in (23), we get the final upper bound

$$\mathbb{E}\left[\left(A^{(l')}\right)^t\right] \le 2^{t-1}\frac{\|\mathfrak{r}^{(0)}\|^{2t}}{n_0^t}c^{tl'}K^{tl'}\exp\left\{\sum_{l=1}^{l'}\frac{t^2}{n_l K}\right\}$$
$$+ (2(l'-1))^{t-1}\sum_{j=2}^{l'}\left(\frac{(cK)^{t(l'-j+1)}}{n_{j-1}^t}\exp\left\{\sum_{l=j}^{l'}\frac{t^2}{n_l K}\right\}\right).$$

*Lower bound* Using that arguments in (21) are non-negative and $t \ge 1$, we can lower bound the power of the sum with the sum of the powers and get,

$$\mathbb{E}\left[\left(A^{(l')}\right)^t\right] \ge \frac{\|\mathfrak{r}^{(0)}\|^{2t}}{n_0^t}\prod_{l=1}^{l'}\left(\frac{c^t}{n_l^t}\mathbb{E}\left[\left(\sum_{i=1}^{n_l}\Xi_{l,i}^2\right)^t\right]\right) + \sum_{j=2}^{l'}\left(\frac{1}{n_{j-1}^t}\prod_{l=j}^{l'}\left(\frac{c^t}{n_l^t}\mathbb{E}\left[\left(\sum_{i=1}^{n_l}\Xi_{l,i}^2\right)^t\right]\right)\right)$$
$$\ge \frac{\|\mathfrak{r}^{(0)}\|^{2t}}{n_0^t}\prod_{l=1}^{l'}\left(\frac{c^t}{n_l^t}\left(\sum_{i=1}^{n_l}\mathbb{E}\left[\Xi_{l,i}^2\right]\right)^t\right) + \sum_{j=2}^{l'}\left(\frac{1}{n_{j-1}^t}\prod_{l=j}^{l'}\left(\frac{c^t}{n_l^t}\left(\sum_{i=1}^{n_l}\mathbb{E}\left[\Xi_{l,i}^2\right]\right)^t\right)\right),$$

where we used the linearity of expectation in both expressions and Jensen's inequality in the last line. Using that $\mathbb{E}[(\Xi_{l,i}(N(0,1),K))^2] = \mathcal{M}$, see Table 4, we get

$$\mathbb{E}\left[\left(A^{(l')}\right)^t\right] \geq \frac{\|\mathfrak{r}^{(0)}\|^{2t}}{n_0^t}(c\mathcal{M})^{tl'} + \sum_{j=2}^{l'}\frac{(c\mathcal{M})^{t(l'-j+1)}}{n_{j-1}^t}. \tag{24}$$

**Variance**    We can use an upper bound on the second moment as an upper bound on the variance.    □

**Remark 22** (Zero bias). Similar results can be obtained for the zero bias case and would result in the same bounds without the second summand. For the proof one would work directly with the vectors $\mathbf{x}^{(l)}$, without defining the vectors $\mathfrak{r}^{(l)}$, and to obtain the equality in distribution one would use Remark 17.

## F    EXPECTED NUMBER OF LINEAR REGIONS

Here we prove the result from Section 5.1.

**Corollary 6** (Value for $C_{\text{grad}}$). *Consider a maxout network with the settings of Section 2. Assume that the biases are independent of the weights but otherwise initialized using any approach. Consider the pre-activation feature $\zeta_{z,k}$ of a unit $z = 1,\ldots,N$. Then, for any $t \in \mathbb{N}$,*

$$\left(\sup_{\mathbf{x}\in\mathbb{R}^{n_0}}\mathbb{E}\left[\|\nabla\zeta_{z,k}(x)\|^t\right]\right)^{\frac{1}{t}} \leq n_0^{-\frac{1}{2}}\max\left\{1,(cK)^{\frac{L-1}{2}}\right\}\exp\left\{\frac{t}{2}\left(\sum_{l=1}^{L-1}\frac{1}{n_lK}+1\right)\right\}.$$

*Proof.* Distribution of $\nabla\zeta_{z,k}$ is the same as the distribution of the gradient with respect to the network input $\nabla\widetilde{\mathcal{N}}_1(\mathbf{x})$ in a maxout network that has a single linear output unit and $\tilde{L} = l(z)$ layers, where $l(z)$ is the depth of a unit $z$. Therefore, we will consider $(\sup_{\mathbf{x}\in\mathbb{R}^{n_0}}\mathbb{E}[\|\nabla\widetilde{\mathcal{N}}_1(\mathbf{x})\|^{2t}])^{1/2t}$. Notice that since $n_{\tilde{L}} = 1$, $\nabla\widetilde{\mathcal{N}}_1(\mathbf{x}) = \mathbf{J}_{\widetilde{\mathcal{N}}}(\mathbf{x})^T = \mathbf{J}_{\widetilde{\mathcal{N}}}(\mathbf{x})^T\mathbf{u}$ for a 1-dimensional vector $\mathbf{u} = (1)$. Hence,

$$\|\nabla\widetilde{\mathcal{N}}_1(\mathbf{x})\| = \sup_{\|\mathbf{u}\|=1,\mathbf{u}\in\mathbb{R}^{n_{\tilde{L}}}}\|\mathbf{J}_{\widetilde{\mathcal{N}}}(\mathbf{x})^T\mathbf{u}\| = \|\mathbf{J}_{\widetilde{\mathcal{N}}}(\mathbf{x})^T\|, \tag{25}$$

where the matrix norm is the spectral norm. Using that a matrix and its transpose have the same spectral norm, (25) equals

$$\|\mathbf{J}_{\widetilde{\mathcal{N}}}(\mathbf{x})\| = \sup_{\|\mathbf{u}\|=1,\mathbf{u}\in\mathbb{R}^{n_0}}\|\mathbf{J}_{\widetilde{\mathcal{N}}}(\mathbf{x})\mathbf{u}\|.$$

Therefore, we need to upper bound

$$\left(\sup_{\mathbf{x}\in\mathbb{R}^{n_0}}\mathbb{E}\left[\left(\sup_{\|\mathbf{u}\|=1,\mathbf{u}\in\mathbb{R}^{n_0}}\|\mathbf{J}_{\widetilde{\mathcal{N}}}(\mathbf{x})\mathbf{u}\|\right)^t\right]\right)^{\frac{1}{t}} \leq \left(\sup_{\mathbf{x}\in\mathbb{R}^{n_0}}\mathbb{E}\left[\left(\sup_{\|\mathbf{u}\|=1,\mathbf{u}\in\mathbb{R}^{n_0}}\|\mathbf{J}_{\widetilde{\mathcal{N}}}(\mathbf{x})\mathbf{u}\|\right)^{2t}\right]\right)^{\frac{1}{2t}},$$

where we used Jensen's inequality.

Now we can use an upper bound on $\mathbb{E}[\|\mathbf{J}_{\widetilde{\mathcal{N}}}(\mathbf{x})\mathbf{u}\|^{2t}]$ from Corollary 2, which holds for any $\mathbf{x},\mathbf{u} \in \mathbb{R}^{n_0},\|\mathbf{u}\|=1$, and thus holds for the suprema. Recalling that $n_{\tilde{L}} = 1$, we get

$$\mathbb{E}\left[\|\mathbf{J}_{\widetilde{\mathcal{N}}}(\mathbf{x})\mathbf{u}\|^{2t}\right] \leq \left(\frac{1}{n_0}\right)^t(cK)^{t(\tilde{L}-1)}\exp\left\{t^2\left(\sum_{l=1}^{\tilde{L}-1}\frac{1}{n_lK}+1\right)\right\}.$$

Hence,

$$\left(\mathbb{E}\left[\|\mathbf{J}_{\widetilde{\mathcal{N}}}(\mathbf{x})\mathbf{u}\|^{2t}\right]\right)^{\frac{1}{2t}} \leq n_0^{-\frac{1}{2}}(cK)^{\frac{\tilde{L}-1}{2}}\exp\left\{\frac{t}{2}\left(\sum_{l=1}^{\tilde{L}-1}\frac{1}{n_lK}+1\right)\right\}.$$

Taking the maximum over $\tilde{L} \in \{1,\ldots,L\}$, the final upper bound is

$$n_0^{-\frac{1}{2}}\max\left\{1,(cK)^{\frac{L-1}{2}}\right\}\exp\left\{\frac{t}{2}\left(\sum_{l=1}^{L-1}\frac{1}{n_lK}+1\right)\right\}.$$

□

Now we provide an updated upper bound on the number of $r$-partial activation regions from Tseran & Montúfar (2021, Theorem 9). In this bound, the case $r = 0$ corresponds to the number of linear regions. For a detailed discussion of the activation regions of maxout networks and their differences from linear regions, see Tseran & Montúfar (2021). Since the proof of Tseran & Montúfar (2021, Theorem 9) only uses $C_{\text{grad}}$ for $t \leq n_0$, we obtain the following statement.

**Theorem 23** (Upper bound on the expected number of partial activation regions). *Consider a maxout network with the settings of Section 2 with $N$ maxout units. Assume that the biases are independent of the weights and initialized so that:*

*1. Every collection of biases has a conditional density with respect to Lebesgue measure given the values of all other weights and biases.*

*2. There exists $C_{\text{bias}} > 0$ so that for any pre-activation features $\zeta_1, \ldots, \zeta_t$ from any neurons, the conditional density of their biases $\rho_{b_1,\ldots,b_t}$ given all the other weights and biases satisfies*

$$\sup_{b_1,\ldots,b_t \in \mathbb{R}} \rho_{b_1,\ldots,b_t}(b_1,\ldots,b_t) \leq C_{\text{bias}}^t.$$

*Fix $r \in \{0,\ldots,n_0\}$. Let $C_{\text{grad}} = n_0^{-1/2} \max\{1, (cK)^{(L-1)/2}\} \exp\{n_0/2(\sum_{l=1}^{L-1} 1/(n_l K) + 1)\}$ and $T = 2^5 C_{\text{grad}} C_{bias}$. Then, there exists $\delta_0 \leq 1/(2C_{\text{grad}} C_{bias})$ such that for all cubes $C \subseteq \mathbb{R}^{n_0}$ with side length $\delta > \delta_0$ we have*

$$\frac{\mathbb{E}[\# \ r\text{-partial activation regions of } \mathcal{N} \text{ in } C]}{\text{vol}(C)} \leq \begin{cases} \binom{rK}{2r}\binom{N}{r}K^{N-r}, & N \leq n_0 \\ \frac{(TKN)^{n_0}\binom{n_0 K}{2n_0}}{(2K)^r n_0!}, & N \geq n_0 \end{cases}.$$

*Here the expectation is taken with respect to the distribution of weights and biases in $\mathcal{N}$. Of particular interest is the case $r = 0$, which corresponds to the number of linear regions.*

## G    EXPECTED CURVE LENGTH DISTORTION

In this section we prove the result from Section 5.2.

Let $M$ be a smooth 1-dimensional curve in $\mathbb{R}^{n_0}$. Fix a smooth unit speed parameterization of $M = \gamma([0,1])$ with $\gamma : \mathbb{R} \to \mathbb{R}^{n_0}, \gamma(\tau) = (\gamma_1(\tau), \ldots, \gamma_{n_0}(\tau))$. Then, parametrization of the curve $\mathcal{N}(M)$ is given by a mapping $\Gamma := \mathcal{N} \circ \gamma, \Gamma : \mathbb{R} \to \mathbb{R}^{n_L}$. Thus, the length of $\mathcal{N}(M)$ is

$$\text{len}(\mathcal{N}(M)) = \int_0^1 \|\Gamma'(\tau)\| d\tau.$$

Notice that the input-output Jacobian of maxout networks is well defined almost everywhere because for any neuron, using that the biases are independent from weights, and the weights are initialized from a continuous distribution, $P(k' = \text{argmax}_{k \in [K]}\{W_{i,k}^{(l)}\mathbf{x}^{(l-1)} + b_{i,k}^{(l)}\}, k'' = \text{argmax}_{k \in [K]}\{W_{i,k}^{(l)}\mathbf{x}^{(l-1)} + b_{i,k}^{(l)}\}) = 0, i = 1, \ldots, n_{L-1}$. Hence, $\Gamma'(\tau) = \mathbf{J}_{\mathcal{N}}(\gamma(\tau))\gamma'(\tau)$, where we used the chain rule, and we can employ the following lemma from Hanin et al. (2021). We state it here without proof which uses Tonelli's theorem, power mean inequality and chain rule.

**Lemma 24** (Connection between the length of the curve and $\|\mathbf{J}_{\mathcal{N}}(\mathbf{x})\mathbf{u}\|$, Hanin et al. 2021, Lemma C.1). *For any integer $t \geq 0$,*

$$\mathbb{E}\left[\text{len}(\mathcal{N}(M))^t\right] \leq \int_0^1 \mathbb{E}\left[\|\mathbf{J}_{\mathcal{N}}(\gamma(\tau))\gamma'(\tau)\|^t\right] d\tau = \mathbb{E}\left[\|\mathbf{J}_{\mathcal{N}}(\mathbf{x})\mathbf{u}\|^t\right],$$

*where $\mathbf{u} \in \mathbb{R}^{n_0}$ is a unit vector.*

Now we are ready to proof Corollary 7.

**Corollary 7** (Expected curve length distortion). *Consider a maxout network with the settings of Section 2. Assume that the biases are independent of the weights but otherwise initialized using any*

*approach. Let $M$ be a smooth $1$-dimensional curve of unit length in $\mathbb{R}^{n_0}$. Then, the following upper bounds on the moments of $\operatorname{len}(\mathcal{N}(M))$ hold:*

$$\mathbb{E}\left[\operatorname{len}(\mathcal{N}(M))\right] \leq \left(\frac{n_L}{n_0}\right)^{\frac{1}{2}}(c\mathcal{L})^{\frac{L-1}{2}}, \qquad \operatorname{Var}\left[\operatorname{len}(\mathcal{N}(M))\right] \leq \frac{n_L}{n_0}(c\mathcal{L})^{L-1},$$

$$\mathbb{E}\left[\operatorname{len}(\mathcal{N}(M))^t\right] \leq \left(\frac{n_L}{n_0}\right)^{\frac{t}{2}}(cK)^{\frac{t(L-1)}{2}}\exp\left\{\frac{t^2}{2}\left(\sum_{l=1}^{L-1}\frac{1}{n_l K}+\frac{1}{n_L}\right)\right\},$$

*where $\mathcal{L}$ is a constant depending on $K$, see Table 4 in Appendix C for values for $K = 2, \ldots, 10$.*

*Proof.* By Lemma 24,

$$\mathbb{E}\left[\operatorname{len}(\mathcal{N}(M))^t\right] \leq \mathbb{E}\left[\|\mathbf{J}_{\mathcal{N}}(\mathbf{x})\mathbf{u}\|^t\right] \leq \left(\mathbb{E}\left[\|\mathbf{J}_{\mathcal{N}}(\mathbf{x})\mathbf{u}\|^{2t}\right]\right)^{\frac{1}{2}},$$

where we used Jensen's inequality to obtain the last upper bound. Hence, using Corollary 2, we get the following upper bounds on the moments on the length of the curve.

**Mean**

$$\mathbb{E}\left[\operatorname{len}(\mathcal{N}(M))\right] \leq \left(\mathbb{E}\left[\|\mathbf{J}_{\mathcal{N}}(\mathbf{x})\mathbf{u}\|^2\right]\right)^{\frac{1}{2}} \leq \left(\frac{n_L}{n_0}\right)^{\frac{1}{2}}(c\mathcal{L})^{\frac{L-1}{2}}.$$

**Variance**

$$\operatorname{Var}\left[\operatorname{len}(\mathcal{N}(M))\right] \leq \mathbb{E}\left[\operatorname{len}(\mathcal{N}(M))^2\right] \leq \frac{n_L}{n_0}(c\mathcal{L})^{L-1}.$$

**Moments of the order $t \geq 3$**

$$\mathbb{E}\left[\operatorname{len}(\mathcal{N}(M))^t\right] \leq \left(\mathbb{E}\left[\|\mathbf{J}_{\mathcal{N}}(\mathbf{x})\mathbf{u}\|^{2t}\right]\right)^{\frac{1}{2}} \leq \left(\frac{n_L}{n_0}\right)^{\frac{t}{2}}(cK)^{\frac{t(L-1)}{2}}\exp\left\{\frac{t^2}{2}\left(\sum_{l=1}^{L-1}\frac{1}{n_l K}+\frac{1}{n_L}\right)\right\}.$$

$\square$

## H  NTK

Here we prove the results from Section 5.3.

**Corollary 9** (On-diagonal NTK)**.** *Consider a maxout network with the settings of Section 2. Assume that $n_L = 1$ and that the biases are initialized to zero and are not trained. Assume that $\mathcal{S} \leq c \leq \mathcal{L}$, where the constants $\mathcal{S}, \mathcal{L}$ are as specified in Table 4. Then,*

$$\|\mathfrak{r}^{(0)}\|^2\frac{(c\mathcal{S})^{L-2}}{n_0}P \leq \mathbb{E}[K_{\mathcal{N}}(\mathbf{x}, \mathbf{x})] \leq \|\mathfrak{r}^{(0)}\|^2\frac{(c\mathcal{L})^{L-2}\mathcal{M}^{L-1}}{n_0}P,$$

$$\mathbb{E}[K_{\mathcal{N}}(\mathbf{x}, \mathbf{x})^2] \leq 2PP_W(cK)^{2(L-2)}\frac{\|\mathfrak{r}^{(0)}\|^4}{n_0^2}\exp\left\{\sum_{j=1}^{L-1}\frac{4}{n_j K}+4\right\},$$

*where $P = \sum_{l=0}^{L-1} n_l$, $P_W = \sum_{l=0}^{L} n_l n_{l-1}$, and $\mathcal{M}$ is as specified in Table 4.*

*Proof.* Under the assumption that biases are not trained, on-diagonal NTK of a maxout network is

$$K_{\mathcal{N}}(\mathbf{x}, \mathbf{x}) = \sum_{l=1}^{L}\sum_{i=1}^{n_l}\sum_{k=1}^{K}\sum_{j=1}^{n_{l-1}}\left(\frac{\partial\mathcal{N}}{\partial W_{i,k,j}^{(l)}}(\mathbf{x})\right)^2.$$

Since in maxout network for all $k$-s except $k = k' = \operatorname{argmax}_{k \in [K]}\{W_{i,k}^{(l)}\mathbf{x}^{(l-1)} + \mathbf{b}_{i,k}^{(l)}\}$ the derivatives with respect to the weights and biases are zero, on-diagonal NTK equals

$$K_{\mathcal{N}}(\mathbf{x}, \mathbf{x}) = \sum_{l=1}^{L}\sum_{i=1}^{n_l}\sum_{j=1}^{n_{l-1}}\left(\frac{\partial\mathcal{N}}{\partial W_{i,k',j}^{(l)}}(\mathbf{x})\right)^2.$$

Notice that since we assumed a continuous distribution over the network weights and the biases are zero, the partial derivatives are defined everywhere except for the set of measure zero.

**Part I. Kernel mean $\mathbb{E}[K_{\mathcal{N}}(\mathbf{x}, \mathbf{x})]$**  Firstly, using the chain rule, a partial derivative with respect to the network weight is

$$\frac{\partial \mathcal{N}}{\partial W_{i,k',j}^{(l)}}(\mathbf{x}) = \frac{\partial \mathcal{N}}{\partial \mathbf{x}_i^{(l)}}(\mathbf{x})\mathbf{x}_j^{(l-1)} = \mathbf{J}_{\mathcal{N}}(\mathbf{x}^{(l)})\mathbf{e}_i\mathbf{x}_j^{(l-1)}.$$

Recall that we assumed $n_L = 1$. Therefore, we need to consider $(\partial \mathcal{N}(\mathbf{x})\partial W_{i,k',j}^{(l)})^2 = \|\mathbf{J}_{\mathcal{N}}(\mathbf{x}^{(l)})\mathbf{u}\|^2 (\mathbf{x}_j^{(l-1)})^2$, where $\mathbf{u} = \mathbf{e}_i$. Combining Theorem 1 and Corollary 21 in combination with Remark 22 for the zero-bias case and using the independence of the random variables in the expressions,

$$
\mathbb{E}\left[\|\mathbf{J}_{\mathcal{N}}(\mathbf{x}^{(l)})\mathbf{u}\|^2 \left(\mathbf{x}_j^{(l-1)}\right)^2\right] \leq_{so} \mathbb{E}\left[\frac{1}{n_l}\chi_{n_L}^2 \prod_{j=l}^{L-1}\frac{c}{n_j}\sum_{i=1}^{n_j}\Xi_{j,i}(\chi_1^2, K)\right]
$$
$$
\cdot \mathbb{E}\left[\|\mathfrak{x}^{(0)}\|^2\frac{1}{n_0}\prod_{j=1}^{l-1}\left(\frac{c}{n_j}\sum_{i=1}^{n_j}\Xi_{j,i}(N(0,1), K)^2\right)\right], \tag{26}
$$

where we treat the $(l-1)$th layer as if it has one unit when we use the normalized activation length result. Then, using Corollaries 2 and 5,

$$\mathbb{E}\left[\|\mathbf{J}_{\mathcal{N}}(\mathbf{x}^{(l)})\mathbf{u}\|^2 \left(\mathbf{x}_j^{(l-1)}\right)^2\right] \leq \|\mathfrak{x}^{(0)}\|^2\frac{c^{L-2}}{n_0 n_l}\mathcal{L}^{L-l-1}\mathcal{M}^{l-1}.$$

Taking the sum, we get

$$
\mathbb{E}[K_{\mathcal{N}}(\mathbf{x}, \mathbf{x})] = \mathbb{E}[K_W] = \mathbb{E}\left[\sum_{l=1}^{L}\sum_{i=1}^{n_l}\sum_{j=1}^{n_{l-1}}\left(\frac{\partial \mathcal{N}}{\partial W_{i,k',j}^{(l)}}(\mathbf{x})\right)^2\right]
$$
$$
\leq \sum_{l=1}^{L}\sum_{i=1}^{n_l}\sum_{j=1}^{n_{l-1}}\|\mathfrak{x}^{(0)}\|^2\frac{c^{L-2}}{n_0 n_l}\mathcal{L}^{L-l-1}\mathcal{M}^{l-1} \leq \|\mathfrak{x}^{(0)}\|^2\frac{(c\mathcal{L})^{L-2}\mathcal{M}^{L-1}}{n_0}P,
$$

where $P = \sum_{l=0}^{L-1} n_l$ denotes the number of neurons in the network up to the last layer, but including the input neurons. Here we used that for $K \geq 2$, both $\mathcal{L}, \mathcal{M} \geq 1$, see Table 4. Similarly,

$$\mathbb{E}[K_W] \geq \sum_{l=1}^{L}\sum_{i=1}^{n_l}\sum_{j=1}^{n_{l-1}}\|\mathfrak{x}^{(0)}\|^2\frac{c^{L-2}}{n_0 n_l}\mathcal{S}^{L-l-1}\mathcal{M}^{l-1} \geq \|\mathfrak{x}^{(0)}\|^2\frac{(c\mathcal{S})^{L-2}}{n_0}P.$$

Here we used that for $K \geq 2$, $\mathcal{S} \leq 1$ and $\mathcal{M} \geq 1$, see Table 4.

**Part II. Second moment $\mathbb{E}[K_{\mathcal{N}}(\mathbf{x}, \mathbf{x})^2]$**  Using equation (26) with Corollaries 2 and 5,

$$
\mathbb{E}\left[\left(\frac{\partial \mathcal{N}}{\partial W_{i,k',j}^{(l)}}(\mathbf{x})\right)^4\right] = \mathbb{E}\left[\|\mathbf{J}_{\mathcal{N}}(\mathbf{x}^{(l)})\mathbf{u}\|^4 \left(\mathbf{x}_j^{(l-1)}\right)^4\right]
$$
$$
\leq_{so} \mathbb{E}\left[\left(\frac{1}{n_l}\chi_{n_L}^2\prod_{j=l}^{L-1}\frac{c}{n_j}\sum_{i=1}^{n_j}\Xi_{j,i}(\chi_1^2, K)\right)^2\right]\mathbb{E}\left[\left(\|\mathfrak{x}^{(0)}\|^2\frac{1}{n_0}\prod_{j=1}^{l-1}\left(\frac{c}{n_j}\sum_{i=1}^{n_j}\Xi_{j,i}(N(0,1), K)^2\right)\right)^2\right]
$$
$$
\leq 2(cK)^{2(L-2)}\frac{\|\mathfrak{x}^{(0)}\|^4}{n_l^2 n_0^2}\exp\left\{4\left(\sum_{j=1}^{L-1}\frac{1}{n_j K} - \frac{1}{n_l K} + 1\right)\right\}. \tag{27}
$$

Notice that all summands are non-negative. Then, using AM-QM inequality,

$$
\begin{aligned}
\mathbb{E}[K_{\mathcal{N}}(\mathbf{x}, \mathbf{x})^2] &= \mathbb{E}\left[\left(\sum_{l=1}^{L}\sum_{i=1}^{n_l}\sum_{j=1}^{n_{l-1}}\left(\frac{\partial \mathcal{N}}{\partial W_{i,k',j}^{(l)}}(\mathbf{x})\right)^2\right)^2\right] \\
&\leq P_W \sum_{l=1}^{L}\sum_{i=1}^{n_l}\sum_{j=1}^{n_{l-1}}\mathbb{E}\left[\left(\frac{\partial \mathcal{N}}{\partial W_{i,k',j}^{(l)}}(\mathbf{x})\right)^4\right] \\
&\leq P_W \sum_{l=1}^{L}\sum_{i=1}^{n_l}\sum_{j=1}^{n_{l-1}} 2(cK)^{2(L-2)}\frac{\|\mathbf{r}^{(0)}\|^4}{n_l^2 n_0^2}\exp\left\{4\left(\sum_{j=1}^{L-1}\frac{1}{n_j K} - \frac{1}{n_l K} + 1\right)\right\} \\
&\leq 2PP_W(cK)^{2(L-2)}\frac{\|\mathbf{r}^{(0)}\|^4}{n_0^2}\exp\left\{\sum_{j=1}^{L-1}\frac{4}{n_j K} + 4\right\},
\end{aligned}
$$

where $P_W = \sum_{l=0}^{L} n_l n_{l-1}$ denotes the number of all weights in the network. $\qquad\square$

## I    EXPERIMENTS

### I.1    EXPERIMENTS WITH SGD AND ADAM FROM SECTION 6

In this subsection we provide more details on the experiments presented in Section 6. The implementation of the key routines is available at https://anonymous.4open.science/r/maxout_expected_gradient-68BD. Experiments were implemented in Python using TensorFlow (Martín Abadi et al., 2015), numpy (Harris et al., 2020) and mpi4py (Dalcin et al., 2011). The plots were created using matplotlib (Hunter, 2007). We conducted all training experiments from Section 6 on a GPU cluster with nodes having 4 Nvidia A100 GPUs with 40 GB of memory. The most extensive experiments were running for one day on one GPU. Experiment in Figure 2 was run on a CPU cluster that uses Intel Xeon IceLakeSP processors (Platinum 8360Y) with 72 cores per node and 256 GB RAM. All other experiments were executed on the laptop ThinkPad T470 with Intel Core i5-7200U CPU with 16 GB RAM.

**Training experiments**    Now we discuss the training experiments. We use MNIST (LeCun & Cortes, 2010), Iris (Fisher, 1936), Fashion MNIST (Xiao et al., 2017), SVHN (Netzer et al., 2011), CIFAR-10 and CIFAR-100 datasets (Krizhevsky et al., 2009). All maxout networks have the maxout rank $K = 5$. Weights are sampled from $N(0, c/\text{fan-in})$ in fully-connected networks and $N(0, c/(k^2 \cdot \text{fan-in}))$, where $k$ is the kernel size, in CNNs. The biases are initialized to zero. ReLU networks are initialized using He approach (He et al., 2015), meaning that $c = 2$. All results are averaged over 4 runs. We do not use any weight normalization techniques, such as batch normalization (Ioffe & Szegedy, 2015). We performed the dataset split into training, validation and test dataset and report the accuracy on the test set, while the validation set was used only for picking the hyper-parameters and was not used in training. The mini-batch size in all experiments is 32. The number of training epochs was picked by observing the training set loss and choosing the number of epochs for which the loss has converged. The exception is the SVHN dataset, for which we observe the double descent phenomenon and stop training after 150 epochs.

**Network architecture**    Fully connected networks have 21 layers. Specifically, their architecture is

$$[5\times\text{fc64}, \ 5\times\text{fc32}, \ 5\times\text{fc16}, \ 5\times\text{fc8}, \ \text{out}],$$

where "5×fc64" means that there are 5 fully-connected layers with 64 neurons, and "out" stands for the output layer that has the number of neurons equal to the number of classes in a dataset. CNNs have a VGG-19-like architecture (Simonyan & Zisserman, 2015) with 20 or 16 layers, depending the input size. The 20-layer architecture is

$$
\begin{aligned}
[2\times\text{conv64}, \ \text{mp}, \ 2\times\text{conv128}, \ \text{mp}, \ 4\times\text{conv256}, \ \text{mp}, \ 4\times\text{conv512}, \ \text{mp}, \ 4\times\text{conv512}, \ \text{mp}, \\
2\times\text{fc4096}, \ \text{fc1000}, \ \text{out}],
\end{aligned}
$$

where "conv64" stands for a convolutional layer with $64$ neurons and "mp" for a max-pooling layer. The kernel size in all convolutional layers is $3 \times 3$. Max-pooling uses $2 \times 2$ pooling windows with stride $2$. Such architecture is used for datasets with the images that have the side length greater or equal to 32: CIFAR-10, CIFAR-100 and SVHN. The 16-layer architecture is used for images with the smaller image size: MNIST and Fashion MNIST. This architecture does not have the last convolutional block of the 20-layer version. Concretely, it has the following layers:

$$[2\times\text{conv64, mp, } 2\times\text{conv128, mp, } 4\times\text{conv256, mp, } 4\times\text{conv512, mp, } 2\times\text{fc4096, fc1000, out}],$$

**Max-pooling initialization**  To account for the maximum in max-pooling layers, a maxout layer appearing after a max-pooling layer is initialized as if its maxout rank was $K \times m^2$, where $m^2$ is the max-pooling window size. The reason for this is that the outputs of a computational block consisting of a max-pooling window and a maxout layer are taking maxima over $K \times m^2$ linear functions, $\max\{W_1 \max\{\mathbf{x}_1, \ldots, \mathbf{x}_{m^2}\} + \mathbf{b}_1, \ldots, W_K \max\{\mathbf{x}_1, \ldots, \mathbf{x}_{m^2}\} + \mathbf{b}_K\} = \max\{f_1, \ldots, f_{Km^2}\}$, where the $f_i$ are $Km^2$ affine functions. Therefore, we initialize the layers that follow max-pooling layers using the criterion for maxout rank $m^2 \times K$ instead of $K$. In our experiments, $K = 5$, $m = 2$, and $m^2 \times K = 20$. Hence, for such layers, we use the constant $c = 1/M = 0.26573$, where $M$ is computed for $K = 20$ using the formula from Remark 14 in Appendix C. All other layers that do not follow max-pooling layers are initialized as suggested in Section 4. We observe that max-pooling initialization often leads to slightly higher accuracy.

**Data augmentation**  There is no data augmentation for fully connected networks. For convolutional networks, for MNIST, Fashion MNIST and SVHN datasets we perform random translation, rotation and zoom of the input images. For CIFAR-10 and CIFAR-100, we additionally apply a random horizontal flip.

**Learning rate decay**  In all experiments, we use the learning rate decay and choose the optimal initial learning rate for all network and initialization types based on their accuracy on the validation dataset. The learning rate was halved every $n$th epoch. For SVHN, $n = 10$, and for all other datasets, $n = 100$.

**SGD with momentum**  We use SGD with Nesterov momentum, with the momentum value of $0.9$. Specific dataset settings are the following.

- MNIST (fully-connected networks). Networks are trained for 600 epochs. The learning rate is halved every 100 epochs. Learning rates: maxout networks with maxout initialization: $0.002$, maxout networks with $c = 0.1$: $0.002$, maxout networks with $c = 2$: $2 \times 10^{-7}$, ReLU networks: $0.002$.

- Iris. Networks are trained for 500 epochs. The learning rate is halved every 100 epochs. Learning rates: maxout networks with maxout initialization: $0.01$, maxout networks with $c = 0.1$: $0.01$, maxout networks with $c = 2$: $4 \times 10^{-8}$, ReLU networks: $0.005$.

- MNIST (convolutional networks). Networks are trained for 800 epochs. The learning rate is halved every 100 epochs. Learning rates: maxout networks with maxout initialization: $0.009$, maxout networks with max-pooling initialization: $0.009$, maxout networks with $c = 0.1$: $0.009$, maxout networks with $c = 2$: $8 \times 10^{-6}$, ReLU networks: $0.01$.

- Fashion MNIST. Networks are trained for 800 epochs. The learning rate is halved every 100 epochs. Learning rates: maxout networks with maxout initialization: $0.004$, maxout networks with max-pooling initialization: $0.006$, maxout networks with $c = 0.1$: $0.4$, maxout networks with $c = 2$: $5 \times 10^{-6}$, ReLU networks: $0.01$.

- CIFAR-10. Networks are trained for 1000 epochs. The learning rate is halved every 100 epochs. Learning rates: maxout networks with maxout initialization: $0.004$, maxout networks with max-pooling initialization: $0.005$, maxout networks with $c = 0.1$: $0.5$, maxout networks with $c = 2$: $8 \times 10^{-8}$, ReLU networks: $0.009$.

- CIFAR-100. Networks are trained for 1000 epochs. The learning rate is halved every 100 epochs. Learning rates: maxout networks with maxout initialization: $0.002$, maxout networks with max-pooling initialization: $0.002$, maxout networks with $c = 0.1$: $0.002$, maxout networks with $c = 2$: $8 \times 10^{-5}$, ReLU networks: $0.006$.

Table 5: Accuracy on the test set for maxout networks trained using SGD with Nesterov momentum for values of $c$ less than or equal to $c = 0.55555$, the value suggested in Section 4.

| VALUE OF c | 0.01 | 0.07 | 0.05 | 0.1 | 0.2 | 0.3 | 0.4 | 0.5 | 0.55555 |
|---|---|---|---|---|---|---|---|---|---|
| | FULLY-CONNECTED | | | | | | | | |
| MNIST | $11.35^{\pm0.00}$ | $11.35^{\pm0.00}$ | $11.35^{\pm0.00}$ | $11.35^{\pm0.00}$ | $11.35^{\pm0.00}$ | $97.63^{\pm0.16}$ | $97.89^{\pm0.12}$ | $97.82^{\pm0.09}$ | $97.92^{\pm0.18}$ |
| Iris | $30^{\pm0.00}$ | $30^{\pm0.00}$ | $31.67^{\pm2.89}$ | $30^{\pm0.00}$ | $30^{\pm0.00}$ | $60.83^{\pm30.86}$ | $85^{\pm10.67}$ | $92.5^{\pm1.44}$ | $90.83^{\pm3.63}$ |
| | CONVOLUTIONAL | | | | | | | | |
| MNIST | $11.35^{\pm0.00}$ | $11.35^{\pm0.00}$ | $11.35^{\pm0.00}$ | $11.35^{\pm0.00}$ | $99.56^{\pm0.03}$ | $99.55^{\pm0.02}$ | $99.6^{\pm0.03}$ | $99.56^{\pm0.05}$ | $99.57^{\pm0.07}$ |
| CIFAR-10 | $10^{\pm0.00}$ | $10^{\pm0.00}$ | $10^{\pm0.00}$ | $10^{\pm0.00}$ | $10^{\pm0.00}$ | $90.97^{\pm0.11}$ | $91.15^{\pm0.07}$ | $91.33^{\pm0.13}$ | $91.4^{\pm0.22}$ |

- SVHN. Networks are trained for 150 epochs. The learning rate is halved every 10 epochs. Learning rates: maxout networks with maxout initialization: 0.005, maxout networks with max-pooling initialization: 0.005, maxout networks with $c = 0.1$: 0.005, maxout networks with $c = 2$: $7 \times 10^{-5}$, ReLU networks: 0.005.

**Adam** We use Adam optimizer Kingma & Ba (2015) with default TensorFlow parameters $\beta_1 = 0.9, \beta_2 = 0.999$. Specific dataset settings are the following.

- MNIST (fully-connected networks). Networks are trained for 600 epochs. The learning rate is halved every 100 epochs. Learning rates: maxout networks with maxout initialization: 0.0008, maxout networks with $c = 2$: 0.0007, ReLU networks: 0.0008.

- MNIST (convolutional networks). Networks are trained for 800 epochs. The learning rate is halved every 100 epochs. Learning rates: maxout networks with maxout initialization: 0.0001, maxout networks with max-pooling initialization: 0.00006, maxout networks with $c = 2$: 0.00004, ReLU networks: 0.00009.

- Fashion MNIST. Networks are trained for 1000 epochs. The learning rate is halved every 100 epochs. Learning rates: maxout networks with maxout initialization: 0.00007, maxout networks with max-pooling initialization: 0.00008, maxout networks with $c = 2$: 0.00005, ReLU networks: 0.0002.

- CIFAR-10. Networks are trained for 1000 epochs. The learning rate is halved every 100 epochs. Learning rates: maxout networks with maxout initialization: 0.00009, maxout networks with max-pooling initialization: 0.00009, maxout networks with $c = 2$: 0.00005, ReLU networks: 0.0001.

- CIFAR-100. Networks are trained for 1000 epochs. The learning rate is halved every 100 epochs. Learning rates: maxout networks with maxout initialization: 0.00008, maxout networks with max-pooling initialization: 0.00009, maxout networks with $c = 2$: 0.00005, ReLU networks: 0.00009.

## I.2 ABLATION ANALYSIS

Tables 5 and 6 show the results of the additional experiments that use SGD with Nesterov momentum for more values of $c$ and $K = 5$. From this, we see that the recommended value of $c$ from Section 4 closely matches the empirical optimum value of $c$. Note that here we have fixed the learning rate across choices of $c$. More specifically, the following learning rates were used for the experiments with different datasets. MNIST with fully-connected networks: 0.002; Iris: 0.01; MNIST with convolutional networks: 0.009; CIFAR-10: 0.004.

## I.3 BATCH NORMALIZATION

Tables 7 and 8 report test accuracy for maxout networks with batch normalization trained using SGD with Nesterov momentum for various values of $c$. The implementation of the experiments is similar to that described in Section I.1, except for the following differences: The networks use batch normalization after each layer with activations; The width of the last fully connected layer is 100, and all other layers of the convolutional networks are 8 times narrower; The learning rate is fixed at 0.01 for all experiments. We use the default batch normalization parameters from TensorFlow. Specifically, the momentum equals 0.99 and $\epsilon = 0.001$. We observe that our initialization strategy is still beneficial when training with batch normalization.

Table 6: Accuracy on the test set for maxout networks trained using SGD with Nesterov momentum for values of $c$ greater than or equal to $c = 0.55555$, the value suggested in Section 4.

| VALUE OF c | 0.55555 | 0.6 | 0.7 | 0.8 | 0.9 | 1 | 1.5 | 2 | 10 |
|---|---|---|---|---|---|---|---|---|---|
| | FULLY-CONNECTED | | | | | | | | |
| MNIST | $97.92^{\pm 0.18}$ | $97.77^{\pm 0.17}$ | $97.91^{\pm 0.11}$ | $75.82^{\pm 38.12}$ | $75.94^{\pm 38.18}$ | $97.89^{\pm 0.10}$ | $9.8^{\pm 0.00}$ | $9.8^{\pm 0.00}$ | $9.8^{\pm 0.00}$ |
| Iris | $90.83^{\pm 3.63}$ | $90.83^{\pm 1.44}$ | $90^{\pm 0.00}$ | $30^{\pm 0.00}$ | $30^{\pm 0.00}$ | $30^{\pm 0.00}$ | $30^{\pm 0.00}$ | $30^{\pm 0.00}$ | $30^{\pm 0.00}$ |
| | CONVOLUTIONAL | | | | | | | | |
| MNIST | $99.57^{\pm 0.07}$ | $99.59^{\pm 00.02}$ | $54.69^{\pm 44.89}$ | $9.8^{\pm 0.00}$ | $9.8^{\pm 0.00}$ | $9.8^{\pm 0.00}$ | $9.8^{\pm 0.00}$ | $9.8^{\pm 0.00}$ | $9.8^{\pm 0.00}$ |
| CIFAR-10 | $91.4^{\pm 0.22}$ | $91.69^{\pm 0.25}$ | $50.83^{\pm 40.83}$ | $10^{\pm 0.00}$ | $10^{\pm 0.00}$ | $10^{\pm 0.00}$ | $10^{\pm 0.00}$ | $10^{\pm 0.00}$ | $10^{\pm 0.00}$ |

Table 7: Accuracy on the test set for maxout networks with batch normalization trained using SGD with Nesterov momentum for values of $c$ less than or equal $c = 0.55555$, the value suggested in Section 4. Observe that the recommended value of $c$ from our theory closely matches the empirical optimum value of $c$.

| VALUE OF c | $10^{-6}$ | $10^{-5}$ | $10^{-4}$ | 0.001 | 0.01 | 0.1 | 0.55555 |
|---|---|---|---|---|---|---|---|
| | CONVOLUTIONAL | | | | | | |
| MNIST | $11.35^{\pm 0.00}$ | $11.35^{\pm 0.00}$ | $99.33^{\pm 0.09}$ | $99.35^{\pm 0.05}$ | $99.32^{\pm 0.05}$ | $99.36^{\pm 0.04}$ | $99.41^{\pm 0.07}$ |
| CIFAR-10 | $10^{\pm 0.00}$ | $10^{\pm 0.00}$ | $10^{\pm 0.00}$ | $74.85^{\pm 3.29}$ | $75.72^{\pm 4.94}$ | $77.16^{\pm 1.84}$ | $77.68^{\pm 1.07}$ |

Table 8: Accuracy on the test set for maxout networks with batch normalization trained using SGD with Nesterov momentum for values of $c$ greater than or equal to $c = 0.55555$, the value suggested in Section 4. Observe that the recommended value of $c$ from our theory closely matches the empirical optimum value of $c$.

| VALUE OF c | 0.55555 | 1 | 10 | 100 | 1000 | $10^4$ | $10^5$ | $10^6$ |
|---|---|---|---|---|---|---|---|---|
| | CONVOLUTIONAL | | | | | | | |
| MNIST | $99.41^{\pm 0.07}$ | $99.39^{\pm 0.04}$ | $99.35^{\pm 0.02}$ | $98.83^{\pm 0.07}$ | $97.69^{\pm 0.31}$ | $95.11^{\pm 1.40}$ | $93.09^{\pm 1.88}$ | $87.63^{\pm 1.86}$ |
| CIFAR-10 | $77.68^{\pm 1.07}$ | $79.26^{\pm 0.76}$ | $75.82^{\pm 1.05}$ | $66.23^{\pm 1.69}$ | $50.97^{\pm 2.28}$ | $43.81^{\pm 1.80}$ | $39.27^{\pm 2.73}$ | $40.27^{\pm 0.92}$ |

## I.4 COMPARISON OF MAXOUT AND RELU NETWORKS IN TERMS OF THE NUMBER OF PARAMETERS

We should point out that what is a fair comparison is not as straightforward as matching the parameter count. In particular, wider networks have the advantage of having a higher dimensional representation. A fully connected network will not necessarily perform as well as a convolutional network with the same number of parameters, and a deep and narrow network will not necessarily perform as well as a wider and shallower network with the same number of parameters.

Nevertheless, to add more details to the results, we perform experiments using ReLU networks that have as many parameters as maxout networks. See Tables 9 and 10 for results. We modify network architectures described in Section I.1 for these experiments in the following way. In the first experiment, we use fully connected ReLU networks 5 times wider than maxout networks. For convolutional networks, however, the resulting CNNs with ReLU activations would be extremely wide, so we only made it 2 times wider. In our setup, a 5 times wider CNN network would need to be trained for longer than 24 hours, which is difficult in our experiment environment. Maxout networks only required a much shorter time of around 10 hours, which indicates possible benefits in some cases. In the second experiment, we consider ReLU networks that are 5 times deeper than maxout networks. More specifically. they have the following architecture: [25×fc64, 25×fc32, 25×fc16, 25×fc8, out]. As expected, wider networks do better. On the other hand, deeper ReLU networks of the same width do much worse than maxout networks.

Table 9: Accuracy on the test set for networks trained using SGD with Nesterov momentum. Fully-connected ReLU networks are 5 times wider than fully-connected maxout networks. Convolutional ReLU networks are 2 times wider than convolutional maxout networks. All networks have the same number of layers.

| | **MAXOUT** Maxout init | **RELU** He init |
|---|---|---|
| **VALUE OF c** | 0.55555 | 2 |
| | FULLY-CONNECTED | |
| MNIST | $97.8^{\pm 0.15}$ | $\mathbf{98.11}^{\pm 0.02}$ |
| Iris | $91.67^{\pm 3.73}$ | $\mathbf{92.5}^{\pm 2.76}$ |
| | CONVOLUTIONAL | |
| MNIST | $\mathbf{99.59}^{\pm 0.04}$ | $99.54^{\pm 0.02}$ |

Table 10: Accuracy on the test set for networks trained using SGD with Nesterov momentum. Fully-connected ReLU networks are 5 times deeper than fully-connected maxout networks but have the same width.

| | **MAXOUT** Maxout init | **RELU** He init |
|---|---|---|
| **VALUE OF c** | 0.55555 | 2 |
| | FULLY-CONNECTED | |
| MNIST | $\mathbf{97.8}^{\pm 0.15}$ | $63.47^{\pm 33.32}$ |

