# OpenReview forum: "Expected Gradients of Maxout Networks and Consequences to Parameter Initialization"
_ICLR.cc/2023/Conference — Submitted to ICLR 2023_

### Official Review · Reviewer_3PQc · 2022-10-25

**Confidence:** 2
**Correctness:** 4
**Technical Novelty And Significance:** 3
**Empirical Novelty And Significance:** 3
**Recommendation:** 8

**Clarity, Quality, Novelty And Reproducibility:**

A minor comment: I appreciate the rigour of the paper, but at times the amount of notation makes the paper hard to read. I would encourage authors to occasionally repeat the definition of a variable again, so that the reader isn't expected to remember each variable for the entire text.

For example, the text introduces $\mathcal{M}$ in passing just before equation 2. The rest of the text heavily uses this variable, but never mentions it by name again. In particular, this makes it hard for a practitioner who skipped to section 4 to understand what is going on.

All in all the paper is well-written and structured though. The appendix is very thorough, and all the details are provided to check the proofs and reproduce the experimental results.

**Strength And Weaknesses:**

Large parts of this paper are outside of my domain of expertise, so I am unable to judge certain aspects confidently.

This looks like a strong paper. The theoretical results seem extensions to existing proofs for ReLU networks (e.g., Hanin), but the extensions seem non-trivial. The theoretical results are validated thoroughly using numerical experiments, and it is nice to see that concrete and simple recommendations are made for practitioners in section 4.

**Summary Of The Paper:**

This paper looks at maxout networks which are randomly initialized with parameters drawn from a Gaussian distribution. It then looks at the directional derivative (i.e., the input-output Jacobian times a fixed vector). The authors derive a variety of statistics for this derivative (bounds, distribution, expected moments). Based on their findings, they suggest a good parameter initialization for wide maxout networks. They experimentally validate that their proposed parameter initialization works well.

**Summary Of The Review:**

Outside of my domain of expertise, but looks like a well-written paper with a significant technical contribution (although quite directly building on previous work done for ReLU units).

---

> ### Author Response · Authors · 2022-11-18
> **Response to Reviewer 3PQc**
>
> Thank you for the valuable and positive feedback! We appreciate your comments on the non-triviality of our theoretical results, thorough numerical experiments, and the value of the recommendations we provide for practitioners. We address your minor comment below and remain attentive to any additional comments or discussion.
>
> > C1: A minor comment: I appreciate the rigour of the paper, but at times the amount of notation makes the paper hard to read. I would encourage authors to occasionally repeat the definition of a variable again, so that the reader isn't expected to remember each variable for the entire text.
> For example, the text introduces $\mathcal{M}$ in passing just before equation 2. The rest of the text heavily uses this variable, but never mentions it by name again. In particular, this makes it hard for a practitioner who skipped to section 4 to understand what is going on.
>
> Thank you for pointing this out. In the revision, we will make sure to go through the paper and remind the definitions of the variables when they are used in the later parts of the text.

---

> ### Comment · Area_Chair_ZGsw · 2022-12-13
> **Notice**
>
> Dear Reviewer,
>
> If you do not want to update your score, please at least acknowledge that you have already read the rebuttal.

---

### Official Review · Reviewer_RjF3 · 2022-10-26

**Confidence:** 2
**Correctness:** 3
**Technical Novelty And Significance:** 3
**Empirical Novelty And Significance:** 3
**Recommendation:** 6

**Clarity, Quality, Novelty And Reproducibility:**

## Clarity (W1)

I have a few questions about the discussion of Eq. (1) on page 3. First, what is the motivation behind the decomposition $\langle \mathcal{L}(\mathcal{N}(x)), J_\mathcal{N}(x_i)^{(l)}\rangle = C(x, W) ||J_\mathcal{N}(x^{(l)}) u ||$? You use $u = e_i$, so $||J_\mathcal{N}(x^{(l)}) u || = ||J_\mathcal{N}(x_i)^{(l)}||$, correct? Then, the term $C(x, W)$ is given by $\langle \mathcal{L}(\mathcal{N}(x)), J_\mathcal{N}(x_i)^{(l)}\rangle / ||J_\mathcal{N}(x_i)^{(l)}||$. In other words, you multiply and divide by $||J_\mathcal{N}(x^{(l)}) u ||$, but then this term is the center of your analysis.

Throughout the paper, you seem to assume that the term $||J_\mathcal{N}(x^{(l)}) u ||$ is the most important term in the Eq. (1).
Why is this justified, given that it was introduced artificially? Why can we ignore $C(x, W)$?

Then, you say that the other term of interest is $A^(l) = ||x^{(l)}||^2 / n_l$. Where did this term come from? Where does it appear in Eq. (1)?

## Novelty and Significance (W2)

In terms of novelty, my understanding is that related results have been described for ReLU networks, but the authors generalize and extend these results to maxout networks. The main practical suggestion of the paper is the initialization strategy, which was also proposed in [1] already, but this work provides theoretical justification for this choice.

In terms of significance, it is not obvious to me how significant the results are, given the limited adoption of maxout networks (although I am not an expert here). However, theoretically, the paper generalizes and extends the results of ReLU network analysis, which is valuable.

## Experiments (W3)

The experiments show promising results, although the immediate practical significance is limited. The authors admit that they do not aim to achieve state of the art results, and instead aim to cleanly test the effect of the proposed regularization strategy. It is unclear if the initialization would have a similar effect if the models were using batch normalization.



**Strength And Weaknesses:**

## Strengths

The paper provides a serious study with multiple results on various properties of the maxout networks, including some practically relevant recommendations. The results of the analysis are presented relatively clearly, and the implications are discussed when possible.

The analysis also suggests an initialization strategy that the authors demonstrate works better than naive initialization.

## Weaknesses

**W1**: I don't fully understand why the terms in Eq. (1) are decomposed the way they are, and why the authors focus on the analysis of the term $\|J_{\mathcal{N}}(x^{(l)}) u\|$ and $\|x^{(l)}\| / n_l$.

**W2**: As the authors mention, the particular initialization strategy derived was already proposed for maxout networks in [1].

**W3**: The significance of the empirical results is not obvious, as the authors remove batch normalization and consequently do not achieve particularly outstanding results for the respective benchmarks.

**Discalimer.** I am not an expert in this area, and not familiar with other works. The paper is very technical and I cannot judge the significance of most of the derived theoretical results. Please assign a low weight to my review.

**Summary Of The Paper:**

The paper derives multiple theoretical insights into the properties of maxout neural networks at initialization. In particular, they derive results for the distribution of Jacobian-vector product norms, number of linear regions, NTK, and curve length distortion. They also provide recommendations for initialization of maxout networks based on their analysis, leading to significantly improved performance when trained with SGD and Adam compared to naive He initialization.

**Summary Of The Review:**

Overall, this is an interesting and dense paper deriving multiple new results for maxout networks. I don't have the background to fully appreciate the contributions of the paper, as I indicated above. I am currently voting for a weak accept, but with a low confidence score.

## References

[1] [_On the Expected Complexity of Maxout Networks_](https://arxiv.org/abs/2107.00379);
Hanna Tseran, Guido Montúfar

---

> ### Author Response · Authors · 2022-11-18
> **Response to Reviewer RjF3 / Part 3**
>
> > C4 (W3): The experiments show promising results, although the immediate practical significance is limited. The authors admit that they do not aim to achieve state of the art results, and instead aim to cleanly test the effect of the proposed regularization strategy. It is unclear if the initialization would have a similar effect if the models were using batch normalization.
>
> The main point of the experiments we present is to demonstrate that our proposed initialization strategy allows us to train deep and realistically wide maxout networks and achieve higher accuracy than other initialization approaches. The experiments were not designed to beat the state of the art on, e.g., MNIST and CIFAR, which are well-studied and well-solved datasets. Instead, the experiments are designed in the way that allows seeing the specific effects of different initializations. The experiments show that other initialization strategies did not progress or reach a lower or comparable accuracy as our approach. Of course, ultimately, in applications, if we want to obtain the highest possible accuracy, we need to incorporate additional tricks and methods. Following your comments, we have started experiments to test batch normalization. We have observed that our initialization strategy still is beneficial when training with batch normalization. We are currently pursuing further experiments in this direction and will seek to add them if time permits.
>
> Table 1. Accuracy on the test set for networks with batch normalization trained using SGD with Nesterov momentum.  Here we have fixed the learning rate across choices of $c$ and will pursue individual tuning of the learning rate if time permits. All networks are $8$ times narrower than the networks used in the SGD experiments in the submission.
>
> | VALUE OF $c$ | $10^{-6}$ | $10^{-5}$ | $10^{-4}$ | $0.001$ | $0.01$ | $0.1$ | $0.55555$ |  $1$ | $10$ | $100$ | $1000$ | $10^4$ | $10^5$ | $10^6$ |
> |---|---|---|---|---|---|---|---|---|---|---|---|---|---|---|
> | CONVOLUTIONAL |||||||||||||||
> | MNIST | $11.35^{\pm 0}$ | $11.35^{\pm 0}$ |$99.33^{\pm 0.09}$ | $99.35^{\pm 0.05}$  |$99.32^{\pm 0.05}$ |$99.36^{\pm 0.04}$ |$99.41^{\pm 0.07}$ |$99.39^{\pm 0.04}$ |$99.35^{\pm 0.02}$ | $98.83^{\pm 0.07}$ | $97.69^{\pm 0.31}$ | $95.11^{\pm 1.4}$ | $93.09^{\pm 1.88}$ | $87.63^{\pm 1.86}$ |
> | CIFAR-10 | $10^{\pm 0}$ | $10^{\pm 0}$ | $10^{\pm 0}$ | $74.85^{\pm 3.29}$ | $75.72^{\pm 4.94}$ | $77.16^{\pm 1.84}$ | $77.68^{\pm 1.07}$ | $79.26^{\pm 0.76}$ | $75.82^{\pm 1.05}$ | $66.23^{\pm 1.69}$ | $50.97^{\pm 2.28}$ | $43.81^{\pm 1.8}$ | $39.27^{\pm 2.73}$ | $40.27^{\pm 0.92}$ |
>
> On this note, we believe that the investigation of the interplay between parameter initialization and batch normalization is a very interesting direction to pursue. At the same time, we observe that this entails several theoretical and practical questions that certainly would merit an individual article. For example, theoretical work on the convergence of gradient descent with weight normalization [1] could demonstrate that the speed of convergence of ReLU networks trained with Weight Norm is affected by the initialization in non-trivial ways that are not yet fully understood. Or the work of [2], which shows that normalization methods can have non-trivial behavior depending on the magnitude of the parameters and added regularizers. We can point out that initialization strategies like ours (Xavier and He) and batch normalization are similar in the sense that they are both trying to normalize a summary of the distribution of pre-activations at different layers. One obvious difference, however, is that initialization may do this a priori and once, whereas dynamic normalization methods repeatedly normalize. Batch normalization can be costly, so simple initialization can be preferential, which might come with its downsides.
>
> We may reiterate that our main results are theoretical and that we derived several relevant implications from our analysis, including initialization, number of linear regions, and NTK. We believe that there is still a lot of interesting work to be done in regard to the theoretical analysis and practical implementation of maxout networks. We think that our article contributes in several important ways to such a program, and of course, acknowledge and hope you will agree that a single article cannot possibly cover all the possible directions of investigation.
>
> Thanks again for your valuable time.
>
> [1] Dukler, Yonatan, Quanquan Gu, and Guido Montúfar. "Optimization theory for relu neural networks trained with normalization layers." International conference on machine learning. PMLR, 2020.
>
> [2] Lobacheva, Ekaterina, et al. "On the periodic behavior of neural network training with batch normalization and weight decay." Advances in Neural Information Processing Systems 34 (2021): 21545-21556.

---

> > ### Comment · Reviewer_RjF3 · 2022-12-05
> > **Thank you for the response**
> >
> > Dear authors, thank you for the response, new experiments, updates to the paper and discussion. I am satisfied with the rebuttal. As I mentioned in my review, I am not an expert in this area, so I maintain the score of 6 with a low confidence.

---

> ### Author Response · Authors · 2022-11-18
> **Response to Reviewer RjF3 / Part 2**
>
> > C2 (W2): In terms of novelty, my understanding is that related results have been described for ReLU networks, but the authors generalize and extend these results to maxout networks. The main practical suggestion of the paper is the initialization strategy, which was also proposed in [1] already, but this work provides theoretical justification for this choice.
>
> The initialization strategy we obtain recovers precisely the initialization proposed by Tseran \& Montúfar (2021). The main difference is that their recommendation was heuristically motivated (in the sense of unrealistic assumptions on the input distribution to each layer), whereas we provide a theoretically justified derivation for wide and deep networks. Our analysis clarifies the difference between wide and narrow networks as well as the behavior of the gradient in maxout and ReLU networks. This allows us, in particular, to formulate architecture recommendations (in Section 4), which was not possible using the previous heuristic approach. Additionally, we introduce an initialization for the networks that use max-pooling layers, which was not present in Tseran \& Montúfar (2021).
>
> > C3 (W2): In terms of significance, it is not obvious to me how significant the results are, given the limited adoption of maxout networks (although I am not an expert here). However, theoretically, the paper generalizes and extends the results of ReLU network analysis, which is valuable.
>
> We appreciate your positive view on the generalization and extension of previous results. Indeed, maxout networks are less commonly used than, for example, ReLU networks. We believe that developing theory beyond those cases we are already familiar with is an important endeavor nonetheless. This is a typical trade-off of exploration and exploitation. The development of activation functions is still a crucial topic, and diversity beyond existing settings can facilitate important advances. As we pointed out in the introduction, one reason why maxout networks might not have been as popular in the past is that the initialization schemes for these models had yet to be studied. Much of the success of other architectures hinges on the fact that considerable research effort went into better understanding how to initialize and optimize their parameters. This program has been conducted to a lesser extent for the case of maxout networks. Our work demonstrates that significant improvements are possible once maxout networks are initialized using an appropriate initialization procedure. We believe that we can also learn from maxout networks for other noteworthy cases. Indeed, maxout networks are a prototype of architectures where multi-argument functions are used, such as graph neural networks, which we mentioned in the discussion of future work.

---

> ### Author Response · Authors · 2022-11-18
> **Response to Reviewer RjF3 / Part 1**
>
> Thank you for the helpful feedback and for appreciating the rigor of our results. We are glad to hear that you valued the clarity of the presentation and also the recommendations we offered, and the implication we obtained. Please see below for our response to your comments.
>
> >C1 (W1): I have a few questions about the discussion of Eq. (1) on page 3. First, what is the motivation behind the decomposition $\langle \mathcal{L}(\mathcal{N}(\mathbf{x})), \mathbf{J}\_{\mathcal{N}} (\mathbf{x}_i^{(l)}) \rangle = C(\mathbf{x}, W) || \mathbf{J}\_{\mathcal{N}} \left(\mathbf{x}^{(l)}\right) \mathbf{u} || $ ? You use $\mathbf{u} = \mathbf{e}_i$, so $|| \mathbf{J}\_{\mathcal{N}} \left(\mathbf{x}^{(l)} \right) \mathbf{u} || = || \mathbf{J}\_{\mathcal{N}} \left(\mathbf{x}^{(l)}_i \right) ||$, correct? Then, the term $C(\mathbf{x}, W)$ is given by $\langle \mathcal{L}(\mathcal{N}(\mathbf{x})), \mathbf{J}\_{\mathcal{N}} (\mathbf{x}_i^{(l)}) \rangle / || \mathbf{J}\_{\mathcal{N}} \left(\mathbf{x}^{(l)}_i\right) ||$. In other words, you multiply and divide by $|| \mathbf{J}\_{\mathcal{N}} \left(\mathbf{x}^{(l)}\right) \mathbf{u} ||$, but then this term is the center of your analysis.
> Throughout the paper, you seem to assume that the term  is the most important term in the Eq. (1). Why is this justified, given that it was introduced artificially? Why can we ignore $C(\mathbf{x}, W)$?
> Then, you say that the other term of interest is $A^{(l)} = || \mathbf{x}^{(l)} ||^2 / n_l$. Where did this term come from? Where does it appear in Eq. (1)?
>
> Your understanding is correct about what $||\mathbf{J}_{\mathcal{N}} \left(\mathbf{x}^{(l)}\right) \mathbf{u} ||$ and $C(\mathbf{x},W)$ are (except your comment is missing $\nabla$ in front of $\mathcal{L}$).
>
> The reason the decomposition makes sense is that at each layer, we are investigating the magnitude of $\frac{\partial \mathcal{L} (\mathbf{x})}{ \partial W_{i,k',j}^{(l)}}$.
> The reason we focus on the Jacobian norm rather than on $C$ is as follows.
> We have
>
> $\frac{\partial \mathcal{L} (\mathbf{x})}{ \partial W\_{i,k',j}^{(l)}} = \langle \nabla\_\mathcal{N} \mathcal{L}(\mathcal{N}(\mathbf{x})),
> \mathbf{J}\_\mathcal{N}(W\_{i,k',j}^{(l)})\rangle $
>
> $= \langle \nabla\_\mathcal{N} \mathcal{L}(\mathcal{N}(\mathbf{x})),
> \mathbf{J}\_{\mathcal{N}} (\mathbf{x}\_i^{(l)}) \rangle  \mathbf{x}\_j^{(l-1)}$
>
> $= \langle \nabla_\mathcal{N} \mathcal{L}(\mathcal{N}(\mathbf{x})),
> \mathbf{J}\_{\mathcal{N}} (\mathbf{x}^{(l)}) \mathbf{u} \rangle  \mathbf{x}\_j^{(l-1)} , \quad \text{$\mathbf{u}=\mathbf{e}\_i$ }$
>
> $= C(\mathbf{x},W) || \mathbf{J}\_{\mathcal{N}} (\mathbf{x}^{(l)}) \mathbf{u} || \mathbf{x}\_j^{(l-1)}.$
>
> Note that $C(\mathbf{x},W) = \langle \nabla\_\mathcal{N} \mathcal{L}(\mathcal{N}(\mathbf{x})),
> \mathbf{v} \rangle$ with $\mathbf{v} = \mathbf{J}\_{\mathcal{N}} \left(\mathbf{x}^{(l)}\right) \mathbf{u} / ||\mathbf{J}\_{\mathcal{N}} \left(\mathbf{x}^{(l)}\right) \mathbf{u}||$, $||\mathbf{v}||=1$.
> Hence $C(\mathbf{x},W)\leq ||\nabla\_\mathcal{N} \mathcal{L}(\mathcal{N}(\mathbf{x}))|| ||\mathbf{v} || = ||\nabla\_\mathcal{N} \mathcal{L}(\mathcal{N}(\mathbf{x}))||$.
> The latter term does not directly depend on the specific parametrization nor the specific architecture of the network but only on the loss function and the prediction.
>
> In view of the description of $\frac{\partial \mathcal{L} (\mathbf{x})}{ \partial W\_{i,k',j}^{(l)}}$, the variance depends on the square of $\mathbf{x}\_j^{(l-1)}$. Similarly, the variance of the gradient $\nabla\_{W^{(l)}\_{i,:}}\mathcal{L}(\mathbf{x}) = (\frac{\partial \mathcal{L} (\mathbf{x})}{ \partial W\_{i,k',j}^{(l)}})_{j=1}^{n\_{l-1}}$ depends on $\mathbf{x}^{(l-1)} = (\mathbf{x}\_j^{(l-1)})\_{j=1}^{n\_{l-1}}$ and thus depends on $\|\mathbf{x}^{(l-1)}\|^2 $.
> This is how activation length appears in Eq. (1).
>
> We hope this clarifies your questions. Motivated by your comments, we have also added these clarifications in Appendix A.

---

### Official Review · Reviewer_Fz1h · 2022-10-29

**Confidence:** 2
**Correctness:** 2
**Technical Novelty And Significance:** 3
**Empirical Novelty And Significance:** 3
**Recommendation:** 6

**Clarity, Quality, Novelty And Reproducibility:**

The novelty and quality are on very good level. Unfortunately  the paper is very hard and difficult to read.

**Strength And Weaknesses:**

1. The introduction paragraph, "Maxout networks," is not clear.
2. The introduction section is long and describes simple concepts, like Parameter initialization, and does not specify details of Maxout networks.
3. The contributions of the paper are long and need to be more convincing. After reading the introduction, it is challenging to specify the paper's main result.
4. Also, in section 2, the paragraph Architecture needs to be clarified. Some figures may help.
5. Theoretical results from section 3 Results are interesting but nontrivial to follow. Authors try to show too many things in very short lines.
6. The paper is longer than 9 pages.
7. The experimental section confirms the results from the theoretical part.


**Summary Of The Paper:**

In the paper, authors analyze the gradients of a maxout network with respect to inputs and parameters. In consequence, the authors improve the training of deep maxout networks in the case of fully-connected and convolutional networks.

**Summary Of The Review:**

Most of the results from the paper are interesting, but the article's presentation and construction of the article caused the paper to be very hard and difficult to read. The paper is better suited to a journal, where we do not have a limit of pages and can describe everything clearly.

---

> ### Author Response · Authors · 2022-11-18
> **Response to Reviewer Fz1h**
>
> Thank you for the valuable feedback and for appreciating the novelty and quality of the results. Please see below for our response to your comments.
>
> >C1: The introduction paragraph, "Maxout networks," is not clear.
>
> We had deferred a more formal definition to Section 2.
> Following your comment, to provide more clarity in the introductory paragraph, we have added a more explicit definition of maxout units, as shown below.
>
> A rank-$K$ maxout unit with $n$ inputs implements a function $\mathbb{R}^n\to\mathbb{R}$; $\mathbf{x} \mapsto \max_{k \in [K]} \{ \langle W_{k}, \mathbf{x} \rangle + b_{k}\}$, where $W_{k} \in \mathbb{R}^{n}$ and $b_{k} \in \mathbb{R}$, $k \in [K] := \{1, \ldots, K\}$, are trainable weights and biases.
> The $K$ arguments of the maximum are called the pre-activation features of the maxout unit.
>
> >C2: The introduction section is long and describes simple concepts, like Parameter initialization, and does not specify details of Maxout networks.
>
> Our work advances on several fronts, and hence we need to provide context on several different and relevant related works.
> Kindly note that the introduction section is under two pages, including related works and a summary of contributions, which is a fairly standard length.
> In the introduction, we provided an informal description of maxout networks and key references, deferring more formal definitions to Section 2. As mentioned above, we have now added an explicit definition of maxout units in the introduction.
>
> >C3: The contributions of the paper are long and need to be more convincing. After reading the introduction, it is challenging to specify the paper's main result.
>
> Your point is well taken. Indeed, our work makes several contributions, which we think is one of the strengths of our work.
>
> To aid clarity, we have now simplified the language in the paragraph Contributions in the introduction and summarized the main contributions into three categories that we have highlighted in boldface: expected gradients, parameter initialization, and several implications.
>
> Also, note that the main results are highlighted in the title and by using theorem environments. Further implications, particularly for the expected number of linear regions, expected curve distortion, and NTK use corollary environments.
>
> >C4: Also, in section 2, the paragraph Architecture needs to be clarified. Some figures may help.
>
> We appreciate your comment and, at the same time, observe that fully-connected networks are reasonably standard, which is why we did not consider it necessary to include a figure.
> However, indeed, maxout networks are less frequently encountered than single argument activation function networks. For clarity, we are adding a more explicit definition as described in the answer to comment C1 above. Furthermore, following your suggestion, we have added a figure in the appendix and an illustration of a maxout network for additional clarity.
>
> >C5: Theoretical results from section 3 Results are interesting but nontrivial to follow. Authors try to show too many things in very short lines.
>
> Given that the topics discussed in Section 3 attract substantial interest, we consider it worthwhile to state them in the main part of the article.
> Our main contributions are introduced in Theorems 1 and 3, and Corollary 2 on the moments is essential for the results presented in Section 5.
> In our article, we worked on giving a complete and transparent but indeed concise presentation of our results, relegating technical discussions to different appendices.
> We hope that the added definitions and simplified summary of the main results that we provided in the revision facilitate easier reading.
> Nonetheless, if the reviewers and AC insist, we can also offer to move some of the corollaries to the appendix.
>
> >C6: The paper is longer than 9 pages.
>
> Kindly note that the only part that extends beyond page 9 is the optional reproducibility statement and that this does not count toward the page limit as per the ICLR guidelines; see https://iclr.cc/Conferences/2023/AuthorGuide.
> Hence, the paper satisfies the page limit.
>
> >C7: Correctness: 2: Several of the paper’s claims are incorrect or not well-supported.
>
> This seems to contradict another comment from the reviewer stating that novelty and quality are on a very good level. We put significant care into our work and take pride in providing complete proofs and supporting all our claims. Therefore, we ask the reviewer to kindly specify which claims they consider incorrect or not well supported.

---

> > ### Comment · Reviewer_Fz1h · 2022-11-18
> > **I raise my score**
> >
> > The authors answered all my questions; therefore, I raised my score.

---

### Official Review · Reviewer_pCnC · 2022-11-01

**Confidence:** 4
**Correctness:** 3
**Technical Novelty And Significance:** 3
**Empirical Novelty And Significance:** 1
**Recommendation:** 5

**Clarity, Quality, Novelty And Reproducibility:**

The paper is well-written, has good clarity, and the authors provide proofs for their theorems. The theoretical results in the paper are original, to the best of my knowledge.

The authors have not provided source code of their experiments to reproduce their results, though they provide a detailed description of the experiment configurations in the appendix.

—————

Update:

I thank the authors for their clarification on the anonymous code repository to reproduce their experiments, and appreciate their contribution to the open science community.

**Strength And Weaknesses:**

Strength:
- Intuitively, maxout networks are more difficult to analyze. Therefore compared with the analysis of ReLU network, it is a step forward.
- The paper is well-organized and the non-technical part is easy to follow.

Weakness:
- It seems that all the questions (Jacobian moments, proper initialization scale, number of linear regions, curve length distortion, NTK) and techniques to answer them have appeared in previous work. The originality of the contributions and the theoretical impact would both be limited.
- Maxout networks (especially the fully-connected version) are not widely used in practice, to the best of my knowledge. Therefore the practical interest of this work would be limited.

Some comments on the experiment section:
- [1] reports a simple linear kernel achieved 1.2% test error on MNIST, which is better that all the fully-connected network performance reported in this paper.
- As a rank-5 maxout network is 5 times as large as a ReLU network with the same architecture, I wonder if their results are directly comparable. Maybe a fair comparison should try to equal the number of parameters of different networks or else indicate their different sizes.

[1] Understanding Deep Learning (Still) Requires Rethinking Generalization. https://dl.acm.org/doi/pdf/10.1145/3446776

**Summary Of The Paper:**

In this paper, the authors study theoretical questions regarding the training of randomly initialized fully-connected neural networks with **maxout** nonlinear activations. In particular, they derive the moments of the input-output Jacobian of the network, which motivates a parameter initialization strategy that could avoid the exploding and vanishing gradient problem in wide networks. They validate their results by experiments, and also derive some other theoretical properties, such as bounds on the expected number of linear regions, the expected curve length distortion, and neural tangent kernels.

**Summary Of The Review:**

I think the paper contains original theoretical results which contribute to people's understanding of maxout networks. However, the theoretical and practical impact of this work may be limited, and the technical questions and proof techniques have appeared in previous work. Therefore, I tend to argue the paper is marginally below the bar of the NeurIPS venue.

—————

Update:

After reading the authors’ rebuttal, I would like to keep my suggestion that the paper is marginally below NeurIPS’s standard, due to my personal judgment on its potential impact, and I am raising my confidence score from 2 to 4.

That said, I very much appreciate the authors’ discussion and clarification during the rebuttal phase.

---

> ### Author Response · Authors · 2022-11-18
> **Response to Reviewer pCnC / Part 3**
>
> > C3: [1] reports a simple linear kernel achieved 1.2\% test error on MNIST, which is better that all the fully-connected network performance reported in this paper.
>
> Kindly note that the purpose of this experiment was not to succeed with MNIST, which is already a very well-studied and well-solved dataset. There are indeed simple methods that can achieve high accuracy on MNIST. Instead, the main point of the experiments we present with fully-connected networks is to demonstrate that our proposed initialization strategy allows us to train deep and realistically wide fully-connected networks and achieve higher accuracy than other initialization approaches. And indeed, the experiments show that other initialization strategies did not progress or reach a lower or comparable accuracy as our approach.
>
> > C4: As a rank-5 maxout network is 5 times as large as a ReLU network with the same architecture, I wonder if their results are directly comparable. Maybe a fair comparison should try to equal the number of parameters of different networks or else indicate their different sizes.
> [1] Understanding Deep Learning (Still) Requires Rethinking Generalization. https://dl.acm.org/doi/pdf/10.1145/3446776
>
> Please note that in the experiments, rather than comparing architectures, we aim to test whether the suggested initialization procedure allows training deep and realistically wide maxout networks. Since comparing ReLU and maxout networks is not our primary goal, the results for the ReLU networks are provided mainly for reference, and we use the networks with the same architecture.
>
> We should point out that what is a fair comparison is not as straightforward as matching the parameter count. In particular, wider networks have the advantage of having a higher dimensional representation. A fully connected network will not necessarily perform as well as a convolutional network with the same number of parameters, and a deep and narrow network will not necessarily perform as well as a wider and shallower network with the same number of parameters.
>
> Nevertheless, to add more details to the results, as suggested by the reviewer, we started experiments using ReLU networks that have as many parameters as maxout networks. See Table 1 for preliminary results. In the first experiment, we use fully connected ReLU networks 5 times wider than maxout networks. For convolutional networks, however, the resulting CNNs with ReLU activations would be extremely wide, so we only made it 2 times wider. In our setup, a 5 times wider CNN network would need to be trained for longer than 24 hrs, which is difficult in our experiment environment. Maxout networks only required a much shorter time of around 10 hours, which indicates possible benefits in some cases. In the second experiment, we consider ReLU networks that are 5 times deeper than the maxout networks. As expected, wider networks do better. On the other hand, deeper ReLU networks of the same width do much worse than maxout networks. We will continue to run more experiments post-rebuttal and add new results to the Appendix.
>
> Table 1.  Accuracy on the test set for ReLU networks that are $5$ times _wider_ than maxout networks trained using SGD with Nesterov momentum.
> | | MAXOUT | RELU |
> |---|---|---|
> | | Maxout init | He init |
> | VALUE OF $c$ | $0.55555$ | $2$ |
> | FULYY-CONNECTED ||
> | MNIST | $97.8^{\pm 0.15}$ | $\mathbf{98.11^{\pm 0.02}}$ |
> | IRIS | $91.67^{\pm 3.73}$ | $\mathbf{92.5^{\pm 2.76}}$|
> | CONVOLUTIONAL ||
> | MNIST | $\mathbf{99.59^{\pm 0.04}}$ | $99.54^{\pm 0.02}$ |
>
>
> Table 2. Accuracy on the test set for ReLU networks that are $5$ times _deeper_ than maxout networks trained using SGD with Nesterov momentum.
> | | MAXOUT | RELU |
> |---|---|---|
> | | Maxout init | He init |
> | VALUE OF $c$ | $0.55555$ | $2$ |
> | FULYY-CONNECTED ||
> | MNIST | $\mathbf{97.8^{\pm 0.15}}$ | $63.47^{\pm 33.32}$ |
>
> > C5: The authors have not provided source code of their experiments to reproduce their results, though they provide a detailed description of the experiment configurations in the appendix.
>
> Please note that, in actuality, we made computer code available with our submission. As we indicated in Section 6, the code for the main routines is provided at https://anonymous.4open.science/r/maxout_expected_gradient-68BD. This kind of anonymous repository is the standard way to make computer code available with article submissions. We take reproducibility very seriously, and that is why we included a detailed description of experiment configurations, as the reviewer noticed, but also included the computer code. We hope this clarification will also factor into the reviewer's final recommendation.

---

> ### Author Response · Authors · 2022-11-18
> **Response to Reviewer pCnC / Part 2**
>
> > C2: Maxout networks (especially the fully-connected version) are not widely used in practice, to the best of my knowledge. Therefore the practical interest of this work would be limited.
>
> Indeed, maxout networks are less commonly used than, for example, ReLU networks. We believe that developing theory beyond those cases we are already familiar with is an important endeavor nonetheless. This is a typical trade-off of exploration and exploitation. The development of activation functions is still a crucial topic, and diversity beyond existing settings can facilitate important advances. As we pointed out in the introduction, one reason why maxout networks might not have been as popular in the past is that the initialization schemes for these models had yet to be studied. Much of the success of other architectures hinges on the fact that considerable research effort went into better understanding how to initialize and optimize their parameters. This program has been conducted to a lesser extent for the case of maxout networks. Our work demonstrates that significant improvements are possible once maxout networks are initialized using an appropriate initialization procedure. We believe that we can also learn from maxout networks for other noteworthy cases. Indeed, maxout networks are a prototype of architectures where multi-argument functions are used, such as graph neural networks, which we mentioned in the discussion of future work.
>
> As for the fully connected setting, indeed, in many cases, such as image classification, fully-connected architectures are less popular than convolutional architectures. In many previous works, one has first developed a theory for the fully-connected case and then extended the analysis to the convolutional case. In this work, we wanted to provide a wide range of results for maxout networks concerning expected gradients. Ultimately we considered that it is more important at this stage to build coherent and insightful implications geared toward parameter initialization. As has been the case in previous studies of other architectures, we have confidence that many of the theoretical results we have derived for the fully connected case will translate to the convolutional case. Our experiments support this. We hope you will agree that our work already includes many theoretical results and implications and that we cannot realistically address every possible topic in a single article.
>
> Nevertheless, we think that the convolutional network case is doable. It will take us some time to work out the details, but we will seek to add this if time permits.

---

> ### Author Response · Authors · 2022-11-18
> **Response to Reviewer pCnC / Part 1**
>
> Thank you for the valuable feedback and for appreciating the difficulty of obtaining results for maxout networks. Please see below for our response to your comments.
>
> > C1: It seems that all the questions (Jacobian moments, proper initialization scale, number of linear regions, curve length distortion, NTK) and techniques to answer them have appeared in previous work. The originality of the contributions and the theoretical impact would both be limited.
>
> Naturally, our work builds on several of the techniques and lessons learned in the community over the years. However, the analysis is novel and nontrivial, and our work makes several contributions to different topics of interest. Generally, since the distribution of the gradients of maxout networks exhibits a dependency on the network inputs, the analysis differs substantially from the ReLU case. The core of the issue is that while for ReLU networks, it is possible to write the equality in distribution that is independent of the network input [1], this is not the case for maxout networks.
>
> More specifically, this leads to the following differences. Theorem 1 uses stochastic orders to bound gradients for any input and order statistics to accommodate that maxout takes a maximum of several arguments. Maxout networks lead to order statistics (a random variable defined as the maximum of a collection of random variables) in Corollary 2. In contrast, order statistics are not needed for ReLU networks, where it is possible to use the readily available formulas for the moments of the chi-squared distribution [2]. Theorem 3 pursues an approach based on a change of basis for the vectors based on the network input $\mathbf{x}$ and vector $\mathbf{u}$ to write the equality in distribution and clarify the dependency on the input, a concept that was not needed for ReLU networks since they do not have this dependency. Furthermore, the analysis in Remark 4 is based on the analysis of the stable points of the angle between these two vectors, which is again developed for maxout networks and is not needed for ReLU networks.
>
> The versatility and usefulness of our results are highlighted by the several implications that we derived, from parameter initialization to the expected number of linear regions, NTK, and distortion length. While the derivations are similar to results that have previously appeared in the literature, they can only be obtained once the gradient results are in place. We think that these results do have a practical interest in the exploration and use of maxout networks.
>
> We believe that the new strategies that we have developed to investigate maxout networks are significantly different from previous works, interesting in their own right and that they can help study other activation functions beyond ReLU and maxout units.
>
> [1] Hanin, Boris, and Mihai Nica. "Products of many large random matrices and gradients in deep neural networks." Communications in Mathematical Physics 376.1 (2020): 287-322.
>
> [2] Boris Hanin, Ryan Jeong, and David Rolnick. Deep ReLU Networks Preserve Expected Length. arXiv:2102.10492 [cs, stat], June 2021. URL http://arxiv.org/abs/2102.10492. arXiv: 2102.10492.

---

> ### Author Response · Authors · 2022-11-25
> **Response to Reviewer pCnC**
>
> Dear Reviewer,
>
> We made thorough efforts to address all your concerns in our initial response. We hope you will find it convincing, and our response might motivate you to raise your score. Kindly let us know if there are any remaining items that you would like to see addressed. We remain attentive to your feedback!

---

> ### Comment · Area_Chair_ZGsw · 2022-12-13
> **Notice**
>
> Dear Reviewer qSsw,
>
> If you do not want to update your score, please at least acknowledge that you have already read the rebuttal.

---

### Official Review · Reviewer_uU8d · 2022-11-03

**Confidence:** 2
**Correctness:** 4
**Technical Novelty And Significance:** 3
**Empirical Novelty And Significance:** 2
**Recommendation:** 6

**Clarity, Quality, Novelty And Reproducibility:**

Clarity: Good
Quality: Medium
Novelty: Medium
Replicability: codes are provided

**Strength And Weaknesses:**

Pros:
1. Interesting theoretical studies from various perspectives, including bounds for directional derivative, moments, distribution of input-output Jacobian.
2. Good implications that refines previous bounds on the expected number of linear regions, and new results on length distortion and the NTK.
3. Justification for the parameter initialization strategy and hyperparameter selection are reasonable.

Cons:
1. Empirical ablation studies are needed to verify the selection strategy of hyperparameters. The authors compute a range of combinations of K and c given $c = 1 / M$. It would be great to test if the selected combination works better than others. For example, you can fix K=5 and vary a range of c to test the performance.
2. Is there any difference between the proposed initialization and the original one proposed by Tseran & Montúfar (2021)? It would good to see if the original strategy can be improved by the new theoretical understanding.
3. About max-pooling initialization, why initialize some layers as $K \times m^2$? Some motivations are needed since the authors report it achieves better performance.
4. More baseline strategies and architectures are needed to verify the effectiveness of the proposed initialization strategy. Currently the authors use VGG and He Initialization as a comparison, but He Init is designed for ResNet and it could be more effective under ResNet. So it would be great to compare maxout and Relu under ResNet, which is more like an apple-to-apple comparison. In addition, since the authors aim for a batch normalization-free network setting, comparing the proposed strategy with recent initialization methods targeting on this setting would be more fair, such as Fixup, Rezero.

**Summary Of The Paper:**

The authors study the gradients of a maxout network with respect to inputs and parameters. Based on their results, they also obtain refined bounds on the expected number of linear regions, results on the expected curve length distortion, and results on the NTK. With their theoretical understanding, they justify a parameter initialization strategy that avoid vanishing and exploding gradients in maxout networks. They also empirically verify the success of their initialization strategy on multiple datasets.

**Summary Of The Review:**

Overall I believe the paper is slightly above the acceptance level, and I would like to raise my score if the authors address the above concerns.

---

> ### Author Response · Authors · 2022-11-18
> **Response to Reviewer uU8d / Part 2**
>
> > C4: More baseline strategies and architectures are needed to verify the effectiveness of the proposed initialization strategy.
>
> Kindly note that our paper has a theoretical focus, and we obtain a diversity of results covering several topics, which we consider our main contribution. Nonetheless, we have indeed conducted a large range of experiments spanning fully connected and convolutional networks both for Adam and SGD, which we consider extensive, particularly in view of the theoretical focus of our work. While we generally agree that more experiments can often be instructive, we observe that our experiments already provide clear evidence in support of our main claims. We would like to ask if the reviewer has specific ideas of what additional insights could be provided by additional experiments or specific suggestions for additional strategies, keeping in mind the scope of our work, to please share these with us.
>
> > C5: Currently the authors use VGG and He Initialization as a comparison, but He Init is designed for ResNet and it could be more effective under ResNet.  So it would be great to compare maxout and Relu under ResNet, which is more like an apple-to-apple comparison.
>
> Please note that the He initialization was, in fact, developed for fully-connected and convolutional networks with ReLU activations, see [1]. It was not designed for ResNets. Furthermore, the paper [2] introducing ResNets was published after [1]. In the Initialization Recommendation paragraph on page 6, we had incorrectly pointed to [2] as the reference for He initialization, but the correct reference is [1]. This probably led to confusion, and we apologize for this. We are fixing the reference in the revision.
>
> > C6: In addition, since the authors aim for a batch normalization-free network setting, comparing the proposed strategy with recent initialization methods targeting on this setting would be more fair, such as Fixup, Rezero.
>
> Both Fixup [3] and ReZero [4] are designed for residual style architectures. In Fixup, the main differences to standard initialization procedures such as He are on the residual connections (Fixup initialization definition on page 5 of [3]), and ReZero proposes residual with zero initialization, whereby the zero is an added multiplicative weight on the function. Therefore a comparison with these methods would make the most sense in the context of residual networks.
> We think they are interesting and are adding pointers to our overview of Parameter Initialization procedures in the introduction. At the same time, while the development of residual maxout networks could be an interesting direction to pursue, in our work, we are focusing on feedforward networks without skip connections.
>
> Generally, the initialization approaches in the literature often focus on ReLU activation. There has been more limited work on maxout activation functions, which is one of the problems we are addressing with our work. We do not know of any other batch-normalization free methods designed specifically for feedforward maxout networks of any rank, and hence we compare our method to the one suitable for this setting and applicable to maxout networks, which is He initialization. Nonetheless, we think that ultimately developing counterparts of other initialization strategies for the case of maxout networks is a worthwhile and interesting endeavor for future work.
>
> [1] Kaiming He, Xiangyu Zhang, Shaoqing Ren, and Jian Sun. Delving Deep into Rectifiers: Surpassing Human-
> Level Performance on ImageNet Classification. In 2015 IEEE International Conference on Computer Vision
> (ICCV), pp. 1026–1034, Santiago, Chile, 2015. IEEE.
>
> [2] Kaiming He, Xiangyu Zhang, Shaoqing Ren, and Jian Sun. Deep residual learning for image recognition. In
> Conference on Computer Vision and Pattern Recognition, pp. 770–778. IEEE, 2016.
>
> [3] Zhang, Hongyi, Yann N. Dauphin, and Tengyu Ma. "Fixup initialization: Residual learning without normalization." arXiv preprint arXiv:1901.09321 (2019).
>
> [4] Bachlechner, Thomas, et al. "Rezero is all you need: Fast convergence at large depth." Uncertainty in Artificial Intelligence. PMLR, 2021.

---

> > ### Comment · Reviewer_uU8d · 2022-12-06
> > **Thanks for the response**
> >
> > I want to thank the authors for their detailed response. The authors address my concerns and questions, and I am willing to keep recommending the acceptance of the paper.

---

> > > ### Author Response · Authors · 2022-12-06
> > > **Re: Thanks for the response**
> > >
> > > Thanks very much for evaluating our rebuttal and recommending to accept our paper. Given that you found our response detailed and that it addressed your concerns and questions, we hope you don't mind us still following up on your initial comment "I would like to raise my score if the authors address the above concerns" and asking you to consider raising your score.

---

> ### Author Response · Authors · 2022-11-18
> **Response to Reviewer uU8d / Part 1**
>
> Thank you for the valuable feedback and for appreciating the theoretical results we obtain. Please see below for our response to your comments.
>
> > C1: Empirical ablation studies are needed to verify the selection strategy of hyperparameters. The authors compute a range of combinations of $K$ and $c$ given $c = 1/M$. It would be great to test if the selected combination works better than others. For example, you can fix $K=5$ and vary a range of $c$ to test the performance.
>
> Please note that in the submitted paper, we provided results for $c=0.1$ and $c=2$ for SGD with Nesterov momentum and $c=2$ for Adam. To add more details to these results following your suggestion, we have conducted additional SGD experiments for more values of $c$ and $K=5$, shown below. From this, we see that, indeed, the recommended value of $c$ from our theory closely matches the empirical optimum value of $c$. Here we show the average accuracy and std over $4$ runs. We have fixed the learning rate across choices of $c$ and will pursue individual tuning of the learning rate if time permits. We are adding these new results to Appendix I.
>
> |VALUE OF $с$ | $0.01$ | $0.05$ | $0.1$ | $0.3$ | $0.4$ | $0.5$ | $0.55555$ | $0.6$ | $0.7$ | $0.8$ | $1$ | $1.5$ | $2$ | $5$ | $10$|
> |---|---|---|---|---|---|---|---|---|---|---|---|---|---|---|---|
> | FULLY-CONNECTED ||||||||||||||||
> | MNIST |$11.35^{\pm 0}$ | $11.35^{\pm 0}$ | $11.35^{\pm 0}$ | $97.63^{\pm 0.16}$ | $97.89^{\pm 0.12}$ | $97.82^{\pm 0.09}$ | $97.92^{\pm 0.18}$ | $97.77^{\pm 0.17}$ | $97.91^{\pm 0.11}$ | $75.82^{\pm 38.12}$ | $97.89^{\pm 0.1}$ | $9.8^{\pm 0}$ | $9.8^{\pm 0}$ | $9.8^{\pm 0}$ | $9.8^{\pm 0}$ |
> | IRIS | $30^{\pm 0}$ | $30^{\pm 0}$ | $30^{\pm 0}$ | $60.83^{\pm 30.86}$ | $85^{\pm 10.67}$ | $92.5^{\pm 1.44}$ | $90.83^{\pm 3.63}$ | $90.83^{\pm 1.44}$ | $90^{\pm 0}$ | $30^{\pm 0}$ | $30^{\pm 0}$ | $30^{\pm 0}$ | $30^{\pm 0}$ | $30^{\pm 0}$ | $30^{\pm 0}$ |
> | CONVOLUTIONAL ||||||||||||||||
> | MNIST | $11.35^{\pm 0}$ | $11.35^{\pm 0}$ | $11.35^{\pm 0}$ |  $99.55^{\pm 0.02}$ | $99.6^{\pm 0.03}$ | $99.56^{\pm 0.05}$ | $99.57^{\pm 0.07}$ | $99.59^{\pm 0.02}$ | $54.69^{\pm 44.89}$ | $9.8^{\pm 0}$ | $9.8^{\pm 0}$ | $9.8^{\pm 0}$ | $9.8^{\pm 0}$ | $9.8^{\pm 0}$ | $9.8^{\pm 0}$ |
> | CIFAR-10 | $10^{\pm 0}$ | $10^{\pm 0}$ | $10^{\pm 0}$ | $90.97^{\pm 0.11}$ | $91.15^{\pm 0.07}$ | $91.33^{\pm 0.13}$ | $91.4^{\pm 0.22}$ | $91.69^{\pm 0.25}$ | $50.83^{\pm 40.83}$ | $10^{\pm 0}$ | $10^{\pm 0}$ | $10^{\pm 0}$ | $10^{\pm 0}$ | $10^{\pm 0}$ | $10^{\pm 0}$ |
>
> > C2: Is there any difference between the proposed initialization and the original one proposed by Tseran \& Montúfar (2021)? It would good to see if the original strategy can be improved by the new theoretical understanding.
>
> The initialization strategy we obtain recovers precisely the initialization proposed by Tseran \& Montúfar (2021). The main difference is that their recommendation was heuristically motivated (in the sense of unrealistic assumptions on the input distribution to each layer), whereas we provide a theoretically justified derivation for wide and deep networks. Our analysis clarifies the difference between wide and narrow networks as well as the behavior of the gradient in maxout and ReLU networks. This allows us, in particular, to formulate architecture recommendations (in Section 4), which was not possible using the previous heuristic approach.
>
> Moreover, in contrast to the work of Tseran and Montúfar (2021), which only presented experiments on fully connected networks, we conducted a much more comprehensive set of experiments covering both fully connected and convolutional networks.
>
> > C3: About max-pooling initialization, why initialize some layers as $K \times m^2$? Some motivations are needed since the authors report it achieves better performance.
>
> Thanks for bringing this up. The motivation for this is that the outputs of a computational block consisting of a max-pooling window and a maxout layer are taking maxima over $K \times m^2$ linear functions, as $\max(W_1 \max(\mathbf{x}_1, \ldots, \mathbf{x}_n) + \mathbf{b}_1, \ldots,  W_K \max(\mathbf{x}\_1, \ldots, \mathbf{x}\_{m^2}) + \mathbf{b}\_K) = \max(f\_1,\ldots, f\_{Km^2})$, where the $f_i$ are $Km^2$ affine functions. Therefore, we initialize the layers that follow max-pooling layers using the criterion for maxout rank $m^2 \times K$ instead of $K$. In our experiments, $K = 5$, $m = 2$, and $m^2 \times K = 20$. Hence, for such layers, we use the constant $c = 1/\mathcal{M} = 0.26573$, where $\mathcal{M}$ is computed for $K = 20$ using the formula from Remark 14 in Appendix C. All other layers that do not follow max-pooling layers are initialized as suggested in Section 4. Following your comment, we are adding a description of this motivation in Appendix I.

---

### Comment · Area_Chair_ZGsw · 2022-11-22
**Please respond as possible if you still have questions on the paper.**

Please respond as possible if you still have questions on the paper.

---

> ### Comment · Area_Chair_ZGsw · 2022-11-29
> **Please respond to the authors by Nov. 30**
>
> Hi Reviewers Review uU8d, pCnC and 3pQc,
>
> Please indicate whether the authors' rebuttal address your concerns.
>
> If you still have questions, please ask as soon as possible.

---

> > ### Comment · Area_Chair_ZGsw · 2022-12-05
> > **Zoom Meeting**
> >
> > For all reviewers, which have not responded to the authors, I will have to ask you to meet via Zoom. If you want to avoid such an additional step, please respond by Dec. 5.

---

> > > ### Comment · Reviewer_RjF3 · 2022-12-05
> > > **Response submitted**
> > >
> > > Hi everyone, I have read the rebuttal and the other reviews, as acknowledged [here](https://openreview.net/forum?id=PfHk0P9lgMy&noteId=o1ZeGcLnwYE). I maintain my score and recommend a weak accept with low confidence, as I am not an expert in the area.

---

### Decision · Program_Chairs · 2023-01-20

**Decision:**

Reject

**Justification For Why Not Higher Score:**

NA

**Justification For Why Not Lower Score:**

NA

**Metareview: Summary, Strengths And Weaknesses:**

This paper investigates the training of randomly initialized fully-connected neural networks with maxout nonlinear activations from a theoretical perspective. The authors derive the moments of the input-output Jacobian of the network, which leads to a proposed parameter initialization strategy to prevent the exploding and vanishing gradient problem in wide networks. They validate their results through experiments and also derive other theoretical properties, such as bounds on the expected number of linear regions, the expected curve length distortion, and neural tangent kernels.

Though several reviewers gave high ratings, unfortunately, they are either non-experts on related topics or less experienced researchers, and they chose low confidence scores as 2.  Reviewer pCnC giving 5 is an experienced expert on related topics. After reading the paper, I shared the same concerns as pCnC. The max-out paper studied in this paper is far less popular than other commonly used architectures, and its performance is also worse in general. The paper does not well motivated the proposed research. Why is working on this problem interesting? The response from the authors is not convincing for answering this question. Another concern is the theoretical part. It is a combination of results, which are derived based on existing technical tools. What are their implications? The paper does not explain why we should study these properties.

I believe there are some merits in the paper, but it is not in a good shape, and needs more work. The paper could be significantly strengthened if these concerns are addressed.

**Summary Of Ac-Reviewer Meeting:**

NA